# The genome of *Acorus* deciphers insights into early monocot evolution

Xing Guo[1,9], Fang Wang[1,2,9], Dongming Fang[1,9], Qiongqiong Lin[1,3], Sunil Kumar Sahu [1], Liuming Luo[1,3], Jiani Li[1,4], Yewen Chen[1,2], Shanshan Dong[5], Sisi Chen[6], Yang Liu[1,5], Shixiao Luo[6], Yalong Guo [2,7] & Huan Liu [1,8] ✉

Acorales is the sister lineage to all the other extant monocot plants. Genomic resource enhancement of this genus can help to reveal early monocot genomic architecture and evolution. Here, we assemble the genome of *Acorus gramineus* and reveal that it has ~45% fewer genes than the majority of monocots, although they have similar genome size. Phylogenetic analyses based on both chloroplast and nuclear genes consistently support that *A. gramineus* is the sister to the remaining monocots. In addition, we assemble a 2.2 Mb mitochondrial genome and observe many genes exhibit higher mutation rates than that of most angiosperms, which could be the reason leading to the controversies of nuclear genes- and mitochondrial genes-based phylogenetic trees existing in the literature. Further, Acorales did not experience tau (τ) whole-genome duplication, unlike majority of monocot clades, and no large-scale gene expansion is observed. Moreover, we identify gene contractions and expansions likely linking to plant architecture, stress resistance, light harvesting, and essential oil metabolism. These findings shed light on the evolution of early monocots and genomic footprints of wetland plant adaptations.

Monocots, along with magnoliids, eudicots, and two smaller clades (Ceratophyllales and Chloranthales), are one of the major clades of mesangiosperms. They are one of the most species-rich, ecologically dominant, and economically important of all land plant lineages, with about 85,000 species[1] in 77 families and 11–12 orders[2–4]. They have radiated into almost every terrestrial and aquatic habitat occupied by angiosperms since they arose 136–140 million years ago (Mya)[5,6], and exhibit remarkable morphological diversity, accounting for 21% of all angiosperm species, and directly or indirectly provide most of the diet of humans. Understanding their morphological differentiation, spatial diversification, and ecological radiation patterns is thus a major challenge for biologists[7]. Over the past three decades, our understanding of monocot relationships has substantially enhanced because of improvements in molecular systematics[8,9]. For instance, plastid gene sequences have overthrown various assumptions about the placement of specific genera and families, and have led to a radical reclassification of monocots at the family and order levels[2]. However, the monophyly of many monocot orders and families were not substantially supported by these phylogenetic studies. Therefore, to resolve these ambiguities, Givnish et al.[7] and several other scientists[10,11] performed

[1]State Key Laboratory of Agricultural Genomics, BGI-Shenzhen, Shenzhen, Guangdong 518083, PR China. [2]College of Life Sciences, University of Chinese Academy of Sciences, Beijing 100049, PR China. [3]College of Life Science, South China Agricultural University, Guangzhou, Guangdong 510642, PR China. [4]College of Life Sciences, Northwest University, Xi'an, Shaanxi 710069, PR China. [5]Key Laboratory of Southern Subtropical Plant Diversity, Fairy Lake Botanical Garden, Shenzhen & Chinese Academy of Sciences, Shenzhen, Guangdong 518004, PR China. [6]Key Laboratory of Plant Resource Conservation and Sustainable Utilization, The Chinese Academy of Sciences, South China Botanical Garden, Guangzhou, Guangdong 510650, PR China. [7]State Key Laboratory of Systematic and Evolutionary Botany, Institute of Botany, Chinese Academy of Sciences, Beijing 100093, PR China. [8]BGI Life Science Joint Research Center, Northeast Forestry University, Harbin, Heilongjiang 150040, PR China. [9]These authors contributed equally: Xing Guo, Fang Wang, Dongming Fang. ✉e-mail: liuhuan@genomics.cn

plastome or genome-scale phylogenomic studies coupled with a high-density taxon sampling throughout monocots, or individual orders and families[12,13]. These investigations provided well-supported monocot phylogenies, with the majority of clades completely resolved; however, uncertainties remain in some major lineages, including the relative placement of Liliales to Asparagales, and the identification of the sister lineage to the Poales.

*Acorus* (sweet flag) is the only genus of the Acoraceae family in the order Acorales[14–16], which are distributed from northern temperate to subtropical regions. Species of *Acorus* are found in wetlands and marshes, where they spread by means of thick rhizomes[17,18]. *Acorus* is among the most anoxia-tolerant plants[19,20], and the rhizome can support the plant's life in anoxic conditions for several months[19]. *Acorus* plants use various strategies to adapt to the wetland habitat and mitigate the damage caused by flooding, such as accelerating branch and leaf growth to avoid flood stress[21], growing slowly to reduce energy consumption, or relying on the number of nutrient reserves under long-term flooding[21,22]. *Acorus* was originally recognized as a member of the Araceae family before being treated as Acoraceae[17]. Although species in this genus have been widely used for medicinal, nutritional, ornamental, and ecological purposes[23,24], the taxonomy of *Acorus* remains unclear[25]. Four species, namely *Acorus calamus* L., *A. gramineus* Aiton, *A. tatarinowii* Schott, and *A. rumphianus* S. Y. Hu, were recorded in Flora Reipublicae Popularis Sinicae (FRPS)[26]. However, only two species, *A. calamus* (and varieties) and *A. gramineus*, were accepted in Flora of China (FOC)[17], the Plant List (http://www.theplantlist.org/), and International Plant Names Index (IPNI) (https://www.ipni.org/). In addition, *Acorus macrospadiceus* was described as a new species by Wei and Li[27], which was supported by recent phylogenetic and metabolomic analyses[28–30].

*Acorus* was consistently placed as the sister to all other monocots in previous phylogenetic studies based on nuclear[30] and chloroplast data[31–33], including discrete single or low-copy nuclear genes[34–36], large-scale chloroplast gene and 1kp transcriptome data[30,37]. The chromosome-scale genome of *A. tatarinowii* was recently published, which demonstrates how gene structural and functional characteristics constrain the ancestral gene order in monocots, and also places *Acorus* as the sister to all sequenced monocots based on 104 single-copy orthologues[38]. However, *Acorus* is considered as a relative of the core alismatids based on mitochondrial genome data[15,39,40]. Concerning the key phylogenetic position of *Acorus*, the question of why the phylogenetic result of the mitochondrial genome differs so much from nuclear and chloroplast genomes in the placement of *Acorus* is still unresolved.

Here, we employ a combination of ultra-long Nanopore reads, PacBio HiFi reads and BGI-DIPSEQ short reads, coupled with Hi-C technology, to generate a high-quality gap-free genome assembly of *A. gramineus*. Our results suggest that *A. gramineus* is the sister group to all the other monocots and high mutation rate is the likely cause of gene tree incongruence between nuclear and mitochondrial genomes. Moreover, *A. gramineus* has undergone genomic changes that are linked to its morphological structure, stress resistance, light harvesting, and essential oil metabolism. These findings provide insights into the evolution of early monocots and the genomic footprints of plant adaptation to wetlands.

## Results

### Genome sequencing, assembly, and annotation

The genome size of *A. gramineus* was predicted to be about 400.3 Mb by *k*-mer analysis[41] (Supplementary Fig. 1 and Supplementary Table 1), which was consistent with the estimate of ~391.2 Mb using flow cytometry[42,43]. Here, we report a high-quality, gap-free genome assembly using ~47.6 Gb of Nanopore long reads, ~20.0 Gb of Nanopore ultra-long reads, ~35.3 Gb of PacBio HiFi reads, ~261.2 Gb of BGI-SEQ short reads and 138.6 Gb Hi-C data (692,804,666 read pairs)

(Supplementary Data 1 and Fig. 1c). The final assembly had a total length of 399.8 Mb, and Hi-C assembly anchored them in 12 pseudo-chromosomes, corresponding to the number of chromosomes determined experimentally in somatic cells (2*n* = 2x = 24) (Supplementary Fig. 2). The 12 chromosomes were assembled as single-contig pseudomolecules without gaps, representing the high completeness of assembly, with a scaffold N50 of 36.5 Mb (Supplementary Table 2). The BUSCO assessment indicated 96.7% completeness (Supplementary Table 3) of the core eukaryote genes recovered for the majority of the genome assemblies. The gap-free reference genome (version 2.0) contained 23, 207 predicted protein-coding genes (Supplementary Data 2). It filled many gaps in the initial chromosome-level assembly (version 1.0, Supplementary Note 1, Supplementary Method 1 and Supplementary Method 2), resulting in ~24.5 Mb extra sequences and 3429 protein-coding genes (Fig. 1d).

The predicted protein-coding genes were then compared with protein sequences in six databases, including Nr, SwissProt[44], KEGG[45], COG, TrEMBL[44] and InterPro[46], with 97.1% of genes being assigned putative functional annotations (Supplementary Fig. 3 and Supplementary Table 4). The assembled draft genome of *A. gramineus* contained 50.1% repetitive sequences (Table 1). Long terminal repeat (LTR) retrotransposons were the most prevalent type of transposable elements (TE) among these repeats, representing nearly 38.4% of the genome, including 28.0% LTR/*Gypsy* and 7.2% LTR/*Copia* retroelements (Supplementary Tables 5 and 6).

### Incongruent phylogenomic placement of *A. gramineus*

To clarify the phylogenetic relationship of Acorales to other angiosperms, we selected 633 single-copy ortholog sets (SCG) from four eudicots, four monocots, three magnoliids, two ANA grade, and *Ceratophyllum demersum* (Supplementary Data 3). Both coalescent and concatenated methods showed an identical and highly supported topology: *A. gramineus* representing Acorales, was placed as a sister to the other monocots species (Fig. 2a and Supplementary Fig. 4). The topology also showed monocots as the sister to the clade including magnoliids, eudicots, and *Ceratophyllum*. Furthermore, we extracted 612 low-copy orthologous genes and generated a 223-species dataset (Supplementary Data 4). Phylogenetic trees based on coalescent and concatenated approaches showed similar topological structures, i.e. placing Acorales as the sister to the other monocot lineages (Fig. 2b, Supplementary Figs. 5 and 6). To clarify the phylogenetic relationships within the genus, we reconstructed a phylogenetic tree using transcriptome data of eight accessions (six are generated in this study, Supplementary Table 7). The results yielded two well-supported clades (see below). One clade included two varieties of *A. calamus*. Another clade included *A. gramineus*, *A. tatarinowii*, *A. macrospadiceus* and *Acorus* sp. HN.

A full-length chloroplast genome of 153,062 bp was also assembled for *A. gramineus*, which was in the range of the previously reported chloroplast genome length of 152–154 kb in *Acorus*[31–33]. Annotation identified 84 chloroplast genes (Supplementary Fig. 7). Phylogenetic trees based on a concatenated dataset of 80 chloroplast genes from 135 species across major clades of land plants showed a consistent result with the nuclear genes (Supplementary Data 5), supporting *A. gramineus* as the sister to the other monocots (Fig. 2c and Supplementary Fig. 8).

The initial attempt to obtain mitochondrial assembly using ~47.6 Gb of Nanopore long reads failed to achieve a complete genome, but the resultant multiple contigs with the longest one over 1 Mb, suggested a possible large mitochondrial genome size. A total of 38 mitochondrial genes were identified and used for phylogenetic reconstructions. The topologies from 38 single mitochondrial genes showed a strikingly different scenario from the nuclear and chloroplast genomes in placing *Acorus* in the angiosperm phylogenetic tree (Supplementary data 6). Only three mitochondrial genes (*atp4*, *nad1*,

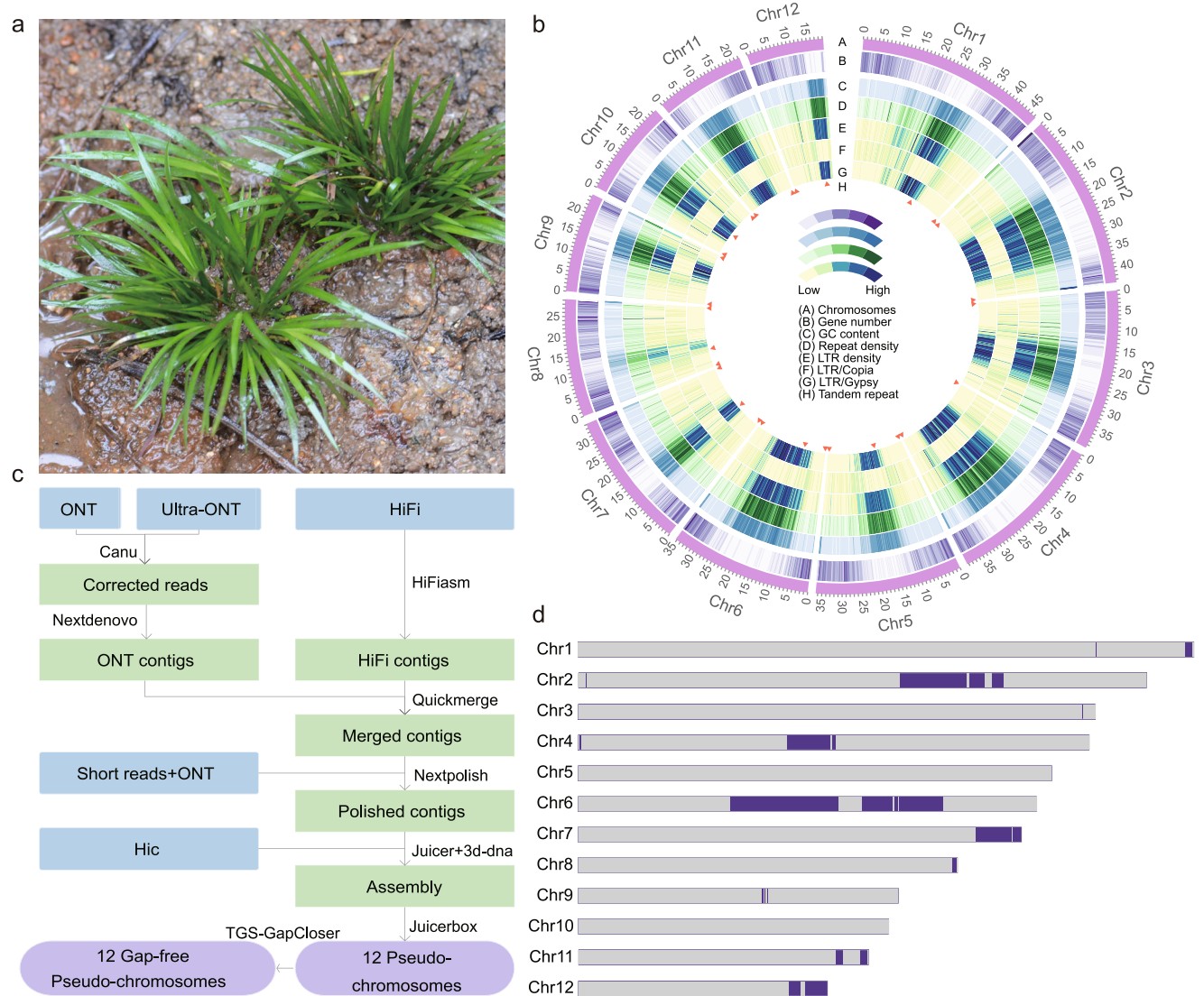

**Fig. 1 | Morphology and genomic architecture of *A. gramineus*. a** *A. gramineus* in its natural environment. **b** Circos plot of *A. gramineus* genome. The tracks from outside to inside display the (A) chromosomes (B) gene number (C) GC content (D) Repeat density (E) LTR density (F) LTR/*Copia* density (G) LTR/*Gypsy* density (H) High-density tandem repeat factors regions. **c** Gap-free assembly pipeline for generating the genome of *A. gramineus*. **d** Ideogram of the features of the gap-free *A. gramineus* genome assembly. The chromosome numbers are shown on the left of each bar. The dark blue segments on the chromosomes indicate the gaps in the preliminary version (version 1.0, Supplementary Note 1) fixed by the gap-free genome.

## Table 1 | Major indicators of the *A. gramineus* genome

| Assembly feature | Statistics |
| --- | --- |
| DNA amount 1C (pg) | 0.4 |
| Estimated genome size (by *k*-mer analysis) (Mb) | ~400.3 |
| Chromosome number (2n) | 24 |
| Assembled genome size (Mb) | 399.8 |
| Contig N50 (Mb) | 12.0 |
| Repeat region % of genome | 50.1 |
| Gene number | 23,207 |
| Pseudochromosomes after Hi-C | 12 |
| Scaffold N50 (Mb) | 36.5 |
| Longest scaffolds (Mb) | 47.5 |
| BUSCO (Complete BUSCOs, %) | 96.7 |

and *rps12*) supported *Acorus* as the sister to other monocots. In contrast, most other mitochondrial genes assigned *Acorus* at misplaces in other lineages of angiosperms, involving Alismatales (*atp1*, *ccmB*, *matR*, *nad4*, *rps3*, and *sdh4*), Poales (*cob*), magnoliids (*rps4*), eudicots (*rpl2*, *ccmFC*, and *cox2*), and Ceratophyllales (*ccmFN*) (Fig. 2d, Supplementary Figs. 9–46 and Supplementary Table 8). The phylogenetic misplacement was reflected in the single-gene alignments. A large number of mutation sites were identified in *Acorus* in contrast to little or no detectable sites in other angiosperms (Supplementary Figs. 47–49).

### A large mitochondrial genome with a rapid mutation rate

To obtain a high-quality mitochondrial genome, ~20.0 Gb ultra-long nanopore reads were added to the analyses. A nearly complete mitochondrial genome was assembled with a full length of 2221 kb (Fig. 3a), which is one of the largest mitochondrial genomes sequenced to date within monocots (Supplementary Data 7). It was considerably larger than the mitochondrial genome of most angiosperms, with the exceptions in *Amborella trichopoda* (3866 kb)[47], *Silene noctiflora*

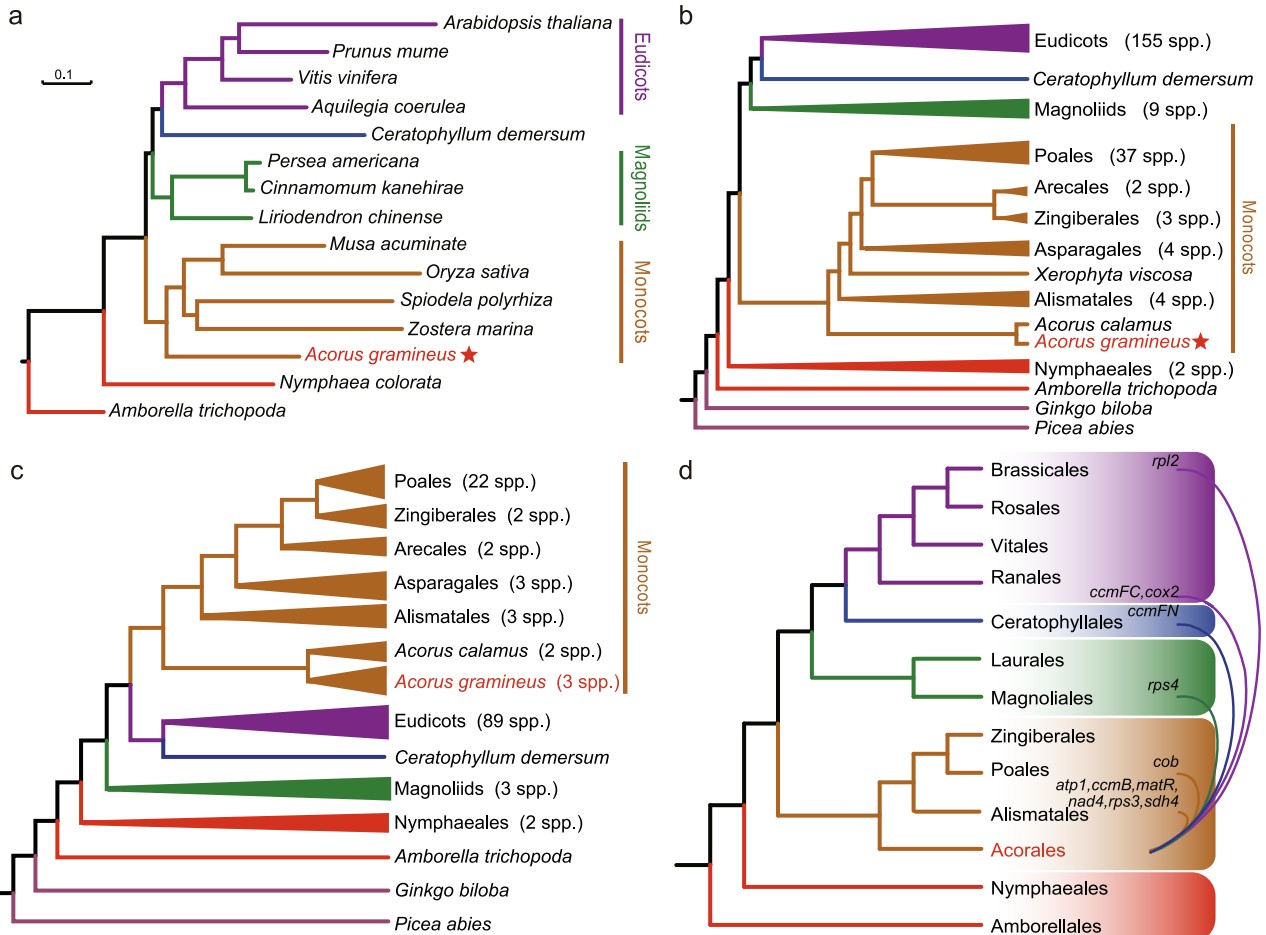

**Fig. 2 | Incongruent phylogenetic placement of *A. gramineus* based on nuclear, chloroplast, and mitochondrial genomic data. a** The phylogenetic tree constructed by IQTREE based on a concatenated dataset of 633 single-copy nuclear genes, with a bootstrap support of 100 for all nodes. **b** The simplified backbone of the phylogenetic tree constructed by IQTREE using 612 low-copy orthologous genes from 223 species using the concatenation method. **c** The simplified backbone of the phylogenetic tree based on the 80 chloroplast gene sequences of 135 plant species. **d** Summary of the results of mitochondrial gene trees, showing the misplacements of *Acorus* in individual gene analyses. Source data underlying Fig. 2 are provided as a Source Data file.

(6728 Kb), and *Silene conica* (11,319 Kb)[48]. The mitochondrial genome annotation of *A. gramineus* identified 37 genes, similar in gene number of the other monocots (Supplementary Data 7): *Oryza sativa* (37 genes, 438 kb genome), *Sorghum bicolor* (32 genes, 469 kb genome), *Phoenix dactylifera* (43 genes, 451 kb genome), *Spirodela polyrhiza* (35 genes, 228 kb genome) and *Zostera marina* (25 genes, 191 kb genome) (Fig. 3b). The mitochondrial gene content of *A. gramineus* did not show detectable increase than other typical monocots. The large content of the genomic expansion was attributed to intergenic regions (accounted for ~99%). Several intergenic regions were even larger than the size of whole mitochondrial genomes, such as *atp8-nad7* (291,698 bp), *matR-atp8* (247,434 bp), *nad4-atp4* (174,883 bp) and *nad7-atp1* (171,603 bp). The gene *rps19* was found missing in the assembled mitochondrial genome, but was present in chromosome eight (Supplementary Fig. 50), indicating a shift event from the mitochondrial to nuclear genome.

To assess the divergence level of mitochondrial protein-coding genes of *Acorus*, $d_S$ and $d_N$ values were estimated for 38 genes. In all mitochondrial genes, *Acorus* has a longer $d_S$ branch length compared to other sampled angiosperms, except *Silene latifolia*, a species with ultrafast substitution rates in mitochondrial genes (Fig. 3c). This divergence occurred at the ancestral node of *Acorus* before the intrageneric diversification, indicating a highly elevated mutation rate before the divergence of the species from the common ancestor of *Acorus*. This pattern is contrary to that of *Silene* with a short stem $d_S$

branch and high variation in branch length among species. As expected, the $d_N$ branch lengths were also highly elevated in *Acorus* relative to other angiosperms in most mitochondrial genes except two conserved genes *atp9* and *cox1* (Supplementary Figs. 51–60). The $d_N/d_S$ value of these two genes was depressed (<0.1), and most other mitochondrial genes fell in the range of 0.1–1, indicating a different interplay of mutational and selective forces among these mitochondrial genes (Supplementary Data 8).

**Ancient lineage-specific whole-genome duplication**

To identify whole-genome duplication (WGD) event in *A. gramineus*, a genome collinearity analysis of *A. gramineus* with *A. trichopoda* was performed. The synteny analysis of *A. gramineus* with *A. trichopoda* identified 271 syntenic blocks that covered 49.4% and 58.6% of the assembled genomes, respectively (Supplementary Table 9). In addition, we identified 1:2 syntenic depth ratios in the *A. trichopoda-A. gramineus* comparison (Supplementary Fig. 61). *A. trichopoda* is considered as the sister lineage to all other extant angiosperms, with no evidence of lineage-specific polyploidy[49]. The widespread synteny and well-maintained 1:2:2 syntenic blocks between *A. trichopoda-A. gramineus-A. tatarinowii* suggests that one WGD event occurred during the evolution of *A. gramineus* (Fig. 4c).

To more precisely infer the timing of the WGD in the *A. gramineus* genome, intragenomic and interspecies homolog $Ks$ (synonymous substitutions per synonymous site) distributions were estimated. The

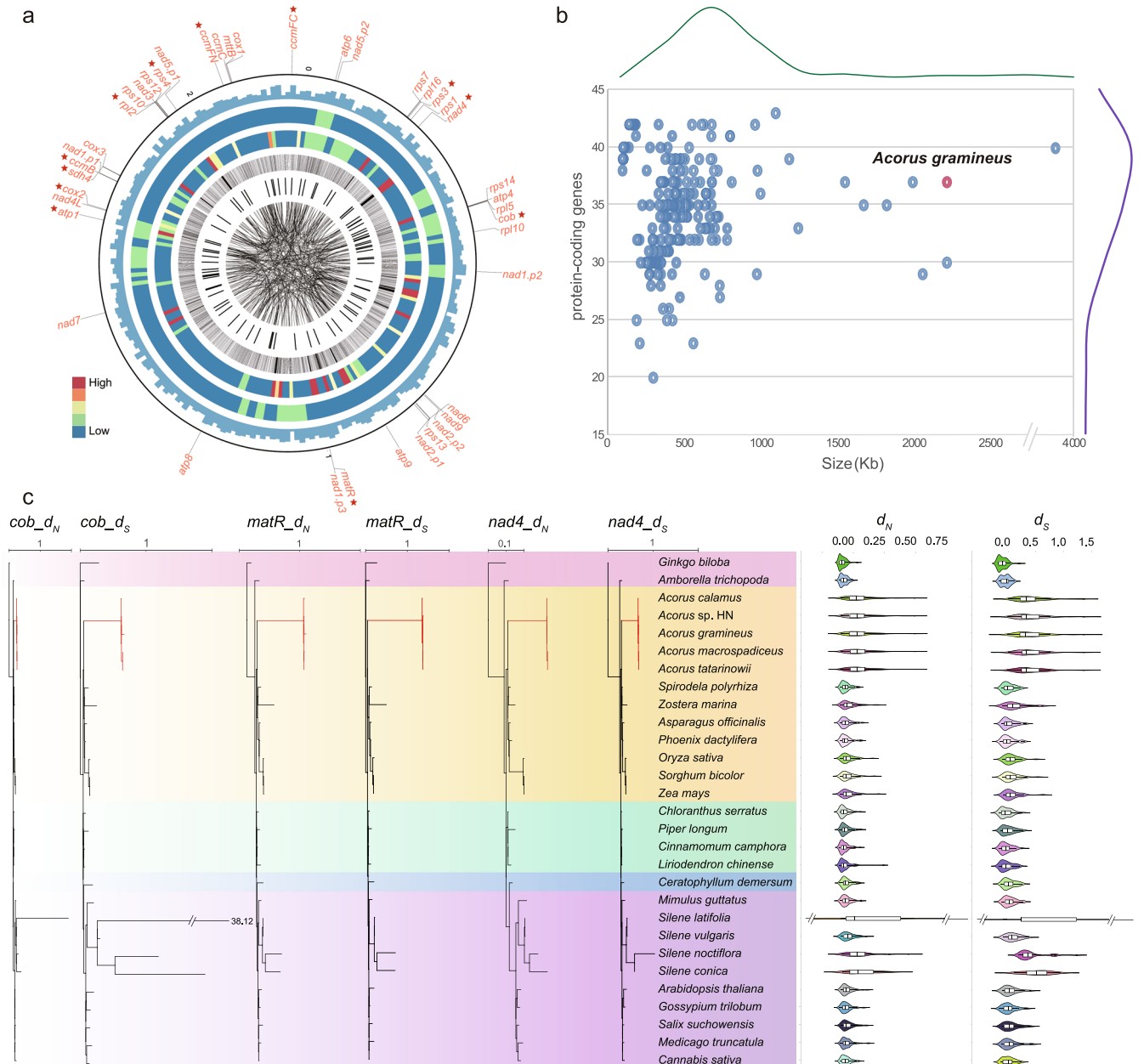

**Fig. 3 | The bizarre mitochondrial genome of *A. gramineus* with a rapid mutation rate. a** Circos map of important features of the assembled mitochondrial genome of *A. gramineus*. Elements are arranged in the following scheme (from outer to inner): The position of the 37 mitochondrial genes, the genes with an incongruent phylogenetic position are marked with stars; Distribution of GC content (non-overlapping, window size, 10 kb; 0.30 - 0.45); Heatmap of depth with long reads mapping (non-overlapping, window size, 10 kb; coverage, 5 - 35×); Heatmap of depth with short reads mapping (non-overlapping, window size, 10 kb; coverage, 170 - 250×); Link of pair-end short reads; Identification and position of tandem repeat (size with >100 bp); Identification of non-tandem repeat (size with >100 bp); **b** The plot of the number of protein-coding genes to genome size of 220 angiosperm mitochondrial genomes. The curve above represents the density distribution of mitochondrial genome size; The curve on the right represents the density distribution of gene numbers in mitochondrial genomes. **c** Phylograms of nonsynonymous nucleotide substitution ($d_N$) and synonymous nucleotide substitution per site ($d_S$), exhibiting the sequence divergence in three representative mitochondrial genes. The violin plot is the distribution of nonsynonymous nucleotide substitution ($d_N$) and synonymous nucleotide substitution ($d_S$) from mitochondrial genes of each species. In the violin plot, the black vertical line in the box shows the median value of the data, and the right and left edges of the white box represent the upper and lower quartiles of the dataset. The scale above each phylogram represents the corresponding branch length ratio. Source data underlying Fig. 3a are provided as a Source Data file.

intragenomic *Ks* distribution of *A. gramineus* genome showed a major peak at ~0.4 (Fig. 4b and Supplementary Data 9). Compared to other monocot species, we identified different peaks in *Ananas comosus*, *O. sativa*, and *S. bicolor* (Fig. 4b), suggesting that *A. gramineus* did not experience the same tau (τ) WGD event as that of other monocots (Fig. 4a)[50]. This result is consistent with the Harkess et al. 2017 and the 1KP transcriptome phylogeny paper[29,30]. After correcting the evolutionary rate, the mean *Ks* values of syntenic blocks were used to compute the time of the WGD event, resulting in an estimated time of the WGD event at approximately ~40.6 to ~58.2 million years ago (Supplementary Data 10). The transcriptome of the other five *Acorus* samples was also used to calculate the *Ks* value, the five intra-transcriptome *Ks* distributions of *Acorus* species showed a major peak at ~0.4, similar to the intragenomic *Ks* distribution of *A. gramineus*

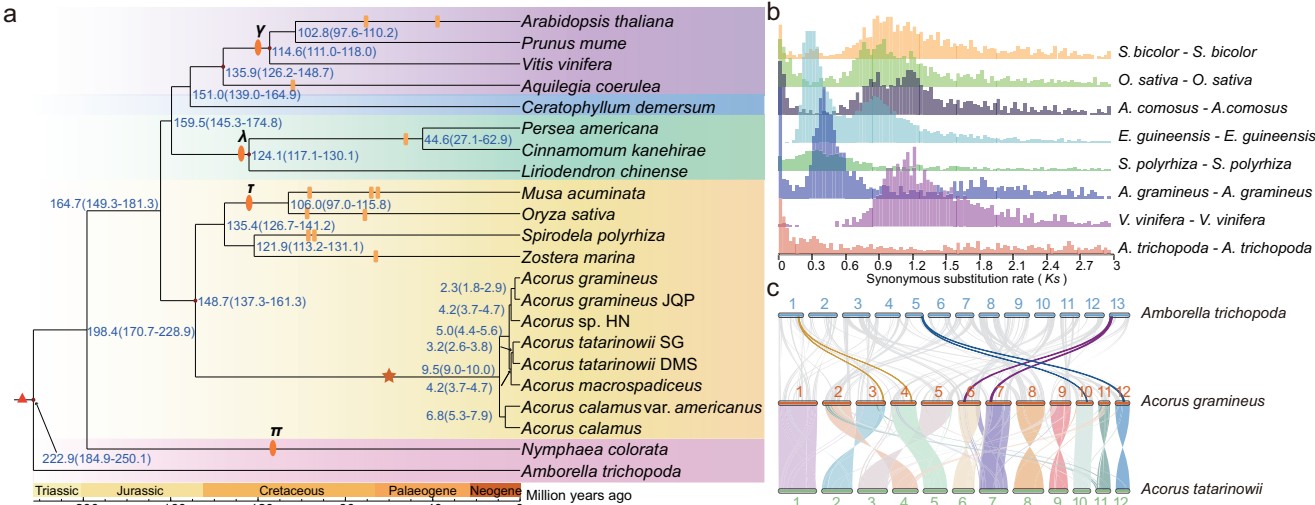

**Fig. 4 | Whole-genome duplication of *A. gramineus*. a** Dating of WGD events on the phylogenetic tree. Brown star indicates the lineage-specific WGD event in *A. gramineus*. Yellow rectangle represents known WGD events identified previously. **b** The intragenomic *Ks* distribution. *S. bicolor*: Sorghum bicolor, *O. sativa*: Oryza sativa, *A. comosus*: Ananas comosus, *E. guineensis*: Elaeis guineensis, *S. polyrhiza*: Spirodela polyrhiza, *A. gramineus*: Acorus gramineus, *V. vinifera*: Vitis vinifera, *A. trichopoda*: Amborella trichopoda. **c** Synteny between genomic regions in

*A. trichopoda*, *A. gramineus* and *A. tatarinowii*. This pattern shows that an ancestral syntenic region in the ANA grade *A. trichopoda* can be tracked up to two regions in *A. gramineus* owing to the one genome duplication event. Gray lines in the background highlight major syntenic blocks spanning the genomes. Colored lines represent the example of syntenic genes found in two species that correspond to one copy in *A. trichopoda*, and two in *A. gramineus*. Source data underlying Fig. 4a are provided as a Source Data file.

(Supplementary Fig. 62). These observations suggest that the detected ancient WGD event is lineage-specific to *Acorus*.

Following the WGD event, *A. gramineus* retained a total of 7902 paralogous genes. Gene duplicates from the WGD events were maintained in these gene families, which were enriched for various photosynthesis-related GO terms (Supplementary Fig. 63 and Supplementary Table 10). Simultaneously, KEGG enrichment analysis of WGD retained genes revealed that they are involved in photosynthesis, energy metabolism, and signaling pathways (Supplementary Fig. 64 and Supplementary Table 11).

## Gene contractions and expansions associated with structural and ecological adaption

We compared the assembled nuclear genome of *A. gramineus* with 13 other sequenced angiosperm genomes representing ten monocot species (*Z. marina, S. polyrhiza, Phalaenopsis equestris, Asparagus officinalis, A. comosus, O. sativa, S. bicolor, Musa acuminata, P. dactylifera, Elaeis guineensis*), two eudicot species (*Arabidopsis thaliana, Vitis vinifera*), and one ANA grade species *A. trichopoda* (Fig. 5a and Supplementary Table 12). Based on the analysis of gene-family clustering, we identified 28,676 gene families, of which 4969 were shared by all 14 species, representing ancestral gene families, and 653 of these shared families were single-copy gene families (Fig. 5b).

In comparison to gene families in their most recent common ancestor (MRCA) of the 14 plant species, *A. gramineus* had 84 expanded and 5 contracted gene families (Fig. 5a). Gene Ontology (GO) studies based on the 84 expanded gene families showed enrichment of genes encoding "carbohydrate binding", "pattern binding" and "organic cyclic compound binding" (Supplementary Fig. 65 and Supplementary Data 11). According to KEGG functional enrichment analysis, the expanded gene families were mostly attributed to "arginine and proline metabolism", "phenylalanine metabolism" and "tryptophan metabolism" pathways (Supplementary Fig. 66 and Supplementary Data 12).

Annotation identified 23,207 genes in *A. gramineus*, which was 45% less than that of *O. sativa* (42,069 genes), 35% less than *M. acuminata* (35,862 genes), and 32% less than *S. bicolor* (34,124 genes).

Gene expansion and constriction analyses were performed to explore gene-family evolution within monocots. Transcription factors (TFs) are essential for plant growth and development[51]. We identified 1,461 TFs in *A. gramineus*, which was much fewer than that of most monocots, such as 1945 in *O. sativa*, and 3081 in *M. acuminata* (Supplementary Data 13). The fewer numbers were reflected in different gene families, including *WRKY, BHLB, MYB, NAC, MADS-box, C2H2* and *bZIP* (Fig. 5d and Supplementary Fig. 67).

*WRKY* transcription factors play a vital functional role in stress resistance and secondary metabolism in plants[52, 53], which are classified into five classes (I, IIa + b, IIc, IIe + d, III). We identified a total of 60 *WRKY* in *A. gramineus* (Fig. 5e and Supplementary Table 13), fewer than most monocot species with the exceptions in *Z. marina* (44 genes), *S. polyrhiza* (49 genes) and *A. comosus* (56 genes). Few gene numbers in *A. gramineus* were also reflected in Resistance (R) genes. Vascular plant defense is based on proteins containing the nucleotide-binding site domain and a leucine-rich repeat domain (NBS-LRR) with an additional coil-coil (CC) domain in some cases[54–56]. We identified 71 nucleotide-binding site-leucine-rich repeats (NBS-LRR) genes in *A. gramineus*, which was fewer than most other species. Meanwhile, Toll/interleukin-1 receptor TIR-NBS-LRR (TNL) were absent in monocots[57], including *A. gramineus* (Fig. 5c, Supplementary Tables 14 and 15).

Few genes were observed in several gene families which control plant development and architecture. Eudicot stems are arranged in a ring with a vascular cambium, whereas vascular bundles are scattered throughout in monocot stems[58,59]. *HD-ZIP-III* is a type of transcription factor that functions in the formation of the vascular system[60,61]. There are five copies of the *HD-ZIP III* TFs in *A. thaliana* and *O. sativa*, including two genes of *PHB/PHV* class. *A. gramineus* also has five copies of the *HD-ZIP III* TFs, but has only one gene of *PHB/PHV* class (Supplementary Fig. 68). Likewise, *CLAVATA3/Embryo* Surrounding Region-Related (*CLE*) peptides have been found to play a role in seed development, vascular bundle formation, lateral root growth, and the balance between stem cell division and differentiation in apical and apical root meristem tissues[62]. Interestingly, eight *CLE* genes were identified in *A. gramineus* in contrast to 31 genes in *A. thaliana* and 34 genes in *O. sativa* (Supplementary Fig. 69). Expansin is a type of protein that has an extensive regulatory function during plant growth such

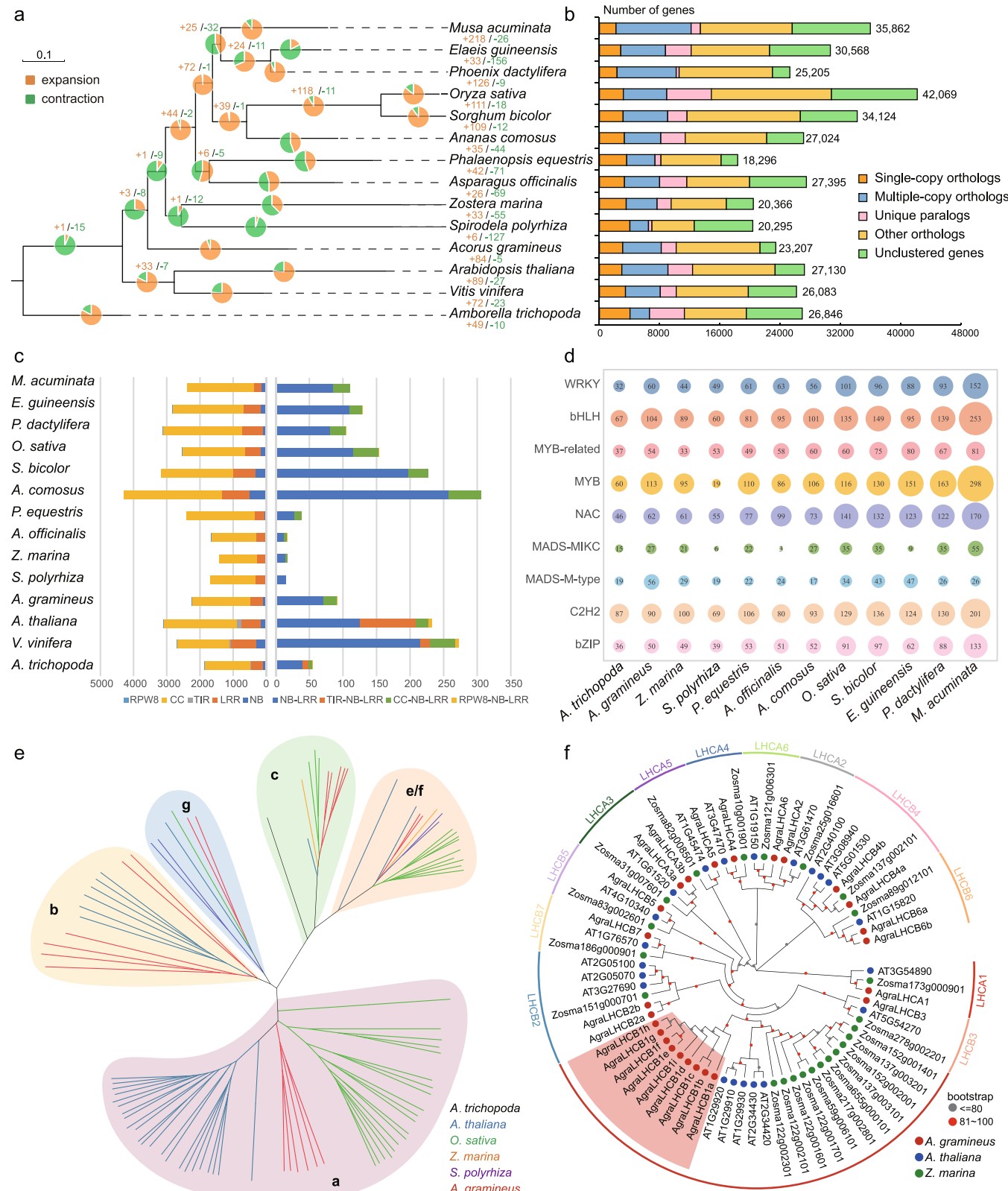

**Fig. 5 | Comparative genomic and gene-family analyses. a** The number of gene-family contraction and expansion events is respectively presented by green and orange numbers or pie charts. **b** The distribution of single-copy, multiple-copy, unique, and other orthologs in the 14 selected plant species. **c** Histogram shows the R-gene number in the 14 plant species, which were used in the contraction and expansion analysis. The different color stands for different domain interaction types and are explained below the histogram. **d** The bubble matrix shows the characteristics of nine TFs in 12 monocots species. The numbers in the bubbles represent the number of these TFs in each species, the size of the bubbles varies with the number. **e** The phylogenetic tree was constructed using putative or characterized *TPS* genes from *A. thaliana, O. sativa, Z. marina, S. polyrhiza* and *A. gramineus*. **f** The tree of light-harvesting complex (*LHC*) superfamily in *A. gramineus* with comparison to *A. thaliana* and *Z. marina*. red circle: *A. gramineus*; blue circle: *A. thaliana*; green circle: *Z. marina*. The red highlighted area is the clade of *A. gramineus*'s *LHCB1* gene family. Source data underlying **a** and **b** and are provided as a Source Data file.

as: cell wall development, cell extension, fruit softening, abscission, the development of root hairs and meristem function[63,64]. Compared with other monocotyledonous plants (56 for rice and 88 for maize), *A. gramineus* has a significantly lower number of expansins. We identified 25 expansins in *A. gramineus*, which were further classified as α-expansins (17), β-expansins (4) and expansin-like (4) genes (Supplementary Fig. 70).

Although many gene families and metabolic pathways were contracted in *A. gramineus*, others still showed expansions, such as some energy metabolism, photosynthesis and oxidative phosphorylation pathways. We found light-harvesting complex (*LHC*) superfamily was expanded in *A. gramineus*, especially in the *LHCB1* subfamily. We also identified 25 *LHC* genes containing nine *LHCB1* genes (Fig. 5f and Supplementary Table 16), compared to five in *A. thaliana*[65]. Interestingly, gene expansion was also detected in terpene synthases b (*TPS-b*) subfamily in *A. gramineus*. The *TPS-b* subfamily gene number is relatively lower in monocots[66], and even absent in several lineages, such as *O. sativa*, *S. polyrhiza* and *Z. marina* (Fig. 5e), whereas, there were six *TPS-b* genes identified in *A. gramineus* (Supplementary Table 17).

## Discussion

In recent years genomic data have been successfully employed to reveal phylogeny and evolution of plants[67–70]. Being the sister lineage to all other extant monocot plants, *Acorus* is an important lineage for studying the genetic architecture and genome evolution of early monocots. In this study, we employed a combination of long-read Nanopore reads, ultra-long Nanopore reads, PacBio HiFi reads and BGI-DIPSEQ short reads, coupled with Hi-C technology to generate a high-quality gap-free genome of *A. gramineus*. This assembly contributes to a better understanding of genome evolution in monocots, and also facilitates the comparative analyses of ecological adaptation in wetland plant species.

Both nuclear and chloroplast phylogenetic trees in this study placed *A. gramineus* as the sister to the remaining monocots, which corroborated the placement of *Acorus* in APG IV system and previous phylogenetic studies[3,28,30–33]. Controversial placement of monocots relative to magnoliids and eudicots between nuclear and chloroplast trees was detected, and likely due to the underlying causes of ancient hybridization and incomplete lineage sorting that have been investigated recently in the *Chloranthus* genomes[67,71] and phylotranscriptomic data of 92 streptophytes[72]. Interestingly, through the analysis of mitochondrial genes, we found *Acorus* was assigned at a misplacement in most single-gene trees, even being placed into magnoliids (*rps4*), eudicots (*rpl2*, *ccmFC*, and *cox2*), and Ceratophyllales (Fig. 2d). Sequence divergence comparison suggested that both $d_N$ and $d_S$ branch lengths were highly elevated in *Acorus* relative to other angiosperms in most genes (Fig. 3c). The high sequence divergence specifically reflected in single-gene alignments. There are a large number of mutation sites identified in *Acorus* in contrast to little or no detectable sites in other angiosperms. Out of these mutation sites, a small proportion of indels coincidently shared among distantly related lineages and contributed to 'phylogenetic synapomorphies' that likely resulting the misplacements of *Acorus* in mitochondrial genes. The highly divergent mitochondrial genes with elevated mutation rates likely have evolved under relaxed selection (if not to some extent positive selection) at the ancestor of genus *Acorus*. The rapid mutational rates also explained the misplacements of *Acorus* in single-gene tree in previous studies using mitochondrial data[15], highlighting the need to exclude mitochondrial genes in future phylogenetic studies involving *Acorus*.

Interspecific relationships among *Acorus* are controversial[28,73,74]. There are two currently accepted species in *Acorus*, however, a recent study[28] suggested that four species should be recognized in the *Acorus*. Our phylogenetic tree using transcriptome data of eight accessions inferred two well-supported clades in genus *Acorus*. One clade included two varieties of *A. calamus* (*A. calamus* var. *calamus* and *A. calamus* var. *americanus*). Another clade included *A. gramineus*, *A. tatarinowii* (synonym of *A. calamus* var. *angustatus*, https://www.catalogueoflife.org/), *A. macrospadiceus* (synonym of *A. gramineus*) and an unidentified species (*Acorus* sp. HN.) that is morphologically distinct from the rest of the species. The interspecific relationships retrieved suggested that *A. calamus* as currently circumscribed is polyphyletic, suggesting *A. tatarinowii* that is currently accepted as the synonym of *A. calamus* var. *angustatus* should be considered as a distinct species. In subsequent research, it will be necessary to collect samples from numerous individuals representing each species to elucidate the intrageneric relationship of *Acorus* and make necessary taxonomic revisions.

Annotation identified 23,207 genes in *A. gramineus*. This species together with representatives in Alismatales (*S. polyrhiza* and *Z. marina*) have relatively fewer gene numbers than other monocots. The few gene number pattern is specifically reflected in gene families and transcription factors. Gene number expansion in most monocots might be the result of the vast gene-retain after τ WGD event that is absent in Acorales and Alismatales. *A. gramineus* holds fewer *WRKY* and *R* genes that play important roles in stress resistance and innate immunity, indicating adaptation to aquatic/wetland habitats where plants suffer from far fewer pests and diseases than land plants[65]. NBS genes are mainly categorized into three classes, depending on the domain composition of the *R* genes. TNL genes were not recognized in monocots including *A. gramineus*, consistent with the previous report[75]. TNL subclass genes exist in *Amborella*, *Nymphaea*, magnoliids and eudicots, suggesting TNLs might present in the ancestor of angiosperms with subsequent loss in the ancestor of monocots.

Although several gene families and transcription factors, such as *CLE*, *WRKY*, and *R* genes, were contracted, gene expansion was detected in *LHC* and *TPS*. The expansion of the *LHC* superfamily, especially *LHCB1*, may be due to the fact that *A. gramineus* naturally lives in wetlands, mostly under trees. Wetlands often have low-light levels, so *A. gramineus* has developed mechanisms to capture more light. This is in accordance with the findings in seagrass *Z. marina*[65] and shade-adapted species *Begonia*[68]. These expansions of *LHCB1* were independent events in several lineages living in low-light habitats suggesting parallel evolution. The expansion of *TPS* gene families, particularly *TPS-a* and *TPS-b* genes, might be responsible for the rich volatile content[76,77].

## Methods

### Sample preparation, library construction and sequencing

The tissue samples of *A. gramineus* (Voucher number: JQP20200401GX, SCBG) were collected from Zhangzhou, Fujian, China (Supplementary Table 7). For genome sequencing, genomic DNA was extracted from *A. gramineus* leaves using the cetyltrimethylammonium bromide (CTAB) method[78]. The library was sequenced on the BGI-DIPSEQ platform[79], generating ~80 Gb of 100 bp paired-end reads with an insert size of ~250 bp. The generated raw reads were filtered according to sequencing quality with Trimmomatic (v0.40)[80]. For subsequent analysis, such as genome size calculation and ONT assembly polish, only high-quality reads were used.

For the ONT library[81], leaf tissues of *A. gramineus* were ground in liquid nitrogen and extraction was performed. The library was generated using LSK108 kit (SQK-LSK108, Oxford) and sequenced on the Nanopore GridION X5 sequencer[82] using 5 flow cells. The base calling was performed using Guppy (v4.0.11) in MinKNOW package. There were ~2.7 million nanopore reads, ~50 Gb raw data in total with an N50 length of ~22.6 kb (Supplementary Data 1).

The construction of Hi-C libraries was performed by following the method developed by BGI QingDao Institute[83]. DNA from young leaves

of *A. gramineus* were digested with MboI using the standard Hi-C library preparation protocol[83]. The Hi-C libraries were sequenced on BGI-DIPSEQ platform, generating ~70 Gb of data with 100 bp paired-end reads (Supplementary Data 1).

The young leaf, stem, and root tissues of *A. gramineus* and other five *Acorus* spp. samples were collected for transcriptome sequencing (Supplementary Table 7). Total RNA was isolated using the TIANGEN Kit with DNase I and then processed using the NEBNextUltra™ RNA Library Prep Kit to create a pair-end library with a 250 bp insert size. Libraries were barcoded and pooled together as an input to the BGI-DIPSEQ platform for sequencing. Following the removal of low-quality data, six Gb of 100 bp paired-end data for each tissue were used for further RNA-seq analysis.

Ultra-long DNA was extracted by the SDS method without a purification step to sustain the length of DNA. After the sample was qualified, size-select of long DNA fragments were performed using the PippinHT system (Sage Science, USA). DNA library was constructed and performed on a Nanopore PromethION sequencer instrument (Oxford Nanopore Technologies, UK) at the Genome Center of Grandomics (Wuhan, China). For each ultra-long Nanopore library, approximately 10 g of gDNA was size-selected (>50 kb) with SageHLS HMW library system (Sage Science, USA), and DNA libraries were constructed and sequenced on the Promethion (Oxford Nanopore Technologies, UK) at the Genome Center of Grandomics (Wuhan, China). For PacBio sequencing, SMRTbell libraries were constructed and sequenced on a PacBio Sequel II system. The consensus reads (HiFi reads) were generated using CCS software (https://github.com/pacificbiosciences/unanimity) with the default parameter. All the software and version details are listed in Supplementary Data 14.

### Genome size estimation
According to the earlier report, *A. gramineus* has a small genome size (~391.2 Mb) based on the flow cytometry[43], and possesses 12 chromosomes (2n = 2x = 24). The genome size of *A. gramineus* was estimated using *k*-mer frequencies, according to the Lander-Waterman theory[84].

To minimize the sequencing error rate, a strict quality control was performed using SOAPfilter (v2.2) with default parameter setting (-q 33 -y -p -M 2 -f -1 -Q 7 -W 5 -B 25): filtering out reads with >5% of bases as missing "N", reads with >25% of poor quality bases (ASCII Q-score 33 < 7), and PCR duplicates[80]. Based on the results of 17-mer frequency distribution analysis with GenomeScope (v2.0; *k* = 17)[85], we estimated the genome size of *A. gramineus* to be 400.3 Mb, which was close to previous estimates by flow cytometry[43].

### Genome assembly and assessment of the assembly quality
For the de novo genome assembly, first, the ultra-long ONT data was corrected using Canu (v2.0)[86] with the following parameters: genomeSize = 400m, minReadLength = 1000, minOverlapLength = 500, corOutCoverage = 120, corMinCoverage = 2. The corrected data were used for assembling in NextDenovo[87] (v2.5.0, https://github.com/Nextomics/NextDenovo) with the following parameters: input type = corrected, genome size = 400 Mb, read cutoff = 1k, seed depth = 45. Second, PacBio HiFi reads were assembled using Hifiasm (0.18.0-r465)[76] with default parameters. These two datasets generated the primary gap-free contig genomes.

Then, initial assembly was polished using both ONT long reads and short reads (task = best model) to generate a high-continuity and high-accuracy genome assembly by NextPolish (v1.5.0, https://github.com/Nextomics/NextPolish)[77]. After that, the Hi-C reads were mapped to the polished genome using Juicer (v1.7.6)[88]. The alignment information was used to produce the Hi-C contact map by 3D-DNA (v180922)[89]. The Hi-C contact map was then visualized by Juicebox (v1.11.08)[90], after some manual adjustments, including correcting

inversion errors and re-joining contigs. Finally, 12 super contigs or scaffolds representing 12 pseudochromosomes were obtained.

The genome assembled from HiFi data was first used to fill the gaps in the pseudochromosomes genome with Quickmerge (https://github.com/mahulchak/quickmerge)[91]. Then, the remaining gaps in the scaffolds were filled by mapping all the corrected ONT and HiFi reads against the assembly with TGS-Gapcloser (https://github.com/BGI-Qingdao/TGS-GapCloser)[92]. The gap boundary regions in the alignment were manually identified and the reliable reads were selected for the final gap-closure.

Furthermore, we identified the centromere and telomeric regions using Tandem Repeats Finder (v4.10.0)[93]. Tandem repeat monomers over 80% similarity were assigned into one sequence clusters by cd-hit (v4.8.1)[94]. Finally, we identified the most abundant tandem repeat clusters for candidate centromeric and telomeric tandem repeats, which occupied the majority in each chromosome.

To assess the completeness of the genome assembly, the following strategies were employed: The software Benchmarking Universal Single-Copy Orthologs (BUSCO) (v5.3.2)[95] with the embryophyta_odb10 database was used to evaluate the genome assembly; The ONT, PacBio long reads, genomic short reads and RNA-Seq reads were mapped to the genome assembly with HISAT2 (v2.2.1)[96], minimap2 (v2.21-r1071)[97] and BWA-MEM2 (v2.2.1)[98], respectively (Supplementary Data 1).

### Annotation of repetitive elements
Repeat sequences in the *A. gramineus* genome were identified as follows: Tandem repeats were searched across the genome using the software Tandem Repeats Finder (v4.10.0)[93]; transposable elements (TEs) were predicted by employing a combination of similarity-based comparisons in RepeatMasker (v4.0.5) and RepeatProteinMask[99], and de novo approaches using LTR_retriever[100], LTR_FINDER (v1.0.7)[101], RepeatModeler2[102] and MITE-hunter (v2.2)[103]. The MITE, LTR and TRIM (Terminal repeat retrotransposon in miniature) repetitive sequence libraries were integrated to make a complete and non-redundant custom library. The repeat library was taken as the input for RepeatMasker (v4.0.5)[99] to identify and classify transposable elements.

Regions of LTR-retrotransposon sequences coding for reverse transcriptase (RT) and integrase (INT) protein domains were identified using DANTE-Protein Domain Finder (https://github.com/kavonrtep/dante/), a tool available at the Repeat Explorer server[104]. The hits were screened to retain at least 80% (-thl) of the reference sequence, with a minimum identity of 35% (-thi) and minimum similarity of 45% (-ths), allowing for a maximum of three interruptions (frameshifts or stop codons).

### Protein-coding gene prediction and functional annotation
The protein-coding gene set of *A. gramineus* was inferred using de novo, homologous and evidence-based gene prediction (RNA-seq data) approaches. De novo gene prediction was performed on a repeat-masked genome using three programs, including Augustus (v3.0.3)[105], GlimmerHMM (v3.0.1)[106] and SNAP (v11/29/2013)[107]. Training models were generated from a subset of the transcriptomic dataset representing 800 distinct genes. Homologous gene prediction was achieved by comparing protein sequences of *A. trichopoda*, *A. thaliana*, *O. sativa*, *Z. marina*, *A. calamus* var. *americanus* and uniprot database (release 2021_04). For each reference, the following steps were executed: (1) Predicting putative homologous genes from alignments with protein sequences covering the complete gene sets (the longest transcripts were chosen to represent each gene) with TBLASTN (v2.2.18) (e-value cutoff: 1e-5)[108]; (2) The corresponding regions were retrieved, together with sequences 2 kb downstream and upstream of the aligned regions. (3) The alignments were

additionally handled using GeneWise (v2.2.0)[109] to obtain precise exon and intron information. Evidence-based gene prediction was conducted by aligning all RNA-seq data generated herein against the assembled genome using Hisat2 (v2.0.4)[96]. cDNAs were identified by a genome-guided approach using StringTie (v1.2.2)[110] and then mapped back to the genome using PASA (v2.3.3)[111]. The resulting cDNA sequence assembly by Trinity (v2.6.6)[112] were aligned to the *A. gramineus* genome sequences using BLAT (v34x12)[113]. Following the prediction of genes, a non-redundant gene set representing homology genes, de novo genes, RNA-seq supported genes, was generated using MAKER pipeline (v2)[114] and integrated into a final set of 23,207 protein-coding genes for annotation.

Predicted genes were subjected to functional annotation by performing a BLASTP homolog search against public protein databases, including KEGG (v59.3)[115], SwissProt (release-2020_05), TrEMBL (release-2020_05)[44] and NCBI non-redundant protein NR database (v20201015), and InterProScan (v5.28-67.0)[116] was also used to provide functional annotation.

### Transcriptome assembly and coding sequences prediction

Prior to assembly, we retrieved the high-quality reads by removing adapter sequences and filtered low-quality reads using Trimmomatic (v0.40)[80]. The resulting high-quality reads were then de novo assembled with the Trinity (v2.6.6) program[112]. Protein sequences and coding sequences of transcripts were predicted using TransDecoder (v5.3.0) (https://github.com/TransDecoder/TransDecoder).

### Phylogenetic analyses of nuclear genes

Single-copy genes from 15 seed plants (Supplementary Data 3) were identified using OrthoMCL[117]. With the 633 single-copy genes, we inferred the phylogenetic placements of *A. gramineus*. For each single-copy gene orthogroup, we first performed multiple amino acid sequence alignments by MAFFT (v.7.471)[118], and then DNA sequences were aligned according to the corresponding amino acid alignments using PAL2NAL (v14.1)[119], followed by gap position removal using trimAl (v1.4.1)[120]. Then each gene tree was constructed by IQTREE (v2.0.5)[121] that automatically selected the best-fit substitution model using ModelFinder[122]. After this, all gene trees were then utilized by ASTRAL (v.5.6.1) to infer species trees with quartet scores and posterior probabilities (coalescent method), and the concatenated supergenes alignments were used for IQTREE (v2.0.5)[121] (concatenation method), while the best model selected by IQTREE (v2.0.5)[121] was GTR + F + R5.

In addition, we built a phylogenetic tree from an extended species sampling from 223 species in total, including 221 angiosperms species, and two gymnosperms as outgroups (Supplementary Data 4). The amino acid sequences from all species were aligned using BLASTP with an e-value of 1e-5, and then grouped using OrthoFinder (v2.3.7)[123]. Here, each gene was required to include sequences from more than 80% species for low-copy genes, which resulted in 612 genes. The 612 low-copy genes were used to generate phylogenetic trees via the concatenation and coalescent methods mentioned above. For the concatenation method, the best model selected by IQTREE[121] (v2.0.5) was GTR + F + R10.

For the interspecific phylogenetic tree among *Acorus*, we added six transcriptomes of *Acorus* samples sequenced in this study and *A. calamus* var. *americanus* transcriptome data published in 1KP[30] (Supplementary Table 7) to 15 seed plants dataset. The amino acid sequences from all species were aligned using BLASTP with an e-value of 1e-5, and then grouped using OrthoFinder (v2.3.7). Here, each orthologous was required to include sequences from more than 18 species for low-copy orthologous genes, which resulted in 1,632 orthologous. The phylogenetic tree was constructed using concatenation methods by IQTREE[121] (v2.0.5) with the best model GTR + F + R4.

### Assembly of chloroplast and mitochondrial genome

The chloroplast genome (cp) of *A. gramineus* was assembled using the whole-genome short sequence raw data in NOVOPlasty (v4.3.1)[124], which was further annotated using the software CpGAVAS2[125].

For mitochondrial genome, ultra-long Nanopore reads were de novo assembled using Unicycler (v0.4.9) with default parameters. After manual confirmation, the final mitochondrial genome assembly resulted in 2.2 Mb size, 37 coding genes and ~38.6% GC content. Moreover, short reads were mapped to the assembly, showing that there was high coverage and a uniform depth in mitochondrial genomes.

### Phylogenetic analyses of chloroplast and mitochondrial genes

For the phylogenetic analyses, the alignments included cp and mt genomes of *A. gramineus* and other available angiosperms from the NCBI database, which covered all major lineages of angiosperms, including 135 cp (80 genes, four rRNAs and 76 protein-coding genes) (Supplementary Data 5) and 112 mt (38 protein-coding genes) genomes (Supplementary Data 6). For mitochondrial genes, we excluded all 1537 RNA editing sites identified in the mitochondrial genomes of the angiosperm ordinal representatives in our individual gene alignment to alleviate the negative impact of RNA editing in phylogenomics analyses[126]. All the protein-coding genes were aligned and trimmed following the same pipeline used for nuclear trees. After removing stop codons, and pseudogenes, all 38 individual alignments were used for phylogenetic analyses in IQTREE[121], with 1000 ultrafast bootstrap. Two gymnosperms, *Ginkgo biloba* and *Cycas taitungensis* were used as outgroups in the cp and mt phylogenetic analyses. The levels of mitochondrial genes synonymous ($d_S$) and nonsynonymous ($d_N$) divergence were estimated using PAML package (CODEML program, model: GY-HKY)[127].

A total 80 chloroplast genes were aligned and trimmed following the same pipeline used in nuclear and mitochondrial gene analyses. All genes were concatenated to infer a species tree using IQTREE[121] with 1000 ultrafast bootstrap.

### Estimation of divergence time

The divergence time of each tree node was inferred using MCMCtree (http://abacus.gene.ucl.ac.uk/software/paml.html) of the PAML package (version 4.9; options: correlated molecular clock, JCmodel and rest being the default)[127]. The final species tree and the concatenated translated nucleotide alignments of 633 low-copy orthologues were used as input of MCMCtree. The phylogeny was calibrated using various fossil records selected from previous publications (Supplementary Data 9) and TimeTree website (http://www.timetree.org).

### Analysis of genome synteny and whole-genome duplication

Synteny searches were performed to identify syntenic blocks between *A. gramineus* and *A. trichopoda* by MCSanX[128], using the modified parameters (e-value 10−20, maximum gaps 15, minimum size of collinear block 8).

To estimate the time of whole-genome duplication events, the low-copy families with the pairwise sequences of paralogous (within the species genome, $1 < N < 5$) and orthologous relationships (between *A. gramineus* and other species, 1:1) which based on gene families with OrthoMCL (v1.4)[129] were filtered. MUSCLE (v3.8.31)[129] was used to align each gene family, and *Ks* estimates for all pairwise comparisons within a gene family were generated by maximum likelihood estimation using the PAML package (CODEML program)[127]. After correcting the redundancy in *Ks* values (a gene family of n members produces $n*(n-1)/2$ pairwise *Ks* estimates for n − 1 retained duplication events), a phylogenetic tree was constructed for each subfamily using PhyML (v3.0)[130] under default settings. All *Ks* estimates between the two child clades were added to the *Ks* distribution with a weight of 1/m (where m is the number of *Ks* estimates for a duplication event), so the weights of all *Ks*

estimates for a single duplication event summed to one for each duplication node in the resulting phylogenetic tree.

The whole-genome duplication events time of *A. gramineus* was calculated by combining the *Ks* value with synonymous substitutions at each site per year (*r*) by using the formula:

$$\text{Divergence date}\,(T) = Ks/(2*r) \tag{1}$$

### Gene-family analysis
Gene families or orthologus groups of *A. gramineus* and 13 other land plants (Supplementary Table 12). were identified using OrthoMCL[117]. Gene-family expansion and contraction were inferred using CAFE' (v 4.1), with an input tree as the species tree constructed by IQTREE based on a concatenated dataset of single-copy orthologues[131]. For *A. gramineus* expansion gene families, we conducted GO and KEGG enrichment analyses via an enrichment pipeline (https://sourceforge.net/projects/enrichmentpipeline/) (parameter setting: *p* Adjust Method: fdr; TestMethod: FisherChiSquare).

### Identification of transcription factors
The TFs of *A. gramineus* were defined by the online tool iTAK (http://itak.feilab.net/cgi-bin/itak/online_itak.cgi)[132] with default parameters. For consistency, the other 12 species (Supplementary Data 13) used in the analysis were also defined by the online tool iTAK. After obtaining all TFs, we used a Perl script to classify transcription factors according to the classification on the iTAK website.

### Identification of *R* genes
*R* genes in *A. gramineus* were initially identified using HMMER (v3.2.1)[133] to find proteins containing the NB-ARC domain as defined by the Pfam entry (PF00931)[134]. The combination of NBS and various domains could be categorized, and additional domains can be identified using the LRR_1 (PF00560.27), LRR_2 (PF07723.7), LRR_3 (PF07725.6), LRR_4 (PF12799.1), Pkinase (PF00069.19), RPW8 (PF05659.10), TIR (PF01582.14), and zf-BED (PF02892.10) domains with an e-value cut-off of 1e-5.

### Functional gene identification
For *MADS-box*, *WRKY*, *MYB* transcription factors and Expansin, we downloaded the HMMER models of the domain structure of these genes (*MADS-box*: PF00319 and PF01486[135], *WRKY*: PF03106[136], *MYB*: PF00249[137], Expansin: DPBB_1 (PF03330) and Expansin_C (PF01357)). Then the candidate gene sequence was searched by HMMER (v3.2.1) software[133].

For *TPS*, two Pfam domains (PF01397 and PF03936)[138] were used to search in the genome by HMMER (v3.2.1)[133]. Pseudogenes and sequence lengths shorter than 500 amino acids were excluded from further analysis.

For *HD-ZIP III* and *LHC* superfamily, we used homologous alignment identification. *A. thaliana* genes were used as reference genes to compare with *A. gramineus* protein using BLASTP (evalue 1e-5)[139] to obtain candidate genes.

After initial identification, these genes were identified by building a phylogenetic tree. The maximum likelihood trees were built using IQTREE (v2.0.5)[121] after sequence alignment in MAFFT (v7.471)[118].

For *CLE*, we used homologous alignment and hmmsearch to identify *CLE* peptides of *A. gramineus*. Firstly, we downloaded *CLE* genes of *A. thaliana* and used them as reference genes to compare with *A. gramineus* proteins using BLASTP (evalue 1e-5). Secondly, we used the HMMER[133] search method to identify *CLE* candidate genes, followed two steps: (1) Building a *CLE* HMM model using hmmbuild based on an alignment of previously reported *CLE* genes provided by Goad et al.[140]; (2) The constructed HMM model was used to identify *A. gramineus CLE*

genes by hmmsearch. The result of hmmsearch was filtered by evalue 1e-5. Thirdly, candidate *CLE* genes obtained from the above two methods were further used for signal peptide site identification by the online analysis tool SignalP (https://services.healthtech.dtu.dk/service.php?SignalP-5.0). Finally, identification of these genes was confirmed by infering a maximum likelihood tree by IQTREE (v2.0.5).

### Software and algorithms
All software used in this article are listed in the Supplementary Data 14.

### Reporting summary
Further information on research design is available in the Nature Portfolio Reporting Summary linked to this article.

## Data availability
The data supporting the findings of this work are available within the paper and its Supplementary Information files. A reporting summary for this article is available as a Supplementary Information file. The whole-genome sequence data and transcriptome sequence reported in this paper have been deposited in the Genome Warehouse in National Genomics Data Center, Beijing Institute of Genomics, Chinese Academy of Sciences/China National Center for Bioinformation, under accession number GWHCBFW00000000. The assembled genome and annotation have been deposited in the Genome Sequence Archive database under accession code GWHCBFW00000000. The whole-genome sequence data, transcriptome sequence data and the assembled genome and annotation of this study also have been deposited into CNGB Sequence Archive (CNSA) of China National GeneBank DataBase (CNGBdb) with accession number CNP0002281. Source data are provided with this paper.

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

## Acknowledgements

This research was supported by grants from the National Natural Science Foundation of China (Grant No. 32000171) awarded to X.G., and the Shenzhen Municipal Government of China (KCXFZ20201221173013035). This work was supported by the Key Laboratory of Genomics, Ministry of Agriculture, BGI-Shenzhen, Shenzhen 518120, China. This work is part of the 10KP project (https://db.cngb.org/10kp/). This work is also supported by China National GeneBank (CNGB; https://www.cngb.org/). We are grateful to Shuai Yang, Yanhong Wu, Zongxin Ren, Chunlin Long and Zhuo Cheng for general technical assistance or discussion.

## Author contributions

H.L. and X.G. led and designed this project. X.G., F.W. and D.F. conceived the study. F.W., X.G. and S.K.S. wrote the manuscript. D.F. generated the whole-genome assembly. D.F., F.W., X.G., Q.L., Y.C., L.L.,

J.L., S.K.S. and S.C. performed the functional annotation, comparative genomics, and transcriptome data analyses, and generated the figures. D.F., F.W., X.G., Q.L., S.D. and Y.L. performed the mitochondrial analyses. H.L., X.G., F.W., S.K.S., S.L., and Y.G. revised and edited the manuscript. All authors read and approved the final paper.

## Competing interests

The authors declare no competing interests.
