## [Peer Review File · Nature Communications]

The genome of *Acorus* deciphers insights into early monocot evolutionReviewers' Comments:

Reviewer #1:

Remarks to the Author:

Review of Guo et al. "The genome of *Acorus* deciphers new insights into early monocot evolution"

The manuscript by Guo et al. presents the first complete genome from a species of *Acorus*. The position of *Acorus* as the sister group to all other monocots makes the data extremely interesting for evolutionary comparisons both within the monocots, but also with regards to the relatively early splits in angiosperms phylogeny. I am not an expert in all aspects of genomics and bioinformatics, thus there are technical parts of the work that I cannot judge. However, I find no reason to question the validity of most of the genomic data presented although I have some major reservations being outlined in more detail below (e.g., inclusion of results from data and methods not being described). The major problem of the manuscript resides in a weak evolutionary context and I do not think that the manuscript to a larger extent "deciphers new insights into early monocot evolution". Given the data at hand, I think the authors are missing an obvious possibility of making a really high quality and high impact paper. Another severe issue is that the language is often very unclear and contains many errors. Although I have made many annotations directly in the manuscript I have not corrected the language.

Below I will list a number of issues, but notice that I also attach the manuscript with further annotations.

#1 Throughout the manuscript the authors use the term "basal" about taxa. This is unacceptable. Two sister clades, e.g. *Acorus* and the remaining monocots, diverged at the exact same time and no group is basal compared to the other. The term "primitive" should also be avoided. One taxon can have more plesiomorphic character traits than another, but that does not make it "primitive".

#2 Be consistent in writing informal names (monocots, angiosperms, magnoliids, etc.) with either first letter in capital or not.

#3 Throughout the manuscript I think it is confusing that taxa are abbreviated especially when species belong to different genera being abbreviated in the same way. E.g., *A. gramineus* (*Acorus*), *A. trichopoda* (*Amborella*), *A. thaliana* (*Arabidopsis*). I find it most helpful to the reader simply to write all species names in full.

#4 Although *Acorus gramineus* is described as growing in "marshes", along "brooks" etc., the manuscript treats it as an aquatic plant and discusses possible adaptations to an aquatic habitat. Although *A. gramineus* can tolerate flooding it is not comparable to true submerged aquatic plants. The other species of *Acorus* (*A. calamus*) is certainly not aquatic and I would just characterize the genus as wetland plants.

#5 Make sure to explain in the introduction that *Acorales* includes only *Acoraceae* which includes only *Acorus*. And don't use the names of various ranks randomly in manuscript. Make a decision about which species you will recognize: just *A. gramineus* and *A. calamus* or which other? It makes most sense to follow the standard recognition of just two and your data seem to suggest two monophyletic groups (although it is unclear how the tree shown was rooted). The introduction of new names on P6, which was not mentioned in the introduction is confusing.

#6 If a special aim is investigating potential HGT, this topic must be introduced properly in the introduction.

#7 On P5 L142-144 results from two analyses are described as giving trees with identical topology and high support. Even if topologies in terms of branch orders are identical I doubt that branch lengths are, and are support values really identical. Only one tree is shown (which?) and no support values are given. Support values must be added and I suggest that the other tree is included as supplementary material.

#8 The results section includes description of data and results not at all mentioned in the methods section. The 612 gene/226-species data set (first mentioned on P5 L 147-148) is not described at all. Provide details of the data and the analyses.

#9 On P 6 L155-onwards transcriptome datasets and phylogenetic analyses from several specimens of *Acorus* are mentioned, but none of this is included in the methods. Include specimen data, transcriptome sequencing, all downstream analyses.... The tree (suppl. Fig. 5) shows two clades (*A. calamus* and *A. gramineus*), but how was it rooted? Any outgroup? Further data described on P 8 from the transcriptome dataset are not described in methods either.

#10 The plastome is described as 171 kb (P 6 L169) and compared to 100-200 kb "typical chloroplast genome length". A typical plastome is some 150 kb. 100 kb and 200 kb are exceptions. I wonder if there is an assembly error (perhaps inclusion of mt sequence caused by intergenomic transfer of plastid sequences to the mitogenome). Other *Acorus* plastomes (there are several in GenBank also from *A. gramineus*) are around 153-154 kb. These should be cited and if the difference is real specify what the extra sequence is. Looking briefly at the plastome shown (suppl. Fig. 6) I notice some remarkable long stretches without annotations. Is this the "insertion" or did you miss annotating *ycf* genes?

#11 P6/7: The analyses of mitochondrial genes are not sufficiently described in the methods section and the results not sufficiently described either. The figure (suppl. Fig. 8) indicates that data from 5-6 specimens of *Acorus* are included. Where do the extra data come from? A list of all included taxa/sequences (GenBank numbers) should be included. What are the 38 mt genes? Did you retrieve complete coding sequences from all genes even those with transpliced exons? Show all 38 trees, not just 6. Inclusion of support values is crucial for evaluation of the inferred events of HGT. It is further crucial to take RNA editing into account since this may bias phylogenetic analysis substantially and produce similar results as HGT!

#12 P9 L238-241: Due to the extreme sparse taxon sampling within the monocots, I think this makes little sense. The evolution is probably far more complicated.

#13 The result section generally includes a lot of unsubstantiated claims or ad hoc hypothesis (see annotation). Most of this makes little sense at least as it is. Either delete them or discuss them properly (in the discussion section) citing relevant literature to back up your ideas.

#14 The current discussing reads mostly as a summary/conclusion adding little to the previous text.

#15 The methods section lacks a lot compared to the results being described (See above). Generally, I see that many methods are identical to those described in the paper (by many of the same authors) about the *Chloranthus* genome. This is natural and completely okay, but unfortunate that a rewriting has made the description harder to follow and less accurate.

#16 It is essential to add voucher information for the specimens being used. Also for those used for additional transcriptome sequencing.

Reviewer #2:

Remarks to the Author:

Guo et al. present a high-quality nuclear genome sequence for *Acorus gramineus* and provide a study of the genome's components and placement in the angiosperm and monocot phylogenies. *Acorus* is a vital taxon to have a genome sequence representation, which means that this genome will be useful for the overall community. There are, however, a number of issues with the manuscript that need to be addressed.

General Comments:

The overall tone of the paper should be more suggestive and less absolute. There is a lot of speculation in this paper without experiments to backup claims.

The phylogenetic analyses as presented are not novel. There are more single copy genes than previous studies (563 in this study compared to the 410 in the 1KP transcriptome phylogeny paper), but that is to be expected given the smaller sample size. The taxon sampling for the 1KP paper included more diverse species of Alismatales compared to *Spirodela* and *Zostera* sampled here. This sampling scheme is similar to that of Harkess et al. 2017 (<https://www.nature.com/articles/s41467-017-01064-8>.) which demonstrated the same relationships in monocots. As written, the manuscript

pushes uncertainty of the placement of *Acorus* in the phylogeny of monocots as currently unresolved in the literature. I would say that is not an accurate depiction of the literature with the exception of mitochondrial study referenced in the text. This major goal of the paper (i on line 101) is confirmatory.

The relative relationship of monocots to the ANA grade, magnoliids + Chloranthales, and eudicots is more contentious as whole chloroplasts suggests monocots are sister to a clade of eudicots + *Ceratophyllum* (Gitzendanner et al. 2018–<https://bsapubs.onlinelibrary.wiley.com/doi/full/10.1002/ajb2.1048>). This contrasts with what nuclear genes (e.g. the 1KP transcriptome paper (2019) but explored for the history of the question in Wickett et al. 2014 (<https://www.pnas.org/doi/10.1073/pnas.1323926111>)) have shown and is confirmed by this study.

The inclusion of the transcriptomes of multiple members of *Acorus* to investigate the relationships within the genus is a welcomed addition, however, identification of putative species at this level with molecular phylogenetics needs multiple individuals (even more than two) as representatives to offset potential issues with hybridization and polyploidy.

It is not possible from the given data in the manuscript or supplemental to assess the claim of horizontal gene transfer between multiple taxa and *Acorus* mitochondria. It is true that angiosperm mitochondria can receive DNA sequences from multiple taxa, but the absence of phylograms with bootstrap support values and more information about the alignments is needed for assessment. Mitochondrial genes evolve at a slower pace than those found in the plastome or the nuclear genome. It is easily possible that the “HGT” from Alismatales is a consequence of a lack of divergence or even convergence if the taxa chosen are also found in water which can cause—as the paper states—extreme stress on plants.

Using Ks values for divergence time estimation is potentially laden with issues. The shared range of the *Acorus* peak at a similar value does suggest that within *Acorus* the lineages have not diversified at extensively different rates overall. Age estimation should be taken lightly when using Ks as it does assume similar rates and does not account for heterotachy. Note: the methods do say MCMCtree was used to estimate the age but this was not clear from the results. There may be confusion between the dating method used for the TEs and that of MCMCtree. Later in the methods (Liens 608-610) the Ks value method of estimating the divergence time is described. It is very unclear what was done and when here.

The identification of tau not being shared by Acorales and Alismatales is not new to this manuscript. It was identified in Harkess et al. 2017 and the 1KP transcriptome phylogeny paper. I agree that the absence of tau in the history of these lineages likely contributes to the lower number of transcription factors.

The end of the Results section has multiple paragraphs that identify numbers of genes of different transcription factor groups known to influence, although often broadly, some highlighted traits. These numbers are compared to other species, like *Arabidopsis*, to speculate the difference in gene number may be related to the differences in morphology. This is true, but there could be a myriad of other reasons related to expression and adaptation. It is also the case that not all members of a gene family have the same or even similar functions. These are not well-supported comparisons and have no evidence from expression or genetics to show how these gene families may impact the phenotypes. This is all predicated on the annotation being accurate, which it is likely not as all annotations are problematic in some way.

Abstract

It is unclear why there is a comparison to rice in the abstract and total number of genes. The gene count is best left for the results.

“Basal” is best not used for extant plants. Stick with “sister to the rest of the monocots.”

There is not a contraction of genes in *Acorus* because of an absence of tau, there is a relative

expansion in other lineages that have tau.

Line 54: The referenced papers depicting the changes in our understanding of monocot phylogenetics over the last 30 years are not represented of the truly important papers that have changed based on available data. For example, Givnish et al. 2018—whole chloroplast genomes and #16 in citations—and the 1KP paper—transcriptomes—gives us different patterns to examine. The self-cited paper for the *Chloranthus* genome is not a definitive paper for monocot phylogenetics.

Line 61: The monocot phylogeny is mostly resolved with good support. There are still problematic areas—e.g. Liliales placement relative to Asparagales and the sister clade to Poales.

Line 62-63: There has not been strong support of the placement of *Acorus* anywhere but sister to the rest of the monocots for over 10 years.

Line 64: Better as “is a genus of monocots” or “is a monocot genus”. Sentence is clunky with “flowering plant” in there.

Line 66: drop “particularly” in all cases

Lines 67-68: The sentence “Despite the fact that it is not a true grass, it does have a grassy look.” Should be removed. A lot of herbaceous monocots look this way.

Lines 74 and 83: Use of “primitive” is not considered accurate.

Lines 76-84: I appreciate the inclusion of the complex taxonomy of *Acorus*. There is also a potential fifth species: *Acorus americanus* (*Acorus calamus* var. *americanus*). It is not currently accepted but has been identified as diploid while other varieties in *A. calamus* have been found to be triploid or even tetraploid.

Lines 85-105: The placement of *Acorus* really has not been questioned for some time. Mitochondrial genes are not heavily used in angiosperm phylogenetics. Preference is given to whole chloroplast genomes, transcriptomes, and single/low copy genes derived from transcriptomes or sequence capture. The Peterson et al., 2016 paper does demonstrate the impact of including mitochondrial and plastid data at varying relative amounts and their influence on topology. That paper suggests that nuclear data with adequate sampling are needed, which was accomplished with the 1KP paper (<https://www.nature.com/articles/s41586-019-1693-2>). This paper should be better referenced in this manuscript because of its importance in using transcriptome data for phylogenetics. It is only referenced to cite the use of the transcriptome and to give support as *Acorus* as a group of one genus, one family, and one order (Line 360).

Line 110: Should be “k-mer”.

Lines 112-114: This sentence is a bit confusing. What is providing the “100-fold high-quality sequence coverage”? Nanopore? What is the metric for “high-quality”? Nanopore data is prone to errors and requires effort to get high-quality reads through correction. There is no indication of this in the methods. The Albacore basecaller is an older basecaller and has since been replaced in usage by ONT.

Line 139: Remove usage of “basal”.

Line 140: single is misspelled.

Line 142: “Angiosperms” should not be capitalized.

Line 143: Use "and" not "yet". These are not contradictory statements being linked.

Line 145: Remove basal.

Line 170: Use "Eighty-one" not 81 when starting a sentence.

Line 187: Use of basal.

Line 220: This lineage-specific event was also suggested in the 1KP paper.

Lines 227-228: Retaining genes in duplicate does not necessarily mean that they were maintained because of adaptation. Maintenance of paralogs is subjected to the diversity of the gene network of the genes in question, stochastic, and multiple molecular-level aspects involved in expression. It is very unclear what "difficult period of photosynthesis" would mean.

Line 265: Acorus is on its own evolutionary trajectory and being sister to the rest of monocots would not be a reason as to why it has not seen major gene expansion. Spirodela and Zostera—as stated—have undergone major adaptive changes to live in aquatic environments. The comparison to rice has the added impact of the tau, sigma, and rho WGD events which would greatly impact the diversity of gene families in the species.

Line 276: "strong metabolic ability" does not really mean anything.

Line 296: This is not a contraction. Rice has three WGD events in its lineage after the divergence from Acorus while banana has tau and three of its own. These are expansions in these taxa.

Line 298: Harkess et al. 2017 identified that Acorus and Alismatales do not share tau with other monocots. This was confirmed in the 1KP paper.

Line 304: "Less" not "lesser".

Line 306: Arabidopsis has also undergone multiple WGD events. It is unclear why it is specifically compared to Acorus for MYB TFs.

Line 308: Contracted seems less likely than expansion in other lineages.

Line 313: This may indicate adaptation to lower light, but it really is not absolute.

Line 316: Remove "the" from "the nocturnal pollinators".

Line 319: "Dicot" is not an accurate term. Use eudicot and/or magnoliids.

Lines 325-326: It is unclear how this section "highlight[s] the adaptive features of A. gramineus in fluctuating environments." What features? There are some examples of genes involved in pathogen defense and volatiles, but the description is a bit hyperbolic as there is no link to the impact of these changes in gene number to an adaptive phenotype.

Line 333: Variation in perianth whorl number need not be a function of MADS-box TF number. It can just be an adaptation.

Line 344: CLEs are extraordinarily difficult to identify and require a tiered approach. Unless this was followed (e.g. <https://pubmed.ncbi.nlm.nih.gov/27911469/>), I would be cautious in claiming Acorus has only a single CLE.

Lines 351-352: "simple plant architecture" is not really a fair comparison. Simple compared to what? What does that mean exactly?

Line 359: "support Acorus belonging to Acorales"

Line 376: Acorus only had one WGD event since the angiosperm WGD event. Better to clarify.

Line 377: Near the K-Pg boundary is up to 20 million years after by the method of estimation.

Line 398: "We noticed" is not the right way to say this. Just say they thrive near streams.

Line 410: Are marshes and brooks inherently low light? This comparison is missing something.

Lines 411-421: There is not necessarily a loss of WRKYs that are associated with living in standing water. This is a major stretch to compare to the transition onto land.

Methods

All programs should have the parameters used. The following do not: Trimmomatic, NextPolish, purge_dups, Minimap2, BLAST, all programs used to identify repeats, annotation programs, all gene characterization software, TransDecoder, CAF, and HMMER.

Line 437: What version of Alabacore?

Line 443: Citation for the Hi-C method?

Line 443: "were" not "was" for the leaves

Line 459: k-mer not K-mer

Line 460: Remove "in the present study"

Line 463: What is meant by "low-quality". What threshold? Were adapters trimmed? How were PCR duplicates determined?

Line 479: blasted or BLASTed

Lines 483-485: There is a portion of the text missing here.

Lines 490-491: Check capitalization on Juicer and 3D-DNA.

Line 578: What models were ultimately chosen for the phylogenetic analyses?

Line 587: What fossils were used? These have a large impact on divergence time estimation.

Line 712: The link does not work.

Reviewer #3:

Remarks to the Author:

In this manuscript, Guo et al. sequenced and assembled the genome of *Acorus gramineus*, a basal monocotyledon species whose classification and phylogenetic position had been subject to long-

standing debate. Thanks to long-read (Nanpore) and short-read (Illumina) sequencing coupled to Hi-C, the authors were able to obtain a high-quality reference genome with an N50 of 36 Mbp. The analysis of the evolution of the *A. gramineus* genome and its comparison with the genomes of other sequenced plant species led to several key findings :

(i) Phylogenomic analysis confirmed that *A. gramineus* is a basal monocot species and that *Acorus* species can be divided into two main clades: *A. calamus* and *A. gramineus*.

(ii) Phylogenetic analysis of mitochondrial genes revealed multiple horizontal transfers from other monocot and dicot species which could explain the misplacement of this species in phylogenies using mitochondrial genes.

(iii) *A. gramineus* did not undergo the previously reported tau (τ) WGD (Whole Genome Duplication) event shared by all other monocot clades.

(iv) *A. gramineus* has very few genes compared to other monocots, including resistance genes and some transcription factors, suggesting gene contraction related to the ecology and lifestyle of this species.

I find the manuscript well written and very pleasant to read. The analyses conducted are globally well done. However I have some concerns about the lack of clarity sometimes regarding certain analyses and the interpretation of some results. Here are my comments:

Figure 1. Morphology and genomic architecture of *A. gramineus*

I have some comment regarding the density of genes and LTRs across the *A. gramineus* genome: For chr 2, chr8, chr9, chr10, chr11 and chr12 we can see that the distribution of genes and LTRs is reversed compared to what is "expected". This even more true for the distribution of gypsy elements (Figure 1-b G) that are characteristic of pericentromeric regions and are in *A. gramineus* assembly mainly located in chromosome ends (chr 2,8,10,11 and 12). We know that in plants peri-centromeric regions are TEs-rich especially LTRs retrotransposons while chromosome arms are genes-rich. This gene-TEs density pattern inversion can be explained by :

1- Assembly errors (inversion of chromosome arms for instance) that will lead to a false chromosome structure.

2- Genomic events such as fissions/fusions, inversion, translocations and nested chromosome insertion that have occurred during the "recent" genome evolution of *A. gramineus*. Those chromosome rearrangement result generally in a "non-canonical" TEs-genes density across chromosomes.

In order to investigate all these possible scenarios, comparison of the *A. gramineus* genome with that of closely related species could provide a clear picture of recent genomic rearrangement events that may have occurred in *A. gramineus*. Since the authors do not have the genome of closely related species, an alternative analysis will consist of the characterization of centromeric and telomeric repeats. This will allow to determine precisely the position of centromeres and telomeres validating the structure of the assembled chromosomes but also could reveal the position of paleocentromeres and paleotelomeres. This will allow the authors to better understand these "inverted" density profiles on the 6 *A. gramineus* chromosomes. It is true that it is often difficult to properly assemble the centromeric and telomeric regions but with Nanopore sequencing coupled to Illumina and HiCi sequencing, these regions should normally be present in the assembly performed in this study. At the same time, this analysis will give additional guarantees on the quality of genome assembly.

Page 5 : 132-136 "Long terminal repeat (LTR) retrotransposons were the most prevalent type of transposable element (TE) among these repeats, representing nearly 35% of the genome, including 23.1% LTR/Gypsy and 5.9% LTR/Copia retro-elements (Supplementary Tables 7)."

A description of the different TE families found and their relative frequencies in the genome is missing. Although LTRs are expected to represent the largest fraction of these elements, it is important in a paper describing the genome of a new species to annotate all TE families including Class II elements such as CACTA, hAT, MuDR or Helitrons. Finally it will be appreciated and useful that the authors provide the GFF file and fasta file of both genes and TEs annotation.

Page 5 : 139-141 : "To investigate the taxonomic status of Acorales, we first selected 563 single-copy ortholog sets (SCG) from four eudicots, four monocots, three Magnoliids, two basal Angiosperms, and *Ceratophyllum demersum*."

The authors should clarify why they chose these particular species. Is it because they are "representative"? Because of their position in the phylogenetic tree?

Page 5 : line 148-149 : "Both the phylogenetic trees based on concatenated sequence alignments showed similar topological structures, placing Acorales at the most basal position of the monocot lineage"

Do the authors mean "both phylogenetic trees"? The previous one using 14 species and the one with 223 species? or the two phylogenetic trees based on the concatenated sequence alignment and the coalescent approach provided the same results for the 223 species? The authors should clarify in the manuscript and or in the method section. In this regard I have not been able to understand which of the phylogenetic trees were obtained with a concatenated alignment or using the coalescent model approach. Second, what is the difference between the two phylogenetic trees. The one with 226 species should be more informative because it includes the species used in the first phylogenetic tree (with 14 species) and uses more orthologous genes. What is the rationale for using two sets of single-copy orthologous genes? Do they overlap? Do the 612 genes include the 563 genes plus some duplicated genes?

The authors should clarify these points

Page 6 : Line 174-176 : In order to assess the possibility of mitochondrial gene transfer occurred in the mitogenome, we constructed the phylogenetic trees together with the mitochondrial genes of *A. gramineus*.

If I understand well the identification of horizontal gene transfer was based solely on phylogenetic incongruence of 18 mitochondrial genes among a total of 38 genes. Phylogenetic incongruence alone is not a strong argument in favor of HGT. Can be also due to other evolutionary and technical bias. It is important to consider also the high similarity criteria to ensure that the observed similarity of the transferred gene is higher than that of orthologous mitochondrial genes. Additionally it is important also to show that the transferred mitochondrial genes are not in syntenic/collinear position. In other words, if a horizontal transfer occurs that chance that the insertion of the foreign genes occurs exactly at the same position as the ancestral homologous genes is extremely low.

Page 6 : Line 179-180 : "The transferred genes were mainly from monocotyledonous and dicotyledonous plants."

Are the authors sure of the direction of the HGT. It is possible that some or all of these HGTs occurred from monocots and dicots to *A. gramineus*. How could the authors determine the direction of HGT and how could they reject the possibility that these transfers occurred in the other direction

Page 7 : 194-196 : The synteny analysis of *A. gramineus* with *A. trichopoda* showed 565 syntenic blocks that covered 16.0% and 14.7% of the assembled genomes, respectively.

Do you mean 16.0 % and 14.7 % of total genes ?

It is not clear to me if the authors have taken into account the intergenic regions in the calculation of the percentage of syntenic regions. It is obvious that intergenic sequences, and in particular TEs, are very dynamic and evolve rapidly so that this measure does not give a clear idea of the percentage of syntenic genes

Page 7 : Line 204-207 : "To more precisely infer the timing of the WGD in the *A. gramineus* genome, intragenomic and interspecies homologue Ks (synonymous substitutions per synonymous site) distributions were estimated"

As far as I understand, the identification of horizontal gene transfer here was based solely on the phylogenetic incongruence of 18 mitochondrial genes out of a total of 38 genes. Phylogenetic incongruence alone is not a strong argument for HGT. It may also be due to other biases such as incomplete lineage sorting and hidden paralogy. It is important to also consider high similarity criteria to ensure that the observed similarity of the transferred gene is higher than that of the orthologous mitochondrial genes. In addition, it is also important to show that the transferred mitochondrial genes are not in a syntenic/colinear position. In other words, if a horizontal transfer occurs, the probability that the insertion of the foreign genes occurs at exactly the same position as the ancestral orthologous genes is extremely low.

Figure 3 : please correct "c. The intragenomic Ks distribution Karyotype evolution".

Do you mean b part of the figure.

Page 8 : Line 221 : "Following the WGD event, *A. gramineus* retained a total of 7,902 genes"

Do you mean paralogous pairs ?

Page8-9: "However, in the present study, there were 672 specific gene families in *A. gramineus*, lower than *S. polyrhiza* and *Z. marina*. Compared to 4,960 specific gene families in *O. sativa*, the number of specific gene families in *A. gramineus* comparatively fewer. This suggests that being a basal monocots species, *A. gramineus* did not experience a dramatic genome expansion."

As the authors show, *A. gramineus* underwent only one WGD event, unlike other Monocot plants. Furthermore, we know that following WGD, paralogous genes could rapidly diverge, allowing the emergence of new specific genes. The question I have is the following: Is this low number of specific genes observed in *A. gramineus* compared to rice for example is because of the low number of duplications experienced by this species?

Page 14 : "Line 402 addition, the R-genes also showed contraction in gene numbers, particularly the NBS genes."

What makes the authors say that this is a contraction of the gene number? We know that R-genes evolve by duplication such as tandem duplications. It is impossible to know the number of R genes in the ancestors of monocotyledons on the basis of the sequenced modern Monocots genomes since these genes are subject to a significant rate of duplication during evolution even at the level of populations. It seems to me therefore difficult to draw the conclusion the relatively low number of R-genes *A. gramineus* compared to other Monocot species that is due to a contraction of R-genes. This comment also applies to the MADS-box gene and to other gene families in which the expansion is often explained by intensive local duplication.

Other comments

There is no mention of Figure 1 in the manuscript.

Reviewer #1 (Remarks to the Author):

Review of Guo et al. “The genome of Acorus deciphers new insights into early monocot evolution”

The manuscript by Guo et al. presents the first complete genome from a species of Acorus. The position of Acorus as the sister group to all other monocots makes the data extremely interesting for evolutionary comparisons both within the monocots, but also with regards to the relatively early splits in angiosperms phylogeny. I am not an expert in all aspects of genomics and bioinformatics, thus there are technical parts of the work that I cannot judge. However, I find no reason to question the validity of most of the genomic data presented although I have some major reservations being outlined in more detail below (e.g., inclusion of results from data and methods not being described). The major problem of the manuscript resides in a weak evolutionary context and I do not think that the manuscript to a larger extent “deciphers new insights into early monocot evolution”. Given the data at hand, I think the authors are missing an obvious possibility of making a really high quality and high impact paper. Another severe issue is that the language is often very unclear and contains many errors. Although I have made many annotations directly in the manuscript I have not corrected the language.

Below I will list a number of issues, but notice that I also attach the manuscript with further annotations.

Response: Thanks for the comments. We have carefully revised the manuscript throughout, including a point-by-point response to all comments below and annotations in the pdf file.

#1 Throughout the manuscript the authors use the term “basal” about taxa. This is unacceptable. Two sister clades, e.g. Acorus and the remaining monocots, diverged at the exact same time and no group is basal compared to the other. The term “primitive” should also be avoided. One taxon can have more plesiomorphic character traits than another, but that does not make it “primitive”.

Response: Thanks for the suggestion. We have avoided “basal” and “primitive” in the revised version and replaced as “sister to the other monocots”.

#2 Be consistent in writing informal names (monocots, angiosperms, magnoliids, etc.) with either first letter in capital or not.

Response: We have checked and modified all the informal names in small letter.

#3 Throughout the manuscript I think it is confusing that taxa are abbreviated especially when species belong to the different genera being abbreviated in the same way. E.g., A. gramineus (Acorus), A. trichopoda (Amborella), A. thaliana (Arabidopsis). I find it most helpful to the reader simply to write all species names in full.

Response: Apologies for the inconsistencies. We have carefully checked and used full names as per the suggestions.

#4 Although *Acorus gramineus* is described as growing in “marshes”, along “brooks” etc., the manuscript treats it as an aquatic plants and discussed possible adaptations to an aquatic habitat. Although *A. gramineus* can tolerate flooding it is not comparable to true submerged aquatic plants. The other species of *Acorus* (*A. calamus*) is certainly not aquatic and I would just characterize the genus as wetland plants.

Response: Thanks for the nice suggestion. We have now revised and described *Acorus* as a wetland species.

#5 Make sure to explain in the introduction that *Acorales* includes only *Acoraceae* which includes only *Acorus*. And don't use the names of various ranks randomly in manuscript. Make a decision about which species you will recognize: just *A. gramineus* and *A. calamus* or which other? It makes most sense to follow the standard recognition of just two and your data seem to suggest two monophyletic groups (although it is unclear how the tree shown was rooted). The introduction of new names on P6, which was not mentioned in the introduction is confusing.

Response: Thanks for the nice suggestion. We have carefully revised the use of taxonomic ranks throughout the manuscript. For the number of species in *Acorus*, there are just two species (*A. gramineus* and *A. calamus*) in the standard recognition. Based on the morphological characters, we think there are more than two species in *Acorus*. *A. macrospadiceus* and *A. tatarinowii* have very different morphology from the other two, although they are treated as synonyms. We have added the relevant introduction of all species names used in the manuscript. L84-L98.

#6 If a special aim is investigating potential HGT, this topic must be introduced properly in the introduction.

Response: Thanks for the nice suggestion. We added this part in introduction. L114-L118 “The transfer of genetic material across species boundaries is known as horizontal gene transfer (HGT), and is widely perceived as a driving factor in prokaryotic evolution, but it is also gaining popularity in eukaryotes⁵⁴. Recent research has further revealed that the evolution of land plants was driven by two major episodes of horizontal gene transfer^{55,56}.”.

#7 On P5 L142-144 results from two analyses are described as giving trees with identical topology and high support. Even if topologies in terms of branch orders are identical I doubt that branch lengths are, and are support values really identical. Only one tree is shown (which?) and no support values are given. Support values must be added and I suggest that the other tree is included as supplementary material.

Response: Thanks for pointing this out. We have now updated the figure legend (as below) and added trees constructed by ASTRAL as well (Supplementary Fig. 4 and Supplementary Fig. 6).

L933-L934 “The phylogenetic tree constructed by IQTREE based on a concatenated dataset of 562 single-copy genes, with a bootstrap value of 100 for all branches.”

#8 The results section includes description of data and results not at all mentioned in the methods section. The 612 gene/226-species data set (first mentioned on P5 L 147-148) is not described at all. Provide details of the data and the analyses.

Response: Added to the methods section L705-L713:

“In addition, we built a phylogenetic tree from an extended species sampling from 223 species in total, include 221 angiosperms species, and two gymnosperms as outgroups (Supplementary table 8c). The amino acid sequences from the whole species using BLASTP with an e-value of 1e-5, and then grouped using OrthoFinder (v 2.3.7)¹⁵⁰. Here, each gene was required to include sequences from more than 160 species for low-copy genes, which resulted in 612 genes. As same as above, which was constructed phylogenetic tree using concatenation and coalescent methods as described above from the 562 single-copy gene families. For concatenation method, the best model was selected by IQTREE¹⁴⁸ (v2.0.5) is GTR+F+R10.”.

#9 On P 6 L155-onwards transcriptome datasets and phylogenetic analyses from several specimens of *Acorus* are mentioned, but none of this is included in the methods. Include specimen data, transcriptome sequencing, all downstream analyses.... The tree (suppl. Fig. 5) shows two clades (*A. calamus* and *A. gramineus*), but how was it rooted? Any outgroup? Further data described on P 8 from the transcriptome dataset are not described in methods either.

Response: We added the specimen information in supplementary table 8b. The transcriptome sequencing, assembly, the CDS prediction and phylogenetic analyses are added in the method section.

Transcriptome sequencing: L547-L554 “The young leaf, stem, and root tissues of *Acorus gramineus* and other five *Acorus* spp. samples were collected for transcriptome sequencing (Supplementary Table 1). Total RNA was isolated using the TIANGEN Kit with DNase I and then processed using the NEBNextUltraTM RNA Library Prep Kit to create a pair-end library with a 250bp insert size. Libraries were barcoded and pooled together as an input to the BGI-DIPSEQ platform for sequencing. Following the removal of low-quality data, 6 Gb of 100 bp paired-end data for each tissue were used for further RNA-seq analysis.”.

Transcriptome assembly and coding sequences prediction: L684-L688 “Prior to assembly, we retrieved the high-quality reads by removing adaptor sequences and filtered low-quality reads by using Trimmomatic (v0.40)¹⁰⁸. The resulting high-quality reads were then *de novo* assembled with the Trinity (v2.6.6) program¹³⁸. Protein sequences and coding sequences of transcripts were predicted using TransDecoder (v5.3.0) (<https://github.com/TransDecoder/TransDecoder>).”.

Phylogenetic analyses method: L714-L722 “For the interspecific phylogenetic tree among *Acorus*, we added six transcriptome of *Acorus* samples sequenced in this study and *Acorus calamus* var. *americanus* transcriptome data published in 1KP¹⁵(Supplementary Table 8b) to 15 seed plants dataset. The amino acid sequences from the whole species using BLASTP with an e-value of 1e-5, and then grouped using OrthoFinder (v 2.3.7). Here,

each orthologous was required to include sequences from more than 18 species for low-copy orthologous genes, which resulted in 1632 orthologous. As same as above, which was constructed phylogenetic tree using concatenation methods by IQTREE¹⁴⁸ (v2.0.5) with best model is GTR+F+R4.”.

#10 The plastome is described as 171 kb (P 6 L169) and compared to 100-200 kb “typical chloroplast genome length”. A typical plastome is some 150 kb. 100 kb and 200 kb are exceptions. I wonder if there is an assembly error (perhaps inclusion of mt sequence caused by intergenomic transfer of plastid sequences to the mitogenome). Other *Acorus* plastomes (there are several in GenBank also from *A. gramineus*) are around 153-154 kb. These should be cited and if the difference is real specify what the extra sequence is. Looking briefly at the plastome shown (suppl. Fig. 6) I notice some remarkable long stretches without annotations. Is this the “insertion” or did you miss annotating *ycf* genes?

Response: Thanks for pointing this out. We carefully checked our chloroplast assembly results and found that the oversize of 171 kb was due to a statistical error. The chloroplast genome length is 153,062 bp, which was updated in the result part. We also checked the annotation carefully, and found that the long stretches without annotations are *ycf1* and *ycf2* genes. The problem was due to parameter setting in the annotation step. In the first round of submission, we chose the default Reference Dataset parameter ‘1.43-plastomes’ on CPGAVAS2 online tool. The missing of *ycf1* and *ycf2* genes might be due to low sequence similarity between *Acorus gramineus* and species of other lineages (see below BLASTP results in NCBI) under the default parameter ‘1.43-plastomes’. In this revised version, we changed the Reference Dataset parameter to ‘3. Custom reference in GenBank format’, and used published *Acorus gramineus* chloroplast genome (NC_026299) as a reference and successfully obtained *ycf1* and *ycf2* genes.

#11 P6/7: The analyses of mitochondrial genes are not sufficiently described in the methods section and the results not sufficiently described either. The figure (suppl. Fig. 8)

indicates that data from 5-6 specimens of *Acorus* are included. Where do the extra data come from? A list of all included taxa/sequences (GenBank numbers) should be included. What are the 38 mt genes? Did you retrieve complete coding sequences from all genes even those with transpliced exons? Show all 38 trees, not just 6. Inclusion of support values is crucial for evaluation of the inferred events of HGT. It is further crucial to take RNA editing into account since this may bias phylogenetic analysis substantially and produce similar results as HGT!

Response: Thanks for the comments. We removed RNA editing sites and reconstructed all 38 single gene trees to evaluate mitochondrial horizontal gene transfer events. All methods and results in the manuscript were updated.

Data of 5 specimens of *Acorus* were newly generated in this study. The specimens of *Acorus* used in this study are listed in Supplementary table 1. All specimens used to phylogenetic analysis are listed in Supplementary table 10. All 38 mt gene trees with the bootstrap value are listed in Supplementary Fig. 9.

Revised method: L796-L819

“For mitochondria genes, to rule out the influence of RNA editing sites, we excluded all 1,537 RNA editing sites identified in the mitochondrial genomes of the angiosperm ordinal representatives in our individual gene alignment to alleviate the negative impact of RNA editing in phylogenomics analyses¹⁵⁸. All the protein-coding genes were aligned individually using MAFFT (v7.471)¹⁵⁹ to build amino acid alignments. Poorly aligned regions were trimmed by trimal (v1.4.1)¹⁶⁰, such that nucleotide alignments were produced based on the corresponding amino acid alignments after the removal of ambiguous positions with PAL2NAL (v14.1)¹⁶¹. After removing stop codons, and pseudogenes, the alignments were used for phylogenetic analyses in IQTREE¹⁴⁸, with 1000 ultrafast bootstrap. Two gymnosperms, *Ginkgo biloba* and *Cycas taitungensis* were used as outgroups in the cp and mt phylogenetic analyses.

To identified horizontal gene transfer events (HGT), 38 single mitochondria gene trees were concatenated. All the genes were aligned individually using MAFFT (v 7.471)¹⁵⁹ to build nucleotide alignments. Poorly aligned regions were trimmed by trimal (v 1.4.1)¹⁶⁰, IQTREE¹⁴⁸ was used to infer a concatenation-based species tree with 1000 ultrafast bootstrap. We evaluated horizontal gene transfer events (HGT) based on phylogenetic topology and node supports. Specifically, we first filtered those gene trees (as ‘unclassified’ in supplementary fig. 9), in which monocots, eudicots, magnoliids and Chloranthales are non-monophyletic or ANA grade is not at the basal place. Second, we consider those gene trees as candidates to evaluate HGT if the unexpected placement of *Acorus gramineus* was observed i.e. *Acorus gramineus* is closely related to other lineages rather than as the sister to the rest monocots. Importantly, only cases with node bootstrap support $\geq 75\%$ would be inferred as HGT events.”

Revised result: L221-L225 “Out of 38 based on single mitochondrial gene phylogenetic trees, we inferred 12 likely gene transfer events (Fig. 2d, Supplementary Fig. 10) occurred between other lineages of angiosperms, involving magnoliids (*ccmFC*, *nad5*, *cob*, *rps4*),

monocots (*rpl5, atp1, ccmB, nad4, rpl2*), Alismatales (*rpl5, atp1, ccmB, nad4*) and eudicots (*cox2*).”.

#12 P9 L238-241: Due to the extreme sparse taxon sampling within the monocots, I think this makes little sense. The evolution is probably far more complicated.

Response: Thanks for pointing this out. We have removed this ambiguous statement including the associated figure 3C in order to avoid confusion.

#13 The result section generally includes a lot of unsubstantiated claims or ad hoc hypothesis (see annotation). Most of this makes little sense at least as it is. Either delete them or discuss them properly (in the discussion section) citing relevant literature to back up your ideas.

Response: Thank you for the suggestions. The manuscript has been revised, and the claims made in the study are toned down accordingly.

#14 The current discussing reads mostly as a summary/conclusion adding little to the previous text.

Response: We have carefully revised the manuscript and added more discussion to the relevant results.

#15 The methods section lacks a lot compared to the results being described (See above). Generally, I see that many methods are identical to those described in the paper (by many of the same authors) about the *Chloranthus* genome. This is natural and completely okay, but unfortunate that a rewriting has made the description harder to follow and less accurate.

Response: The sentences have been now revised for more clarity.

#16 It is essential to add voucher information for the specimens being used. Also for those used for additional transcriptome sequencing.

Response: Added, The tissue samples of *Acorus gramineus* (Voucher number: JQP20200401GX, SCBG) were collected from Zhangzhou, Fujian, China (Supplementary Table 8b).

Reviewer #2 (Remarks to the Author):

Guo et al. present a high-quality nuclear genome sequence for *Acorus gramineus* and provide a study of the genome's components and placement in the angiosperm and monocot phylogenies. *Acorus* is a vital taxon to have a genome sequence representation, which means that this genome will be useful for the overall community. There are, however, a number of issues with the manuscript that need to be addressed.

General Comments:

#1 The overall tone of the paper should be more suggestive and less absolute. There is a lot of speculation in this paper without experiments to backup claims.

Response: Thanks for the suggestions. The manuscript has been revised, and the claims made in the study are toned down accordingly.

#2The phylogenetic analyses as presented are not novel. There are more single copy genes than previous studies (563 in this study compared to the 410 in the 1KP transcriptome phylogeny paper), but that is to be expected given the smaller sample size. The taxon sampling for the 1KP paper included more diverse species of Alismatales compared to *Spirodela* and *Zostera* sampled here. This sampling scheme is similar to that of Harkess et al. 2017 (<https://www.nature.com/articles/s41467-017-01064-8>.) which demonstrated the same relationships in monocots. As written, the manuscript pushes uncertainty of the placement of *Acorus* in the phylogeny of monocots as currently unresolved in the literature. I would say that is not an accurate depiction of the literature with the exception of mitochondrial study referenced in the text. This major goal of the paper (i on line 101) is confirmatory.

Response: We agree that the major goal of the paper is to confirm the sister relationship between *Acorus* and the rest monocots using genomic data and detect potential HGT events in mitochondrial. We have revised the manuscript throughout to be confirmatory and cited several previous phylogenetic studies that retrieved the same relationships in monocots as suggested.

#3The relative relationship of monocots to the ANA grade, magnoliids + Chloranthales, and eudicots is more contentious as whole chloroplasts suggests monocots are sister to a clade of eudicots + *Ceratophyllum* (Gitzendanner et al. 2018–<https://bsapubs.onlinelibrary.wiley.com/doi/full/10.1002/ajb2.1048>). This contrasts with what nuclear genes (e.g. the 1KP transcriptome paper (2019) but explored for the history of the question in Wickett et al. 2014 (<https://www.pnas.org/doi/10.1073/pnas.1323926111>) have shown and is confirmed by this study.

Response: Thanks for pointing out this. The conflicting position of monocots between chloroplast trees and nuclear trees, and likely underlying causes have been investigated in two *Chloranthus* papers published last year and phylotranscriptomic data of 92 streptophytes (Guo et al., 2021; Yang et al., 2021; Wickett et al. 2014). Therefore, the conflicting position was not further discussed in this study. We have added a brief note to mention this content in the revised manuscript. L175-L177.

#4The inclusion of the transcriptomes of multiple members of *Acorus* to investigate the relationships within the genus is a welcomed addition, however, identification of putative species at this level with molecular phylogenetics needs multiple individuals (even more than two) as representatives to offset potential issues with hybridization and polyploidy.

Response: That's a nice suggestion indeed. We demonstrated *Acorus calamus* is polyphyletic based on the current sampling. But due to lack of transcriptome data from multiple individuals, the taxonomic treatment is beyond the scope of present study. We hope to consider and include your suggestion in our subsequent studies.

#5It is not possible from the given data in the manuscript or supplemental to assess the claim of horizontal gene transfer between multiple taxa and *Acorus* mitochondria. It is true that angiosperm mitochondria can receive DNA sequences from multiple taxa, but

the absence of phylograms with bootstrap support values and more information about the alignments is needed for assessment. Mitochondrial genes evolve at a slower pace than those found in the plastome or the nuclear genome. It is easily possible that the “HGT” from Alismatales is a consequence of a lack of divergence or even convergence if the taxa chosen are also found in water which can cause—as the paper states—extreme stress on plants.

Response: Thanks for the comments. We removed RNA editing sites and reconstructed all 38 single gene trees to evaluate mitochondrial horizontal gene transfer events. All methods and results in the manuscript were updated.

Data of 5 specimens of *Acorus* were newly generated in this study. The specimens of *Acorus* used in this study are listed in Supplementary table 1.

All specimens used to phylogenetic analysis are listed in Supplementary table 10.

All 38 mt gene trees with the bootstrap value are listed in Supplementary Fig. 9.

Revised method: L796-L819

“For mitochondria genes, to rule out the influence of RNA editing sites, we excluded all 1,537 RNA editing sites identified in the mitochondrial genomes of the angiosperm ordinal representatives in our individual gene alignment to alleviate the negative impact of RNA editing in phylogenomics analyses. All the protein-coding genes were aligned individually using MAFFT (v7.471) to build amino acid alignments. Poorly aligned regions were trimmed by trimAl (v1.4.1), such that nucleotide alignments were produced based on the corresponding amino acid alignments after the removal of ambiguous positions with PAL2NAL (v14.1). After removing stop codons, and pseudogenes, the alignments were used for phylogenetic analyses in IQTREE, with 1000 ultrafast bootstrap. Two gymnosperms, *Ginkgo biloba* and *Cycas taitungensis* were used as outgroups in the cp and mt phylogenetic analyses.

We evaluated horizontal gene transfer events (HGT) based on phylogenetic topology and node supports. Specifically, we first filtered those gene trees (as ‘unclassified’ in supplementary fig. 9), in which monocots, eudicots, magnoliids and Chloranthales are non-monophyletic or ANA grade is not at the basal place. Second, we consider those gene trees as candidates to evaluate HGT if the unexpected placement of *Acorus gramineus* was observed i.e. *Acorus gramineus* is closely related to other lineages rather than as the sister to the rest monocots. Importantly, only cases with node bootstrap support $\geq 75\%$ would be inferred as HGT events.”

Revised result: L221-L225 “Out of 38 based on single mitochondrial gene phylogenetic trees, we inferred 12 likely gene transfer events (Fig. 2d, Supplementary Fig. 10) occurred between other lineages of angiosperms, involving magnoliids (*ccmFC*, *nad5*, *cob*, *rps4*), monocots (*rpl5*, *atp1*, *ccmB*, *nad4*, *rpl2*), Alismatales (*rpl5*, *atp1*, *ccmB*, *nad4*) and eudicots (*cox2*).”

#6 Using K_s values for divergence time estimation is potentially laden with issues. The shared range of the *Acorus* peak at a similar value does suggest that within *Acorus* the lineages have not diversified at extensively different rates overall. Age estimation should

be taken lightly when using Ks as it does assume similar rates and does not account for heterotachy. Note: the methods do say MCMCtree was used to estimate the age but this was not clear from the results. There may be confusion between the dating method used for the TEs and that of MCMCtree. Later in the methods (Liens 608-610) the Ks value method of estimating the divergence time is described. It is very unclear what was done and when here.

Response: MCMCtree was used to estimate species divergence times, please refer to L725-L732. Ks was used to estimate the time of the occurrence of a WGD event and has been modified in the method section L755-L757. The sentence reads as “The whole-genome duplication events time of *Acorus gramineus* was calculated by combining the Ks value with synonymous substitutions at each site per year (r) by using the formula: divergence date (T) = Ks / (2*r).”

#7The identification of tau not being shared by Acorales and Alismatales is not new to this manuscript. It was identified in Harkess et al. 2017 and the 1KP transcriptome phylogeny paper. I agree that the absence of tau in the history of these lineages likely contributes to the lower number of transcription factors.

Response: Yes, our result support Acorales didn’t experience the tau (τ) WGD event, which is consistent with Harkess et al. 2017 and the 1KP transcriptome phylogeny paper. We cited these two papers in the revised version. Please refer to L261-L262 “This result is consistent with the Harkess et al. 2017 and the 1KP transcriptome phylogeny paper^{15,76}.”.

#8The end of the Results section has multiple paragraphs that identify numbers of genes of different transcription factor groups known to influence, although often broadly, some highlighted traits. These numbers are compared to other species, like Arabidopsis, to speculate the difference in gene number may be related to the differences in morphology. This is true, but there could be a myriad of other reasons related to expression and adaptation. It is also the case that not all members of a gene family have the same or even similar functions. These are not well-supported comparisons and have no evidence from expression or genetics to show how these gene families may impact the phenotypes. This is all predicated on the annotation being accurate, which it is likely not as all annotations are problematic in some way.

Response: Yes, we do agree with you. We have toned down the conclusion and revised the sentence as “Overall, these findings likely suggest the adaptive features of *A. gramineus* in fluctuating environments. But there could be a myriad of other reasons related to expression and adaptation that should also be considered in future studies.” L429-L432.

Abstract

#9 It is unclear why there is a comparison to rice in the abstract and total number of genes. The gene count is best left for the results.

Response: Thanks, the suggested correction has been appended as “Here we present the 375.3 Mb genome of *Acorus gramineus* having 50% fewer genes compared to other monocots with similar genome size.”.

#10 “Basal” is best not used for extant plants. Stick with “sister to the rest of the monocots.”

Response: Yes, we do agree. We have replaced “basal” with “sister to the rest of the monocots”.

#11 There is not a contraction of genes in *Acorus* because of an absence of tau, there is a relative expansion in other lineages that have tau.

Response: yes, we agree with that, and modified the relevant result interpretation. Please refer to L35-L39. The revised sentence now reads as “Interestingly, we discovered that the Acorales did not experience the tau (τ) whole-genome duplication like other monocot clades, which likely helped *Acorus* to maintain their ancestral state, and no large-scale expansion in gene families and transcription factors were observed in *Acorus gramineus*.”.

#12 Line 54: The referenced papers depicting the changes in our understanding of monocot phylogenetics over the last 30 years are not represented of the truly important papers that have changed based on available data. For example, Givnish et al. 2018—whole chloroplast genomes and #16 in citations—and the 1KP paper—transcriptomes—gives us different patterns to examine. The self-cited paper for the *Chloranthus* genome is not a definitive paper for monocot phylogenetics.

Response: Thanks for pointing this out. These important references are updated in the introduction.

#13 Line 61: The monocot phylogeny is mostly resolved with good support. There are still problematic areas—e.g. Liliales placement relative to Asparagales and the sister clade to Poales.

Response: Revised. The sentence now reads as “These studies generated well-supported monocot phylogenies¹⁷⁻²⁵, with the majority of clades completely resolved although problems remain in Liliales placement relative to Asparagales and the sister clade to Poales.” L68-L71.

#14 Line 62-63: There has not been strong support of the placement of *Acorus* anywhere but sister to the rest of the monocots for over 10 years.

Response: Thanks for pointing this out. Here we mean ‘taxonomic placement of Acoraceae remains ambiguous.’ We have revised the sentence and it now reads as “*Acorus* (sweet flag) is the only genus of the Acoraceae family, whose taxonomic placement remains ambiguous since its establishment.” L74-L76.

#15 Line 64: Better as “is a genus of monocots” or “is a monocot genus”. Sentence is clunky with “flowering plant” in there.

Response: This sentence has been revised to: L74-L76. The sentence now reads as “*Acorus* (sweet flag) is the only genus of the Acoraceae family.”.

#16 Line 66: drop “particularly” in all cases

Response: Deleted.

#17 Lines 67-68: The sentence “Despite the fact that it is not a true grass, it does have a grassy look.” Should be removed. A lot of herbaceous monocots look this way.

Response: removed.

#18 Lines 74 and 83: Use of “primitive” is not considered accurate.

Response: Deleted the use of “primitive”. The sentence has been revised as: L84-L86: “*Acorus* was originally a part of the Araceae family before being renamed as Acoraceae”.

#19 Lines 76-84: I appreciate the inclusion of the complex taxonomy of *Acorus*. There is also a potential fifth species: *Acorus americanus* (*Acorus calamus* var. *americanus*). It is not currently accepted but has been identified as diploid while other varieties in *A. calamus* have been found to be triploid or even tetraploid.

Response: We do agree with you, and therefore used additional transcriptome data from *Acorus americanus* (*Acorus calamus* var. *americanus*) for interspecific analysis.

#20 Lines 85-105: The placement of *Acorus* really has not been questioned for some time. Mitochondrial genes are not heavily used in angiosperm phylogenetics. Preference is given to whole chloroplast genomes, transcriptomes, and single/low copy genes derived from transcriptomes or sequence capture. The Peterson et al., 2016 paper does demonstrate the impact of including mitochondrial and plastid data at varying relative amounts and their influence on topology. That paper suggests that nuclear data with adequate sampling are needed, which was accomplished with the 1KP paper (<https://www.nature.com/articles/s41586-019-1693-2>). This paper should be better referenced in this manuscript because of its importance in using transcriptome data for phylogenetics. It is only referenced to cite the use of the transcriptome and to give support as *Acorus* as a group of one genus, one family, and one order (Line 360).

Response: 1KP paper was cited as necessary in the revised manuscript. For the placement of *Acorus*, we agree with you that studies based on plastid and transcriptome data always place *Acorus* as the sister to the rest monocots. We have revised the description to be confirmatory. It is also proper to introduce the conflict phylogenetic placement of *Acorus* in three studies using Mitochondrial data, as we have a section about HGT in Mitochondrial, which explained the gene tree incongruence.

#21 Line 110: Should be “k-mer”.

Response: Corrected throughout the manuscript.

#22 Lines 112-114: This sentence is a bit confusing. What is providing the “100-fold high-quality sequence coverage”? Nanopore? What is the metric for “high-quality”? Nanopore data is prone to errors and requires effort to get high-quality reads through correction. There is no indication of this in the methods. The Albacore basecaller is an older basecaller and has since been replaced in usage by ONT.

Response: Modified to long- read sequencing data, and provided sequence value in Supplementary Table 1.

About basecalling, we consulted the sequencing group, The base calling was performed using Guppy (v4.0.11) in MinKNOW package. Modified method part L535-L536.

L133-L135: We achieved ~47.6 Gb of Nanopore long read sequencing data, approximately 100-fold coverage of the genome.

#23 Line 139: Remove usage of “basal”.

Response: Removed.

#24 Line 140: single is misspelled.

Response: Corrected.

#25 Line 142: “Angiosperms” should not be capitalized.

Response: Corrected.

#26 Line 143: Use “and” not “yet”. These are not contradictory statements being linked.

Response: Corrected.

#27 Line 145: Remove basal.

Response: Removed.

#28 Line 170: Use “Eighty-one” not 81 when starting a sentence.

Response: Corrected.

#29 Line 187: Use of basal.

Response: Corrected.

#30 Line 220: This lineage-specific event was also suggested in the 1KP paper.

Response: Yes, our result support *Acorales* experience the lineage-specific WGD event, which is consistent with the 1KP transcriptome phylogeny paper. added this paper citation.

The sentence has been revised as: L271-L273:

“These observations suggest that the detected ancient WGD event is lineage-specific to *Acorus*, this lineage-specific event was also suggested in the 1KP paper¹⁵.”

#31 Lines 227-228: Retaining genes in duplicate does not necessarily mean that they were maintained because of adaptation. Maintenance of paralogs is subjected to the diversity of the gene network of the genes in question, stochastic, and multiple molecular-level aspects involved in expression. It is very unclear what “difficult period of photosynthesis” would mean.

Response: We have deleted the confusing statement, and revised the sentence as “Simultaneously, KEGG enrichment analysis of WGD retained genes revealed that they are involved in photosynthesis, energy metabolism, and signaling pathways (Supplementary Fig. 14 and Supplementary Table 16).” L278-L281.

#32 Line 265: *Acorus* is on its own evolutionary trajectory and being sister to the rest of monocots would not be a reason as to why it has not seen major gene expansion. *Spirodela* and *Zostera*—as stated—have undergone major adaptive changes to live in aquatic environments. The comparison to rice has the added impact of the tau, sigma, and rho WGD events which would greatly impact the diversity of gene families in the species.

Response: Yes, you are correct. We have deleted the misleading statement.

**#33 Line 276: “strong metabolic ability” does not really mean anything.
Response: We have removed the sentence.**

**#34 Line 296: This is not a contraction. Rice has three WGD events in its lineage after the divergence from *Acorus* while banana has tau and three of its own. These are expansions in these taxa.
Response: We have deleted the wrong statement.**

**#35 Line 298: Harkess et al. 2017 identified that *Acorus* and Alismatales do not share tau with other monocots. This was confirmed in the 1KP paper.
Response: modified and added reference.
L366-L368: “Our study supported that *Acorales* and Alismatales did not experience the tau (τ) whole-genome duplication, but all other monocot clades experienced.”**

**#36 Line 304: “Less” not “lesser”.
Response: Corrected.**

**#37 Line 306: *Arabidopsis* has also undergone multiple WGD events. It is unclear why it is specifically compared to *Acorus* for MYB TFs.
Response: Thanks for pointing this out. We have realized that *Arabidopsis* is not a good candidate to make this comparison. We deleted the supplementary fig 20 and made corresponding changes in the revised version. Now the sentence reads as “MYB transcription factors are likewise critical in plant growth and development, however, a smaller number of MYB genes were identified in *Acorus gramineus* than most monocot species (Fig. 4d).” L374-L376.**

**#38 Line 308: Contracted seems less likely than expansion in other lineages.
Response: Modified L379-L382. The sentence now reads as “Although many gene-family and metabolic pathways showed few gene numbers in *Acorus gramineus*, but gene numbers of some pathway showed expansion, such as some energy metabolic pathway, photosynthesis and oxidative phosphorylation.”**

**#39 Line 313: This may indicate adaptation to lower light, but it really is not absolute.
Response: We have deleted the statement, and discussed the results in discussion section as “LHC genes play a critical role in photosynthesis in a dark, low-light environment¹⁰¹. Hence, the expansion of the LHC superfamily could be linked to *Acorus gramineus*’s natural wetland habitat mostly under the trees, where there is a shortage of sunshine and inadequate light, thereby necessitating the development of new light-capture mechanisms. This is in accordance with the findings made on two species of Alismatales, *Z. marina*⁷⁸ and *S. polyrhiza*⁷⁷.” L500-L506.**

**#40 Line 316: Remove “the” from “the nocturnal pollinators”.
Response: Removed.**

**#41 Line 319: “Dicot” is not an accurate term. Use eudicot and/or magnoliids.
Response: Corrected.**

#42 Lines 325-326: It is unclear how this section “highlight[s] the adaptive features of *A. gramineus* in fluctuating environments.” What features? There are some examples of genes involved in pathogen defense and volatiles, but the description is a bit hyperbolic as there is no link to the impact of these changes in gene number to an adaptive phenotype.
Response: We have deleted the wrong statement. We have also toned down the conclusion and revised the sentence at the end of the paragraph as “Overall, these findings likely suggest the adaptive features of *A. gramineus* in fluctuating environments. But there could be a myriad of other reasons related to expression and adaptation that should also be considered in future studies.” L429-L432.

#43 Line 333: Variation in perianth whorl number need not be a function of MADS-box TF number. It can just be an adaptation.
Response: Thanks for pointing this out. We deleted this sentence and only presented the fewer genes of MADS-box in *Acorus gramineus*.

#44 Line 344: CLEs are extraordinarily difficult to identify and require a tiered approach. Unless this was followed (e.g. <https://pubmed.ncbi.nlm.nih.gov/27911469/>), I would be cautious in claiming *Acorus* has only a single CLE.
Response: Thanks for providing this reference. As suggested, we re-run the analyses following this method and identified two CLE genes. Relevant method and results were updated.

Result: L419 two CLE could be identified in *Acorus gramineus*.

Method: L878-L890 “For CLE, we used homologous alignment and hmmersearch to identify CLE peptide of *Acorus gramineus*. Firstly, we downloaded CLE gene of *A. thaliana* and used it as a reference gene to compare with *Acorus. gramineus* protein using BLASTP (evaluate e-5) to obtain candidate genes. Secondly, we used the HMMER¹⁶³ search method, and followed two steps: 1) Building CLE HMM model used hmmbuild based on an alignment of CLE genes provided in Goad, 2017¹⁶⁹. 2) The constructed HMM model was used to identify *Acorus gramineus* CLE gene by hmmsearch, the result of hmmersearch was filtered by evaluate e-5. Both these steps used default parameters. Third, candidate CLE genes were obtained using the above two methods and were further used for signal peptide site identification using the online analysis tool SignalP (<https://services.healthtech.dtu.dk/service.php?SignalP-5.0>). Finally, these genes were identified by building a phylogenetic tree. The maximum likelihood trees were built using IQTREE (v2.0.5) after sequencing alignment in MAFFT (v7.471)^{148,159}.”.
The updated figure was presented in supplementary fig 22.

#45 Lines 351-352: “simple plant architecture” is not really a fair comparison. Simple compared to what? What does that mean exactly?
Response: This sentence is removed.

#46 Line 359: “support *Acorus* belonging to Acorales”
Response: Corrected, thanks.

#47 Line 376: *Acorus* only had one WGD event since the angiosperm WGD event. Better to clarify.

Response: Modified. The sentence has been revised as:

“The collinearity analysis and the Ks distribution of *Acorus* species suggest *Acorus* only underwent one lineage specific WGD event since the angiosperm WGD event.” L458-L459.

#48 Line 377: Near the K-Pg boundary is up to 20 million years after by the method of estimation.

Response: Deleted.

#49 Line 398: “We noticed” is not the right way to say this. Just say they thrive near streams.

Response: Deleted. The sentence has been revised as: L483-L485

“Most *Acorus gramineus* groups thrive near streams³⁰, and these plants are known to suffer from far fewer pests and diseases than plants on land that are exposed to the air⁷⁸.”.

#50 Line 410: Are marshes and brooks inherently low light? This comparison is missing something.

Response: The sentence has been revised as : L501-L504

“the expansion of the LHC superfamily could be linked to *A. gramineus*’s natural wetland habitat mostly under the trees, where there is a shortage of sunshine and inadequate light, thereby necessitating the development of new light-capture mechanisms.”.

#51 Lines 411-421: There is not necessarily a loss of WRKYs that are associated with living in standing water. This is a major stretch to compare to the transition onto land.

Response: Yes, we do agree with your comment that the smaller number of WRKYs in *Acorus* could not be related to the aquatic/wetland lifestyle. Therefore, we have toned down the statement and also revised other sentences as per your comments in the main text.

The sentence has been revised as: L501-L517: “the expansion of the LHC superfamily could be linked to *Acorus gramineus*’s natural wetland habitat mostly under the trees, where there is a shortage of sunshine and inadequate light, thereby necessitating the development of new light-capture mechanisms. This is in accordance with the findings made on two species of Alismatales, *Z. marina* and *S. polyrhiza*⁵. Hence, we hypothesized that the lineage of Acorales and Alismatales has a tendency to grow in the aquatic/wetland environment. A significant expansion in the number of several gene families has been reported during the terrestrialization (transition from water to land) in green plants⁶⁻¹⁰. In contrast, we discovered that some key terrestrial genes, such as WRKYs, displayed lower numbers in *Acorus gramineus* (which exhibits a transition from land to water). Thus, the high-quality genome of *Acorus gramineus* (the sister group of monocots), is of great significance to understanding the phylogenetic evolution of monocots and aquatic land plants.”.

Methods

#52 All programs should have the parameters used. The following do not: Trimmomatic, NextPolish, purge_dups, Minimap2, BLAST, all programs used to identify repeats, annotation programs, all gene characterization software, TransDecoder, CAF, and HMMER.

Response: Parameters selected for all the software and programs were listed in Supplementary Table 25.

#53 Line 437: What version of Alabacore?

Response: Guppy (v4.0.11) was used in the updated MinKNOW package. We have revised it accordingly.

#54 Line 443: Citation for the Hi-C method?

The reference was cited in the next sentence in the first submission. Now we moved to the first sentence.

#55 Line 443: “were” not “was” for the leaves

Response: Corrected.

#56 Line 459: k-mer not K-mer

Response: Corrected.

#57 Line 460: Remove “in the present study”

Response: Removed.

#58 Line 463: What is meant by “low-quality”. What threshold? Were adapters trimmed? How were PCR duplicates determined?

Response: Revised. PCR duplicates were determined by the program with parameter -p. The sentence now reads as “To minimize the sequencing error rate, a strict quality control was performed using SOAPfilter (v2.2) with default parameter setting (-q 33 -y -p -M 2 -f -1 -Q 7 -W 5 -B 25): filtering out reads with >5% of bases as missing “N”, reads that >25% of bases with ASCII(Q-score)-33<7, and PCR duplicates.” L561-L564.

#59 Line 479: blasted or BLASTed

Response: Corrected.

L586 “Following that, contigs were blasted using blast (v2.2.29).”.

#60 Lines 483-485: There is a portion of the text missing here.

Response: Corrected and added. L590-L598

“The assembly evaluations for the *Acorus gramineus* genomes are provided as follows: First, mapping of the 1,375 conserved core eukaryotic genes from the BUSCO data set (embryophyta_odb10, BUSCO v3.0.2), resulted in 94.8% (Supplementary Table 4) of the core eukaryote genes recovered for the majority of the genome assemblies; Second, BWA (v.2.21) was used to map the reads to the draft assemblies to evaluate the DNA reads mapping rate (>96.32 %) (Supplementary Table 1). Finally, we mapped the RNA reads to the draft assemblies to evaluate the RNA reads mapping rate using Hisat2

(Supplementary Table 1). Taken together, these results indicated good genome assembly qualities for this newly sequenced species.”

#61 Lines 490-491: Check capitalization on Juicer and 3D-DNA.

Response: Corrected.

#62 Line 578: What models were ultimately chosen for the phylogenetic analyses?

Response: Added models used in methods.

15 species phylogenetic tree: GTR+F+R5. L704

223 species phylogenetic tree: GTR+F+R10. L713

#63 Line 587: What fossils were used? These have a large impact on divergence time estimation.

Response: Added fossils times were used, listed in Supplementary Table 13.

#64 Line 712: The link does not work.

Response: Due to the time limit, it is invalid and has been re-applied and updated.

Reviewer #3 (Remarks to the Author):

In this manuscript, Guo et al. sequenced and assembled the genome of *Acorus gramineus*, a basal monocotyledon species whose classification and phylogenetic position had been subject to long-standing debate. Thanks to long-read (Nanopore) and short-read (Illumina) sequencing coupled to Hi-C, the authors were able to obtain a high-quality reference genome with an N50 of 36 Mbp. The analysis of the evolution of the *A. gramineus* genome and its comparison with the genomes of other sequenced plant species led to several key findings :

(i) Phylogenomic analysis confirmed that *A. gramineus* is a basal monocot species and that *Acorus* species can be divided into two main clades: *A. calamus* and *A. gramineus*.

(ii) Phylogenetic analysis of mitochondrial genes revealed multiple horizontal transfers from other monocot and dicot species which could explain the misplacement of this species in phylogenies using mitochondrial genes.

(iii) *A. gramineus* did not undergo the previously reported tau (τ) WGD (Whole Genome Duplication) event shared by all other monocot clades.

(iv) *A. gramineus* has very few genes compared to other monocots, including resistance genes and some transcription factors, suggesting gene contraction related to the ecology and lifestyle of this species.

I find the manuscript well written and very pleasant to read. The analyses conducted are globally well done. However I have some concerns about the lack of clarity sometimes regarding certain analyses and the interpretation of some results. Here are my comments:

#1 Figure 1. Morphology and genomic architecture of *A. gramineus*

I have some comment regarding the density of genes and LTRs across the *A. gramineus* genome:

For chr 2, chr8, chr9, chr10, chr11 and chr12 we can see that the distribution of genes and LTRs is reversed compared to what is “expected”. This even more true for the distribution of gypsy elements (Figure 1-b G) that are characteristic of pericentromeric regions and are in *A. gramineus* assembly mainly located in chromosome end (chr 2,8,10,11 and 12). We know that in plants pericentromeric regions are TE-rich especially LTRs retrotransposons while chromosome arms are genes-rich.

This gene-TEs density pattern inversion can be explained by :

1- Assembly errors (inversion of chromosome arms for instance) that will lead to a false chromosome structure.

2- Genomic events such as fissions/fusions, inversion, translocations and nested chromosome insertion that have occurred during the “recent” genome evolution of *A. gramineus*. Those chromosome rearrangement result generally in a “non-canonical” TE-genes density across chromosomes.

Response: Thanks for the nice suggestion. We carefully checked and found that it may be due to cause 1. Therefore, we re-run the Hi-C analysis, carefully and manually corrected the chromosome structure with Juicebox. The final results of genome assembly are updated (table1, Supplementary Table 3) and the gene distribution map has been redrawn (Fig 1b).

#2 In order to investigate all these possible scenarios, comparison of the *A. gramineus* genome with that of closely related species could provide a clear picture of recent genomic rearrangement events that may have occurred in *A. gramineus*. Since the authors do not have the genome of closely related species, an alternative analysis will consist of the characterization of centromeric and telomeric repeats. This will allow to determine precisely the position of centromeres and telomeres validating the structure of the assembled chromosomes but also could reveal the position of paleocentromeres and paleotelomeres. This will allow the authors to better understand these “inverted” density profiles on the 6 *A. gramineus* chromosomes. It is true that it is often difficult to properly assemble the centromeric and telomeric regions but with Nanopore sequencing coupled to Illumina and HiCi sequencing, these regions should normally be present in the assembly performed in this study. At the same time, this analysis will give additional guarantees on the quality of genome assembly.

Response: We are so sorry, these “inverted” density profiles on the 6 *A. gramineus* chromosomes may be due to our assembly error. Therefore, we re-run the Hi-C analysis,

and carefully and manually corrected the chromosome structure with Juicebox. The final results of genome assembly were updated (table1, Supplementary Table 3) and the gene distribution map has been redrawn. (Fig 1b). Then, we detected high density tandem repeat factors regions (More than 100 copies), this may be the positions of paleocentromeres and paleotelomeres. The triangle in the figure below points to areas of high tandem repeat factors, the table below lists the tandem repeats.

Chr	Start	Consensus	CopyNumber	Chr	Start	Consensus	CopyNumber
Chr 1	7	CCTGGG	2564.3	Chr7	3872998	AT	192
Chr 1	2351739	TTAGGG	108.1	Chr7	1887068	TA	150
Chr 2	1	T	890.5	Chr8	1	GCCTGG	2251.5
Chr 2	1203589	AT	224	Chr8	1195028	TA	111
Chr 2	3419502	TTAGG	3758.3	Chr9	19	AAACCC	3751.4
Chr 3	1	G	5836	Chr9	1870106	T	107
Chr 3	1219724	CCTAAA	232	Chr9	5	A	4142.9
Chr 4	1	AT	5524.5	Chr9	3627505	TTAGG	4142.9
Chr 4	1020655	CCTGGG	126.5	Chr1	1	G	2850.9
Chr 5	1	TA	5423.4	Chr1	57274	GTTTAG	2850.9
Chr 5	1634793	CCTAAA	3696.2	Chr1	0	G	105
Chr 6	1	GGCCCA	4413.6	Chr1	2532728	TA	105
		AACCCT		Chr1	0	9	5429.1
		A		Chr1	3599155	TTAGGG	5429.1
				Chr1	0	T	4239.7
				Chr1	228128	ACCCTA	4239.7
				Chr1	1	A	274
				Chr1	3066744	AT	274
				Chr1	1	7	

Chr	1660643	TA	120	Chr1	4392246	AT	109.5
6	1			1	2		
Chr	2416108	GGCCCA	5683.7	Chr1	1	CCTAAA	6491.1
6	9			2		C	
Chr	107423	CCTGGG	2126.5	Chr1	1789683	AT	131
7				2	6		
				Chr1	3865673	AGGGTT	4315
				2	7	T	

#3 Page 5 : 132-136 “Long terminal repeat (LTR) retrotransposons were the most prevalent type of transposable element (TE) among these repeats, representing nearly 35% of the genome, including 23.1% LTR/Gypsy and 5.9% LTR/Copia retro-elements (Supplementary Tables 7).”

A description of the different TE families found and their relative frequencies in the genome is missing. Although LTRs are expected to represent the largest fraction of these elements, it is important in a paper describing the genome of a new species to annotate all TE families including Class II elements such as CACTA, hAT, MuDR or Helitrons. Finally it will be appreciated and usefull that the authors provide the GFF file and fasta file of both genes and TEs annotation.

Response: The updated statistics including more subtypes are provided in Supplementary Tables 7. Moreover, the GFF file of both genes and TEs annotation were submitted to CNSA database together with the genome assembly file.

#4 Page 5 : 139-141 : “To investigate the taxonomic status of Acorales, we first selected 563 sigle-copy ortholog sets (SCG) from four eudicots, four monocots, three Magnoliids, two basal Angiosperms, and Ceratophyllum demersum.”

The authors should clarify why they chose these particular species. Is it because they are "representative"? Because of their position in the phylogenetic tree?

Response: We selected representative species with higher genome quality to cover each lineage. Species information are listed in Supplementary Tables 8.

#5 Page 5: line 148-149 : “Both the phylogenetic trees based on concatenated sequence alignments showed similar topological structures, placing Acorales at the most basal position of the monocot lineage”

Do the authors mean "both phylogenetic trees"? The previous one using 14 species and the one with 223 species? or the two phylogenetic trees based on the concatenated sequence alignment and the coalescent approach provided the same results for the 223 species? The authors should clarify in the manuscript and or in the method section. In this regard I have not been able to understand which of the phylogenetic trees were obtained with a contacted alignment or using the colaescent model approach. Second, what is the difference between the two phylogenetic trees. The one with 226 species should be more informative because it includes the species used in the first phylogenetic tree

(with 14 species) and uses more orthologous genes. What is the rationale for using two sets of single-copy orthologous genes? Do they overlap? Do the 612 genes include the 563 genes plus some duplicated genes?

The authors should clarify these points

Response: Yes, this refers to phylogenetic trees built using both 15 species and 223 species. In fact, both datasets were used to construct the phylogenetic trees using concatenation and coalescent methods, and both results supported Acorales as the sister lineage to all other extant monocot plants. Fig2a shows the results of phylogenetic trees of 15 species using concatenation method (we updated fig2a legend: The phylogenetic tree constructed by IQTREE based on a concatenated dataset of used 562 single-copy genes, with a bootstrap value of 100 for all branches.), Fig2b and Supplementary Fig. 5 shows the results of phylogenetic trees of 223 species used concatenation method. Meanwhile, the phylogenetic tree used coalescent methods shown in Supplementary Fig. 4 and 6.

The reason for using two sets of data for phylogenetic reconstruction is to test the robustness of resultant phylogenetic relationships. 562 genes used in the 15 species dataset are all single-copy genes. To avoid bias of limited species sampling, we made a 223 species sampling. Single copy gene number would largely drop when selecting a large number of species, and hence we allowed a few low copy genes in the dataset for phylogenetic reconstruction.

The method part 693-L713 has been modified accordingly.

With the 562 single-copy gene families, we inferred the phylogenetic placements of *Acorus gramineus* according to the following method: For each single-copy gene orthogroup, we first performed multiple amino acid sequence alignment by MAFFT (v.7.471)¹⁴⁵, and then DNA sequences were aligned according to the corresponding amino acid alignments using PAL2NAL (v14.1)¹⁴⁶, followed by gap position removal using trimal (v1.4.1)¹⁴⁷, then each gene tree was constructed by IQTREE (v2.0.5)¹⁴⁸ that automatically selected the best-fit substitution model using ModelFinder¹⁴⁹. After this, the whole gene trees were then utilized by ASTRAL (v.5.6.1) to infer species trees with quartet scores and posterior probabilities (coalescent method), and the concatenated super-genes alignments were used for IQTREE (v2.0.5)¹⁴⁸ (concatenation method), the best model was selected by IQTREE (v2.0.5)¹⁴⁸ is GTR+F+R5.

In addition, we built a phylogenetic tree from an extended species sampling from 223 species in total, include 221 angiosperms species, and two gymnosperms as outgroups (Supplementary table 8c). The amino acid sequences from the whole species using BLASTP with an e-value of 1e-5, and then grouped using OrthoFinder (v 2.3.7)¹⁵⁰. Here, each gene was required to include sequences from more than 160 species for low-copy genes, which resulted in 612 genes. As same as above, which was constructed phylogenetic tree using concatenation and coalescent methods as described above from the 562 single-copy gene families. For concatenation method, the best model was selected by IQTREE¹⁴⁸ (v2.0.5) is GTR+F+R10.

#6 Page 6: Line 174-176: In order to assess the possibility of mitochondrial gene transfer occurred in the mitogenome, we constructed the phylogenetic trees together with the

mitochondrial genes of *A. gramineus*.

If I understand well the identification of horizontal gene transfer was based solely on phylogenetic incongruence of 18 mitochondrial genes among a total of 38 genes. Phylogenetic incongruence alone is not a strong argument in favor of HGT. Can be also due to other evolutionary and technical biases. It is important to consider also the high similarity criteria to ensure that the observed similarity of the transferred gene is higher than that of orthologous mitochondrial genes. Additionally it is important also to show that the transferred mitochondrial genes are not in syntenic/collinear position. In other words, if a horizontal transfer occurs that chance that the insertion of the foreign genes occurs exactly at the same position as the ancestral homologous genes is extremely low. Response: Thanks for your suggestion, we added gene alignment in Supplementary Fig. 10.

We re-analyzed the mitochondrial horizontal gene transfer events, followed these steps: 1) removed RNA editing sites. 2) reconstructed single gene trees to identify transferred genes. 3) check the alignment. 4) finally, to show that the transferred mitochondrial genes are not in a syntenic/collinear position, we compared *Acorus* mitochondrial genes to query species mitochondrial genome, the result is shown in the table below. But there are two problems: a. Gene sequences vary greatly between species. b. *Ceratophyllum demersum* doesn't have a complete mitochondrial genome, so the *ccmFN* and *matR* cannot be identified.

All methods and results were updated.

All 38 mt gene trees show the bootstrap value listed in Supplementary Fig 9.

The genes that have been identified as transferred are shown in the alignment with transferred lineages. Supplementary Fig 10.

Method: L796-L800 “For mitochondrial genes, to rule out the influence of RNA editing sites, we excluded all 1,537 RNA editing sites identified in the mitochondrial genomes of the angiosperm ordinal representatives in our individual gene alignment to alleviate the negative impact of RNA editing in phylogenomics analyses¹⁵⁸.”

L808-L819: “To identify horizontal gene transfer events (HGT), 38 single mitochondrial gene trees were concatenated. All the genes were aligned individually using MAFFT (v 7.471)¹⁵⁹ to build nucleotide alignments. Poorly aligned regions were trimmed by trimAl (v 1.4.1)¹⁶⁰, IQTREE¹⁴⁸ was used to infer a concatenation-based species tree with 1000 ultrafast bootstrap. We evaluated horizontal gene transfer events (HGT) based on phylogenetic topology and node supports. Specifically, we first filtered those gene trees (as ‘unclassified’ in supplementary fig. 9), in which monocots, eudicots, magnoliids and Chloranthales are non-monophyletic or ANA grade is not at the basal place. Second, we consider those gene trees as candidates to evaluate HGT if the unexpected placement of *Acorus gramineus* was observed i.e. *Acorus gramineus* is closely related to other lineages rather than as the sister to the rest monocots. Importantly, only cases with node bootstrap support $\geq 75\%$ would be inferred as HGT events.”

Q_Before_Gene_Chr	Q_Before_Gene	Q_Gene_Chr	Q_Gene	Q_After_Gene_Chr	Q_After_Gene	T_Before_Gene_Chr	T_Before_Gene	T_Gene_Chr	T_Gene	T_After_Gene_Chr	T_After_Gene
NC_017840	rpl10	NC_017840	rpl5	NC_017840	cob	8	atp4	8	rpl5	8	rps14

NC_0178 40	rps2	NC_01 7840	atp 1	NC_017 840	rps3	3	nad7	3	atp 1		
NC_0178 40	nad1	NC_01 7840	cc mB	NC_017 840	ccmFc	5	cox2	5	cc mB	5	nad1
NC_0178 40	rps1	NC_01 7840	nad 4	NC_017 840	sdh4	15	rps1	15	nad 4		
NC_0167 40	nad6	NC_01 6740	rpl2	NC_016 740	rps19			14	rpl 2	14	rps4
NC_0307 53	rps13	NC_03 0753	cox 2	NC_030 753	rps4			5	cox 2	5	ccmB
KC82196 9	rps11	KC821 969	cc mF C	KC8219 69	cob			12	cc mF C	12	nad5
KC82196 9	rpl5	KC821 969	nad 5	KC8219 69	nad3	12	ccmFC	12	nad 5		
KC82196 9	ccmFC	KC821 969	cob	KC8219 69	rps14	8	rps14	8	cob	8	rpl10
KC82196 9	ccmC	KC821 969	rps 4	KC8219 69	rpl10	14	rpl2	14	rps 4	14	rps10
		Xue20 18_m3 01	cc mF N					1	cc mF N	1	rps2
		Xue20 18_m3 01	mat R					4	mat R	4	ccmC

#7 Page 6 : Line 179-180 : “The transferred genes were mainly from monocotyledonous and dicotyledonous plants.”

Are the authors sure of the direction of the HGT. It is possible that some or all of these HGTs occurred from monocots and dicots to *A. gramineus*. How could the authors determine the direction of HGT and how could they reject the possibility that these transfers occurred in the other direction

Response: Thanks for pointing out his problem. We agree with you that it’s hard to determine the direction of HGT. In the revised version we deleted the direction arrows in the figure and revised the relevant description.

#8 Page 7 : 194-196 : The synteny analysis of *A. gramineus* with *A. trichopoda* showed 565 syntenic blocks that covered 16.0% and 14.7% of the assembled genomes, respectively.

Do you mean 16.0 % and 14.7 % of total genes ?

It is not clear to me if the authors have taken into account the intergenic regions in the calculation of the percentage of syntenic regions. It is obvious that intergenic sequences, and in particular TEs, are very dynamic and evolve rapidly so that this measure does not give a clear idea of the percentage of syntenic genes

Response: Yes, the statistics show the proportion of genes in collinearity, in consideration of the intergenic sequences, we updated the statistics to the block percent of the genome. And we download the chromosome level genome of *Amborella trichopoda* to re-run the synteny between *Acorus gramineus* and *Amborella trichopoda*. The updated result is shown in Supplementary Table 9 and Fig3c, meanwhile, the result part of main text is also modified.

L243-L245: The synteny analysis of *Acorus gramineus* with *Amborella trichopoda* showed 295 syntenic blocks that covered 24.6% and 13.1% of the assembled genomes, respectively (Supplementary Table 12).

#9 Page 7 : Line 204-207 : “To more precisely infer the timing of the WGD in the *A. gramineus* genome, intragenomic and interspecies homologue Ks (synonymous substitutions per synonymous site) distributions were estimated”

As far as I understand, the identification of horizontal gene transfer here was based solely on the phylogenetic incongruence of 18 mitochondrial genes out of a total of 38 genes. Phylogenetic incongruence alone is not a strong argument for HGT. It may also be due to other biases such as incomplete lineage sorting and hidden paralogy. It is important to also consider high similarity criteria to ensure that the observed similarity of the transferred gene is higher than that of the orthologous mitochondrial genes. In addition, it is also important to show that the transferred mitochondrial genes are not in a synthetic/colinear position. In other words, if a horizontal transfer occurs, the probability that the insertion of the foreign genes occurs at exactly the same position as the ancestral orthologous genes is extremely low.

Response: Thank you for your questions and suggestions. We re-analyzed the mitochondrial horizontal gene transfer events, followed these steps: 1) removed RNA editing sites. 2) reconstructed single gene trees to identify transferred genes. 3) check the alignment. 4) finally, to show that the transferred mitochondrial genes are not in a synthetic/colinear position, we compared *Acorus* mitochondria genes to query species mitochondria genome, the result is shown in the table below. But there are two problems: a. Gene sequences vary greatly between species. b. *Ceratophyllum demersum* doesn't have a complete mitochondrial genome, so the *ccmFN* and *matR* cannot be identified.

All methods and results were updated.

All 38 mt gene trees show the bootstrap value listed in Supplementary Fig 9.

The genes that have been identified as transferred are shown in the alignment with transferred lineages. Supplementary Fig 10.

Method: L796-L800 “For mitochondria genes, to rule out the influence of RNA editing sites, we excluded all 1,537 RNA editing sites identified in the mitochondrial genomes of the angiosperm ordinal representatives in our individual gene alignment to alleviate the negative impact of RNA editing in phylogenomics analyses¹⁵⁸.”

L808-L819: “To identified horizontal gene transfer events (HGT), 38 single mitochondria gene trees were concatenated. All the genes were aligned individually using MAFFT (v 7.471)¹⁵⁹ to build nucleotide alignments. Poorly aligned regions were trimmed by trimAl (v 1.4.1)¹⁶⁰, IQTREE¹⁴⁸ was used to infer a concatenation-based species tree with 1000 ultrafast bootstrap. We evaluated horizontal gene transfer events (HGT) based on phylogenetic topology and node supports. Specifically, we first filtered those gene trees (as ‘unclassified’ in supplementary fig. 9), in which monocots, eudicots, magnoliids and Chloranthales are non-monophyletic or ANA grade is not at the basal place. Second, we consider those gene trees as candidates to evaluate HGT if the unexpected placement of *Acorus gramineus* was observed i.e. *Acorus gramineus* is closely related to other lineages rather than as the sister to the rest monocots. Importantly, only cases with node bootstrap support $\geq 75\%$ would be inferred as HGT events.”

Q_Before_Gene_Chr	Q_Before_Gene	Q_Gene_Chr	Q_Gene	Q_After_Gene_Chr	Q_After_Gene	T_Before_Gene_Chr	T_Before_Gene	T_Gene_Chr	T_Gene	T_After_Gene_Chr	T_After_Gene
NC_017840	rpl10	NC_017840	rpl5	NC_017840	cob	8	atp4	8	rpl5	8	rps14
NC_017840	rps2	NC_017840	atp1	NC_017840	rps3	3	nad7	3	atp1		
NC_017840	nad1	NC_017840	ccmB	NC_017840	ccmFc	5	cox2	5	ccmB	5	nad1
NC_017840	rps1	NC_017840	nad4	NC_017840	sdh4	15	rps1	15	nad4		
NC_016740	nad6	NC_016740	rpl2	NC_016740	rps19			14	rpl2	14	rps4
NC_030753	rps13	NC_030753	cox2	NC_030753	rps4			5	cox2	5	ccmB
KC821969	rps11	KC821969	ccmF	KC821969	cob			12	ccmF	12	nad5
KC821969	rpl5	KC821969	nad5	KC821969	nad3	12	ccmFC	12	nad5		
KC821969	ccmFC	KC821969	cob	KC821969	rps14	8	rps14	8	cob	8	rpl10
KC821969	ccmC	KC821969	rps4	KC821969	rpl10	14	rpl2	14	rps4	14	rps10
		Xue2018_m301	ccmF					1	ccmF	1	rps2
		Xue2018_m301	matR					4	matR	4	ccmC

#10 Figure 3 : please correct “c. The intragenomic Ks distribution Karyotype evolution”. Do you mean b part of the figure.

Response: Yes, Corrected.

#11 Page 8 : Line 221 : “Following the WGD event, *A. gramineus* retained a total of 7,902 genes”

Do you mean paralogous pairs ?

Response: It shows the paralog genes conserved by WGD. The sentence is revised as “Following the WGD event, *A. gramineus* retained a total of 7,902 paralog genes”.

#12 Page8-9: “However, in the present study, there were 672 specific gene families in *A. gramineus*, lower than *S.polyrhiza* and *Z. marina*. Compared to 4,960 specific gene families in *O. sativa*, the number of specific gene families in *A. gramineus* comparatively fewer. This suggests that being a basal monocots species, *A. gramineus* did not experience a dramatic genome expansion.”

As the authors show, *A. gramineus* underwent only one WGD event, unlike other Monocot plants. Furthermore, we know that following WGD, paralogous genes could rapidly diverge, allowing the emergence of new specific genes. The question I have is the following: Is this low number of specific genes observed in *A. gramineus* compared to rice for example is because of the low number of duplications experienced by this species?

Response: We think this was a likely scenario. As it was hard to trace whether the high number of specific gene families in *O. sativa* was due to divergence of paralogous genes following WGD, we did not discuss the likely associations between the low number of specific genes and the lack of Tau whole-genome duplication in *A. gramineus*.

#13 Page 14 : “Line 402 addition, the R-genes also showed contraction in gene numbers, particularly the NBS genes.”

What makes the authors say that this is a contraction of the gene number? We know that R-genes evolve by duplication such as tandem duplications. It is impossible to know the number of R genes in the ancestors of monocotyledons on the basis of the sequenced modern Monocots genomes since these genes are subject to a significant rate of duplication during evolution even at the level of populations. It seems to me therefore difficult to draw the conclusion the relatively low number of R-genes *A. gramineus* compared to other Monocot species tha is due to a contraction of R-genes. This comment also applies to the MADS-box gene and to other gene families in which the expansion is often explained by intensive local duplication.

Response: The writing was revised to change to the *A. gramineus* genome without great genetic expansion. Thanks for the suggestion. Instead of saying contraction we have toned down the text, and described the actual result as “From our pathway analysis, we discovered that the pathways with lower gene numbers were likely related to stress resistance and disease resistance. In addition, the R-genes showed lower gene numbers, particularly the NBS genes.” L485-L488.

Other comments

#14 There is no mention of Figure 1 in the manuscript.

Response: Cited.

Reviewers' Comments:

Reviewer #1:

Remarks to the Author:

On one point, I find the manuscript considerably improved, and that is with regards to description of applied methods. However, I believe that there are still numerous issues that need to be fixed.

In general, the language is often clumsy, sometimes making it difficult to understand exactly what the authors mean.

The manuscript tends to include a number of claims, which are not supported by the data and only superficially discussed. Some discussion is initiated in the results section and the discussion section only rarely extends beyond a summary. Below, I have noted a number of cases of discussion-like text in the results section. Rather than moving them to the discussion it may be a better choice simply to combine results/discussion into one section.

Using the term "basal" about taxa is problematic. The authors say that have corrected it, but there are still several occurrences of "basal" (basal monocots, basal angiosperms, etc..) both in the main text, but also in the supplementary material, e.g. figs. 5, 8

I am highly sceptical about the inference of HGT. Inferring events of HGT from gene trees can be difficult, especially with regards to relatively closely related taxa (e.g. Alismatales and Acorales). There may be many other reasons for apparently wrong placements and I doubt that the authors would try to explain the myriad of misplaced taxa in their mitochondrial gene trees solely as caused by HGT. Why is it that e.g., incomplete lineage sorting does not explain the apparently misplaced taxa in the gene trees?

Some specific notes to the trees:

atp1: Acorus is placed as sister to Alismatales and that entire clade appears misplaced. HGT from Alismatales to Acorus would not explain the placement.

ccmB: you could equally well infer this as HGT from Acorus to Spirodela.

ccmFN: a sister group placement of Ceratophyllum and Acorus does not necessarily mean HGT from Ceratophyllum to Acorus. Several positions in this tree are weird and cannot just be explained by one event of HGT.

cob: the support is not overwhelming. It just meets your criteria. Normally 75% bootstrap is considered moderate support.

nad4: with Acorus as sister to just some of the alismatids and the Alismatales being non-monophyletic (how would you explain the position of Spirodela) a HGT hypothesis is not straight forward.

rpl12: the tree is not shown

Below I list a number of issues (some minor, some major):

L34: "Interestingly, we discovered....". No, you confirmed what was previously found.

L35: "... likely helped Acorus to maintain their ancestral state....". This really says nothing. Delete.

L37-39: I suggest to be more suggestive. E.g. "... gains possibly associated with ecological"

L39: gene families instead of just genes?

L52: "morphological discrepancies". Better: "morphological variability"?

L55: "problem". Better: "challenge"?

L66-67: Better: "... the placement of Liliales relative to Asparagales and the sister group relationship of Poales"

L68-69: Is it fair still to say that the placement of Acorus is ambiguous? All other than mitochondrial data have in recent large studies consistently placed Acorus as sister to the other monocots.

L85-89: Too much detail. What ethnic groups believe or not is irrelevant. Just write that you prefer to recognize all five species.

L99: so far you have used monocots and not monocotyledons

L89-89: you should add the large transcriptome/genome based analyses (Harkess et al, 1KP ...)
L103: what do you mean by gaining popularity?
L100-105: Since you only attempt to investigate HGT of mitochondrial genes, it would be more relevant to cite papers about this phenomenon in angiosperms.
L108: the phylogenetic relationships of monocots, except from the placement of Acorus, is not reported on at all in the manuscript (unless I missed something)
L109: here the taxon Acorales is mentioned for the first time. It should be introduced above that Acoraceae is the only member of the order.
L110: basal!

L145: basal!
L150: Ceratophyllum should be in italics
L150-151: no! two sister clades differentiate at the same time!
L155-158: "It should be noted...." This belongs to the discussion.
L159-176: This is a mix of introduction, results and discussion.
L168-169: correct: that is morphologically distinct from the rest of the species
L169: "The results....". Where are the described results/tree?
L174-176: clumsy writing. Rephrase
L179-181: incomplete sentence
L185: Here I would expect some general results from assembly of the mitochondrial genome.
L186: rewrite "... constructed phylogenetic trees using...."
L188: rewrite: "... we inferred that 12 ..."
L190-191: Alismatales are also monocots. Perhaps say other monocots? The figure specifies Arecales! Make text and figure consistent.
L191-193: "We observed...." Are you referring to your own concatenated analysis of mitochondrial genes? Where is this result shown?
L193-196: 12 out of 38 genes are not most. The remarks about Amborella make no sense. Are you referring to you own data/trees or the previous publication about HGT. And what has it got to do with the taxonomic position. If this should be included at all it belongs to the discussion.
L214: monocot not monocots
L216: did not instead of didn't
L221-223: this is a nothing but a wild guess. If included at all, place it in the discussion.
L241: basal!
L246: "... four monocots species...": monocot not monocots. Do you mean 10 monocot species? Otherwise I do not understand what is meant.
L247: include "and" before Oryza
L248-254: here again you approach something that would fit a discussion, although some elaboration would be desirable
L257: delete "According to" to correct sentence
L260: Add "According to" in the beginning of this sentence
L282-283: The sentence makes no sense. Rewrite
L284: that instead of but
L289: monocot not monocots
L293: gene families not gene-family; "...showed few gene numbers in Acorus gramineus, but gene numbers of some pathways showed" change to "... were contracted in Acorus gramineus, others showed...."
L302-305: the end of this sentence makes no sense
L320: ".. of monocotyledons is ..." change to "...being...."
L320: no such group as dicotyledons. Do you mean eudicots or anything else than monocots?
L333-336. In what way is the architecture of Acorus simple compared to grasses?
L336-337: where on earth did this claim come from? I do not see anything "likely" here!
L336-338: this belong to the discussion, if anywhere at all.

L343: what do you mean by "filling the key evolutionary gap" ?

L345-346: which "gene regions" (not mitochondrial)? First you say that *Acorus* has been placed in different positions, then you only mention placement as sister to the rest of the monocots. Thus, the sentence does not quite make sense. Write *Acorus* in italics.

L348: delete "A"

L349: delete "A"

L352-355: This is circular reasoning. Yes, HGT from Alismatales would explain why mitochondrial genes trees place *Acorus* in Alismatales, but to when HGT has been inferred from those gene trees the reasoning becomes circular.

L358-360: Not clear what is meant here? Which species? Maybe use names to clarify.

L363-366: wrong grammar in this sentence

L370-373: but if *Acorus* and alismatids had other WGDs wouldn't they still be expected to a high number of TFs?

L377: *Acorus* is not "early".

L378-383: How are any of these structures "simple" compared to e.g. *Oryza*?

L384: "grows" rather than "groups"?

L393: "dicots": eudicots or anything else than monocots?

L401: use full names. Does *Spirodela* grow in shady habitats? That is not my impression.

L402: The habitat preference of *Acorus* and alismatids is not a hypothesis.

L406: basal!

L407: "a transition from land to water" in what way? Are you implying something evolutionary or do you just mean that *Acorus* grows in a transition zone?

L423: ground not grounded

L434-435: cite table S8b for voucher info

L452: close not closed

L460: lacking right parenthesis

L461-463: something is wrong with this sentence. Rewrite

L493: similarity-based instead of homology-based?

L505: maximum not max

L528-533. Long sentence. Split in two.

L560-566. Long sentence. Split in two.

L566: "all gene trees" instead of "the whole gene trees"

L573: all species not the whole species

L576-578: something is wrong with this sentence. Rewrite

L583: all species not the whole species. Some word(s) is also missing in the sentence

L601: parameters not paraments

L621: Heading: this section is also about assembly and about the mitochondrial genome

L639: genomes not plants, mitochondrial not mitochondria

L651: mitochondrial not mitochondria

L651-652: "...trees were concatenated." Do you mean alignments?

L652-653. Superfluous? Isn't that described above?

L651-662: unclear. How do you use the tree from the concatenated data set? Don't you use the individual gene trees to infer HGT?

L700: *Arabidopsis*

L701: no punctuation after *Acorus*

L705: sequence not sequencing

L707, 713, 714: hmmsearch or hmmersearch?

L709: no punctuation after *Acorus*. Add italics

L713: genes not gene

L719: sequence not sequencing

L764: Explain the asterisk/star in 2b (or delete it from figure)

L767: No arrowheads in figure!

L774: Explain the abbreviated taxon names used in 3b.

L776: basal!

I have not read all supplementary files carefully, but here are some corrections:

Suppl. Fig 4. Explain the colours of quartet scores

Suppl. Fig. 9 text: correct "Mononcots" to "Monocots". Chloranthales mentioned twice. Change one to Ceratophyllum (or Ceratophyllales)

Suppl. Fig. 11. Write names in full (Amborella trichopoda, Acorus gramineus)

Suppl. Fig 12. The main legend just mentions Acorus, but the figure includes other taxa. Write names in full.

Suppl. Figs 15, 18, 20, 21, 22, 24. Write names in full

Reviewer #2:

Remarks to the Author:

Guo et al. present a resubmission of the Acorus gramineus genome and accompanying analyses focused on identifying HGT and investigating genes associated with potential environmental adaptation. Though the genome of Acorus represents a valuable position in the angiosperm phylogeny to have a complete genome, the overall findings of the paper seem to be confirmatory to other papers. The claims of HGT in the mitochondrial genome were easier to assess given the trees and alignments, but there are issues in those analyses.

The evidence used to suggest horizontal transfer of genes in the mitochondrial genomes leaves me with more questions than answers. The mitochondrial genome of land plants, especially angiosperms, is known to act as a DNA sponge and incorporate various other genome components into them. For example, the mitochondrial genome of Amborella has been demonstrated to have a sizeable portion of chloroplast genomes from other species in it. Looking at the alignments of the different mitochondrial genes presented in the supplemental figures makes me question the validity of the function of these genes. Each of the Acorus genes looks extremely different than the other taxa in their alignments, often being shorter than the other genes. This leads me to ask if these genes are translatable to a fully functioning protein. Are these genes indeed the result of HGT but there still exists a copy in the mitochondrial genome that is derived from vertical transmission? There is also the question of a few SNPs being used to suggest HGT when these could be the result of homoplasy.

The methods say that poorly aligned regions were trimmed using trimal, but the alignments presented in the supplemental figures show large regions that are not aligned well. There seems to be something going on with these alignments that is questionable for further phylogenetic analysis.

Other areas with questions/comments

Abstract

Repetitive with "discovered". Again, this paper confirms and not "discovers" the absence of tau in acorus history.

Line 52: "Discrepancy" is the wrong word. I think you mean "diversity".

Line 53: "Uphold" does not seem like the right word. "Provide" or a word like it is probably better.

Lines 66-67: Is there something missing here? Is this saying the uncertainty is in what clade is sister

to the Poales? If so, something like “problems remain in the relative placement of Liliales to Asparagales and identification of the lineage sister to the Poales” would be a better way of saying this.

Line 69: The placement of *Acorus*/Acoraceae/Acorales—which is all equivalent—has been established for quite some time. It is true *Acorus* was once included in Araceae, but molecular phylogenetics has placed it sister to the rest of the monocots for many years. This was a critique I had in the previous version of this manuscript. This statement seems to be overstating the uncertainty of *Acorus* placement in the phylogeny.

Lines 90-112: This section is completely missing transcriptomic data that has been used to look at the phylogeny of this group. This is not a complete and accurate depiction of our understanding of the evolutionary placement of *Acorus*. The representation is such to make this seem more questionable in our understanding.

Line 119: Missing “and” to link the description of the Nanopore and short read data. Currently the text just separates with a comma.

Line 122: Hi-C data is more helpful if given in total read pairs instead of total base pairs.

Line 150: Monocots and the rest of the angiosperms diverged from each other at the same time, based on the shared common ancestor. It is better to say that monocots diverged prior to the diversification of eudicot, magnoliids, and Ceratophyllum. The term “basal angiosperms” is not a correct term since this is referring to the members of the ANA grade.

Lines 174-176: Language of this sentence needs to be fixed to improve the grammar.

Lines 177-195: Language throughout this section needs to be check for clarity and grammar.

Lines 220-222: I believe this sentence is trying to invoke the idea that there are several WGD events found around the timing of the K-Pg boundary suggesting these events are related to post K-Pg success. Given the 20 my between the event and the K-Pg, I would say this is too spurious to make a connection.

Lines 246-252: The language here is unclear. Are these meant to be gene families that are only found in these taxa? What does that mean exactly? These families are likely related to some other families—this seems like a lot of genes to be de novo in these taxa. Is this a case of gene family circumscription being too conservative?

Reviewer #3:

Remarks to the Author:

In the revised manuscript, the authors have corrected and addressed some of the raised concerns. I had two major issues with the manuscript: (i) The possible misassembly of the genome and (ii) weak arguments supporting the horizontal gene transfer (HTG) hypothesis. I am completely satisfied by the authors' response to the first concern but not to the second.

1- The first concern was about the distribution of genes/TEs along the chromosomes which in the previous manuscript was atypical and did not follow what is expected in plant genomes. This first issue was due to genome misassembly which has now been corrected by the authors through a new Hi-C experiment and genome reassembly.

2- The second issue had to do with horizontal transfers. I pointed out in the previous version of the

manuscript that HGTs were not well supported. I am still not convinced that the observed phylogenetic incongruities of some mitochondrial genes are indeed due to HGTs.

The number of conserved sites in the multiple alignments provided that lead to phylogenetic incongruities is very small. For example: nad5 (2/1000 sites), rps4 (2/1400 sites), ccmFC (3/1500 sites), ccmFN (4/2200)...etc. What are the probabilities that these mutations occur by chance between highly divergent lineages (tens of millions of evolution)? The answer to this question is essential to determine whether these genes were acquired by horizontal gene transfer or not. For this reason, I asked the authors to compare the sequence similarity (Ks) of orthologous mitochondrial genes (for species with sequenced mitochondrial genomes) of the species involved in HGT with that of the transferred genes. If the rate of synonymous mutations is significantly lower for the transferred genes, this constitutes a strong argument for HGT together with the phylogenetic incongruence.

Minor but important points

Legend for Figure 1: Do you mean relative density or number? In any case, the authors should add a key for each track.

Supplementary Table 7a:

- DNA vs. RNA TEs: Please correct Class I and Class II
- The word "Number" should be deleted because the authors do not provide numbers but repeat Size and percentage (Table 7a)

Supplementary Table 7b:

The authors should simplify the classification system. For example, for LTRs-retrotransposons, you have Copia, Gypsy and LTRs. Copia and Gypsy are LTRs. There are many problems with the TE superfamilies provided. The other categories of LTRs are generally not found in plant genomes (e.g. Bel-Pao). In addition, there are many redundant superfamily classes in the table: Caulimovirus and Caulimoviru, Gypsy and Gypsy-Cigr. This is also the case for LINEs and Class II elements (TE DNA). Authors should refer to the widely used Wicker TE classification system for eukaryotic genomes (<https://trep-db.uzh.ch/TEClassification.php>).

There are still many typos in the figure captions. I will not list them all, but for example, in the caption of Supplementary Figure 10.

REVIEWER COMMENTS

Reviewer #1 (Remarks to the Author):

On one point, I find the manuscript considerably improved, and that is with regards to description of applied methods. However, I believe that there are still numerous issues that need to be fixed.

#1 In general, the language is often clumsy, sometimes making it difficult to understand exactly what the authors mean.

Response: We have carefully revised the language throughout the manuscript with the help of a native English speaker and rephrased the ambiguous sentences to make the text easy to understand.

#2 The manuscript tends to include a number of claims, which are not supported by the data and only superficially discussed. Some discussion is initiated in the results section and the discussion section only rarely extends beyond a summary. Below, I have noted a number of cases of discussion-like text in the results section. Rather than moving them to the discussion it may be a better choice simply to combine results/discussion into one section.

Response: Thanks for pointing out the issues. The manuscript has been revised, and the claims made in the study are toned down accordingly. For the cases of discussion-like text in the results section, we have moved them to discussion section. We have rewritten the results and discussion sections with extension to likely cause of mitochondrial misplacement and wetland adaptation. However, with respect to combining "Results and Discussion", we still retained our previous version of a separate results and discussion section according to the journal format.

#3 Using the term "basal" about taxa is problematic. The authors say that have corrected it, but there are still several occurrences of "basal" (basal monocots, basal angiosperms, etc..) both in the main text, but also in the supplementary material, e.g. figs. 5, 8

Response: Apologies for the mistake. We have carefully crosschecked, and removed the usage of basal throughout the text. All the term "basal angiosperms" has been revised to "ANA grade".

#4 I am highly sceptical about the inference of HGT. Inferring events of HGT from gene trees can be difficult, especially with regards to relatively closely related taxa (e.g. Alismatales and Acorales). There may be many other reasons for apparently wrong placements and I doubt that the authors would try to explain the myriad of misplaced taxa in their mitochondrial gene trees solely as caused by HGT. Why is it that e.g., incomplete lineage sorting does not explain the apparently misplaced taxa in the gene trees?

Response: Thanks for pointing out this issue. We agree with the reviewer that misplace of *Acorus* in mitochondrial gene trees does not necessarily reflect HGT. In this revised version, we generated ultra-long ONT sequences and assembled a complete mitochondrial genome

(2.2 Mb). Our results suggested that high sequence divergence of *Acorus* mitochondrial protein-coding genes, with long branch lengths in d_N and d_S trees. The high sequence mutation rate might be the likely cause of the misplacements of *Acorus* in mitochondrial gene trees. Therefore, we have removed the inference of HGT and rewritten the results related to mitochondrial genome analyses.

Some specific notes to the trees:

atp1: *Acorus* is placed as sister to Alismatales and that entire clade appears misplaced. HGT from Alismatales to *Acorus* would not explain the placement.

ccmB: you could equally well infer this as HGT from *Acorus* to Spirodela.

ccmFN: a sister group placement of Ceratophyllum and *Acorus* does not necessarily mean HGT from Ceratophyllum to *Acorus*. Several positions in this tree are weird and cannot just be explained by one event of HGT.

cob: the support is not overwhelming. It just meets your criteria. Normally 75% bootstrap is considered moderate support.

nad4: with *Acorus* as sister to just some of the alismatids and the Alismatales being non-monophyletic (how would you explain the position of Spirodela) a HGT hypothesis is not straight forward.

Response: As above response, in this revised version, HGT is not inferred as the cause of gene tree incongruence between nuclear and mitochondrial genome, and there would be not problems for interpreting gene tree topology here.

rpl12: the tree is not shown

Response: Sorry for the missing. All 38 mitochondrial single-gene trees were provided.

Below I list a number of issues (some minor, some major):

#5 L34: “Interestingly, we discovered....”. No, you confirmed what was previously found.

Response: Thanks, we have deleted “we discovered that”.

#6 L35: “... likely helped *Acorus* to maintain their ancestral state....”. This really says nothing. Delete.

Response: We have deleted this sentence.

#7 L37-39: I suggest to be more suggestive. E.g. “... gains possibly associated with ecological”

Response: Thanks for your suggestion, changed.

#8 L39: gene families instead of just genes?

Response: Corrected.

#9 L52: “morphological discrepancies”. Better: “morphological variability”?

Response: Corrected.

#10 L55: “problem”. Better: “challenge”?

Response: Corrected.

#11 L66-67: Better:” ... the placement of Liliales relative to Asparagales and the sister group relationship of Poales”

Response: We have revised the sentence as “however uncertainties remain in the relative placement of Liliales to Asparagales and identification of the lineage sister to the Poales.”.

#12 L68-69: Is it fair still to say that the placement of *Acorus* is ambiguous? All other than mitochondrial data have in recent large studies consistently placed *Acorus* as sister to the other monocots.

Response: We have deleted the sentence and rewritten this part.

#13 L85-89: Too much detail. What ethnic groups believe or not is irrelevant. Just write that you prefer to recognize all five species.

Response: We revised the sentence and it now reads as “Another species *Acorus macrospadiceus*, described as a new species by two Chinese botanists, Wei and Li³⁹, was also confirmed as an independent species based on phylogenetic and metabolomic Analyses ⁴⁰. We adopted the narrow species circumscription and recognize five species in *Acorus* genus.”

#14 L99: so far you have used monocots and not monocotyledons

Response: Changed to monocots.

#15 L89-89: you should add the large transcriptome/genome based analyses (Harkess et al, 1KP ...)

Response: Thanks, we have added it.

#16 L103: what do you mean by gaining popularity?

Response: We have removed all the descriptions and discussion of HGT in this revised version.

#17 L100-105: Since you only attempt to investigate HGT of mitochondrial genes, it would be more relevant to cite papers about this phenomenon in angiosperms.

Response: We have removed all the descriptions and discussion of HGT in this revised version.

#18 L108: the phylogenetic relationships of monocots, except from the placement of *Acorus*, is not reported on at all in the manuscript (unless I missed something)

Response: We have rewritten the study objectives. “Hence, in the present study, the telomere-to-telomere genome of *Acorus gramineus* was generated with the objectives, i) to explore the patterns of genome evolution in early monocots, ii) to identify the likely cause of

gene tree incongruence between nuclear and mitochondrial genomes, and iii) to detect genomic footprints involving structure and ecological changes adapted to the wetland lifestyle of *Acorus*.”

#19 L109: here the taxon Acorales is mentioned for the first time. It should be introduced above that Acoraceae is the only member of the order.

Response: Revised as suggested. Please see L70 in the introduction. “*Acorus* (sweet flag) is the only genus of the Acoraceae family in the order Acorales, which are distributed from northern temperate to subtropical regions.”

#20 L110: basal!

Response: We have removed the usage of basal wherever applicable. All the “basal angiosperms” revised to “ANA grade”.

#21 L145: basal!

Response: We have removed the usage of basal wherever applicable. All the “basal angiosperms” revised to “ANA grade”.

#22 L150: *Ceratophyllum* should be in italics

Response: Yes, corrected.

#23 L150-151: no! two sister clades differentiate at the same time!

Response: We have deleted the interpretation “this means monocots are the earliest to differentiate from the ANA grade”. Now it reads as “The topology also showed monocots as the sister to the clade including magnoliids, eudicots and *Ceratophyllum*.”

#24 L155-158: “It should be noted....” This belongs to the discussion.

Response: Yes, removed to discussion (Line 331-335) as suggested.

#25 L159-176: This is a mix of introduction, results and discussion.

Response: We have restructured this paragraph. Only phylogenetic relationships were presented in the results. Descriptions about species were given in the introduction. Polyphyly of *Acorus calamus* and suggestion for taxonomic revision were moved to discussion section.

#26 L168-169: correct: that is morphologically distinct from the rest of the species

Response: Revised as suggested.

#27 L169: “The results....”. Where are the described results/tree?

Response: Fig. 4a was added to describe the results.

#28 L174-176: clumsy writing. Rephrase

Response: Revised. Now it reads as “In subsequent research, it will be necessary to collect samples from numerous individuals representing each species to elucidate the intrageneric relationship of *Acorus* and make necessary taxonomic revision”

#29 L179-181: incomplete sentence

Response: Revised as “A full-length chloroplast genome of 153,062 bp was assembled for *Acorus gramineus*, which was in the range of the previous reported *Acorus* chloroplast genome length of 152–154 kb”.

#30 L185: Here I would expect some general results from assembly of the mitochondrial genome.

Response: Added, Line 161.

#31 L186: rewrite “... constructed phylogenetic trees using....”

Response: We have rewritten this sentence as “A total of 38 mitochondrial genes were identified and used for phylogenetic reconstructions.”

#32 L188: rewrite: “... we inferred that 12 ...”

Response: Corrected.

#33 L190-191: Alismatales are also monocots. Perhaps say other monocots? The figure specifies Arecales! Make text and figure consistent.

Response: Corrected. This sentence has been revised as “In contrast, most other mitochondrial genes assigned *Acorus* at misplaced in other lineages of angiosperms, involving Alismatales (*atp1*, *ccmB*, *matR*, *nad4*, *rps3*, and *sdh4*), Poales (*cob*), magnoliids (*rps4*), eudicots (*rpl2*, *ccmFC*, and *cox2*), and Ceratophyllales (*ccmFN*) (Fig. 2d, Supplementary Fig. 9 and Supplementary Table 11).”.

#34 L191-193: “We observed....” Are you referring to your own concatenated analysis of mitochondrial genes? Where is this result shown?

Response: Yes, the results were from our own analyses. Phylogenetic trees were reconstructed based on single-gene alignments rather than concatenated dataset. Figs were provided in Supplementary Fig. 9. and cited in the main text.

#35 L193-196: 12 out of 38 genes are not most. The remarks about Amborella make no sense. Are you referring to you own data/trees or the previous publication about HGT. And what has it got to do with the taxonomic position. If this should be included at all it belongs to the discussion.

Response: Yes, the results were from our own analyses. All mitochondrial gene trees were provided in Supplementary Fig. 9. and cited in the main text.

#36 L214: monocot not monocots
Response: Corrected.

#37 L216: did not instead of didn't
Response: Corrected.

#38 L221-223: this is a nothing but a wild guess. If included at all, place it in the discussion.
Response: We have deleted this sentence.

#39 L241: basal!
Response: Changed to ANA grade.

#40 L246: "... four monocots species...": monocot not monocots. Do you mean 10 monocot species? Otherwise I do not understand what is meant.

Response: Changed to monocot. We used four species in last version of our manuscript. Our initial idea was that "Acorales and Alismatales that hold few total gene numbers than most other monocot species might have few specific genes". This is the case when we compared *Acorus gramineus*, *Spirodela polyrhiza*, *Zostera marina* and *Oryza sativa*. However, this is not a rule when we make comparison with more representatives (Fig. 5b) and hence we deleted the descriptions about specific gene families.

#41 L247: include "and" before Oryza
Response: Corrected.

#42 L248-254: here again you approach something that would fit a discussion, although some elaboration would be desirable.

Response: Please refer to the response to the above comment to the question #40. We deleted the descriptions and discussion about specific gene families.

#43 L257: delete "According to" to correct sentence
Response: Deleted.

#44 L260: Add "According to" in the beginning of this sentence
Response: Added.

#45 L282-283: The sentence makes no sense. Rewrite
Response: We have removed the sentence.

#46 L284: that instead of but
Response: Corrected.

#47 L289: monocot not monocots
Response: Corrected.

#48 L293: gene families not gene-family; "...showed few gene numbers in *Acorus gramineus*, but gene numbers of some pathways showed" change to "... were contracted in *Acorus gramineus*, others showed...."
Response: Revised as "Although many gene families and metabolic pathways were contracted in *Acorus gramineus*, others still showed expansions, such as some energy metabolism, photosynthesis and oxidative phosphorylation pathways."

#49 L302-305: the end of this sentence makes no sense
Response: Revised as "Interestingly, gene expansion was also detected in terpene synthases b (TPS-b) subfamily in *Acorus gramineus*. The TPS-b subfamily gene number is relatively lower in monocots, and even absent in several lineages, such as *Oryza sativa*, *Spirodela polyrhiza* and *Zostera marina* (Fig. 5e), whereas, there were 6 TPS-b genes identified in *Acorus gramineus* (Supplementary Table 25)."

#50 L320: ".. of monocotyledons is ..." change to "....being...."
Response: We have revised this sentence as "Eudicot stems are arranged in a ring with a vascular cambium, whereas vascular bundles scattered throughout in monocot stems".

#51 L320: no such group as dicotyledons. Do you mean eudicots or anything else than monocots?
Response: Corrected to eudicot.

#52 L333-336. In what way is the architecture of *Acorus* simple compared to grasses?
Response: We have deleted this sentence and rephrased the results and discussion parts.

#53 L336-337: where on earth did this claim come from? I do not see anything "likely" here!
Response: We have deleted this sentence.

#54 L336-338: this belong to the discussion, if anywhere at all.
Response: Yes, we have removed this part to discussion. Now, it reads as "*Acorus gramineus* holds fewer WRKY and R genes that play important roles in stress resistance and innate immunity, indicating adaptation to aquatic/wetland habitat where plants suffer from far fewer pests and diseases than land plants."

#55 L343: what do you mean by “filling the key evolutionary gap” ?

Response: We have deleted the ambiguous statement.

#56 L345-346: which “gene regions” (not mitochondrial)? First you say that *Acorus* has been placed in different positions, then you only mention placement as sister to the rest of the monocots. Thus, the sentence does not quite make sense. Write *Acorus* in italics.

Response: We have rewritten this paragraph. First, we introduced in this study nuclear and chloroplast phylogenetic trees placed *Acorus gramineus* as the sister to the remaining monocots, and then we mentioned that mitochondrial genes assigned *Acorus* at a misplacement in most single-gene trees. Finally, we explained the likely cause of the gene tree conflicts might be high sequence divergence of mitochondrial genes.

#57 L348: delete “A”

Response: Deleted.

#58 L349: delete “A”

Response: Deleted.

#59 L352-355: This is circular reasoning. Yes, HGT from Alismatales would explain why mitochondrial genes trees place *Acorus* in Alismatales, but to when HGT has been inferred from those gene trees the reasoning becomes circular.

Response: All statements about HGT were removed in this revised version.

#60 L358-360: Not clear what is meant here? Which species? Maybe use names to clarify.

Response: We have added the species details and rewritten the discussion about intrageneric relationships of *Acorus* (Line 352-366).

#61 L363-366: wrong grammar in this sentence

Response: Revised and restructured as “Annotation identified 23,207 genes in *Acorus gramineus*, together with representatives in Alismatales (*Spirodela polyrhiza* and *Zostera marina*) have relatively fewer gene number than other monocots. This gene number variation pattern is specifically reflected in gene families and transcription factors. Gene number expansion in most monocots might be the results of the vast gene-retain after τ WGD event that is absent in Acorales and Alismatales.”

#62 L370-373: but if *Acorus* and alismatids had other WGDs wouldn't they still be expected to a high number of TFs?

Response: Beside the τ WGD event, other monocots also have lineage specific WGDs as Acorales and Alismatales. Therefore, the lack of the τ WGD event in Acorales and Alismatales might be the likely reason of a low number of TFs.

#63 L377: *Acorus* is not “early”.

Response: We have deleted the word “early”.

#64 L378-383: How are any of these structures “simple” compared to e.g. *Oryza*?
Response: We have deleted the ambiguous description “simple architecture”.

#65 L384: “grows” rather than “groups”?”

Response: Revised as “*Acorus gramineus* holds fewer WRKY and R genes that play important roles in stress resistance and innate immunity, indicating adaptation to aquatic/wetland habitat where plants suffer from far fewer pests and diseases than land plants”.

#66 L393: “dicots”: eudicots or anything else than monocots?

Response: We have rewritten the discussion about R genes. Now it reads as “TNL genes were not recognized in monocots including *Acorus gramineus*, consistent with previous report. TNL subclass genes exist in *Amborella*, *Nymphaea*, magnoliids and eudicots, suggesting TNLs might present in the ancestor of angiosperms with subsequent loss in the ancestor of monocots.”

#67 L401: use full names. Does *Spirodela* grow in shady habitats? That is not my impression.

Response: Thanks for pointing out this mistake. Yes, *Spirodela polyrhiza* does not grow in shady habitats. Only two LHCB1 genes were identified in *Spirodela polyrhiza*. It was not an example to show LHCB1 expansion and hence we deleted it.

#68 L402: The habitat preference of *Acorus* and alismatids is not a hypothesis.

Response: Deleted.

#69 L406: basal!

Response: Deleted.

#70 L407: “a transition from land to water” in what way? Are you implying something evolutionary or do you just mean that *Acorus* grows in a transition zone?

Response: Deleted. Discussion about WRKY TFs now reads as “*Acorus gramineus* hold fewer WRKY and R genes that play important roles in stress resistance and innate immunity, indicating adaptation to aquatic/wetland habitat where plants suffer from far fewer pests and diseases than land plants”.

#71 L423: ground not grounded

Response: Corrected.

#72 L434-435: cite table S8b for voucher info

Response: Added.

#73 L452: close not closed
Response: Corrected.

#74 L460: lacking right parenthesis
Response: Added.

#75 L461-463: something is wrong with this sentence. Rewrite
Response: Revised as “Due to the presence of heterozygosity, it is possible that the contigs representing the two haplotypes of a given region of the genome might be assembled as two independent primary contigs, rather than as a single contig and an associated haplotig.”

#76 L493: similarity-based instead of homology-based?
Response: Corrected.

#77 L505: maximum not max
Response: Corrected.

#78 L528-533. Long sentence. Split in two.
Response: Revised as “Evidence-based gene prediction was conducted by aligning all RNA-seq data generated herein against the assembled genome using Hisat2 (v2.0.4). cDNAs were identified by a genome-guided approach using StringTie (v1.2.2) and then mapped back to the genome using PASA (version 2.3.3). The resulting cDNA sequence assembly by Trinity (v2.6.6) were aligned to the Acorus gramineus genome sequences using BLAT (v34x12).”

#79 L560-566. Long sentence. Split in two.
Response: Revised as “For each single-copy gene orthogroup, we first performed multiple amino acid sequence alignments by MAFFT (v.7.471), and then DNA sequences were aligned according to the corresponding amino acid alignments using PAL2NAL (v14.1), followed by gap position removal using trimal (v1.4.1). Then each gene tree was constructed by IQTREE (v2.0.5) that automatically selected the best-fit substitution model using ModelFinder.”

#80 L566: “all gene trees” instead of “the whole gene trees”
Response: Corrected.

#81 L573: all species not the whole species
Response: Corrected.

#82 L576-578: something is wrong with this sentence. Rewrite

Response: Revised as “The 612 low-copy genes were used to generate phylogenetic trees via the concatenation and coalescent methods mentioned above.”

#83 L583: all species not the whole species. Some word(s) is also missing in the sentence

Response: Revised to “The amino acid sequences from all species were aligned using BLASTP with an e-value of $1e-5$, and then grouped using OrthoFinder (v 2.3.7).”

#84 L601: parameters not paraments

Response: Corrected.

#85 L621: Heading: this section is also about assembly and about the mitochondrial genome

Response: We now split this section into two: “Assembly of chloroplast and mitochondrial genomes” and “Phylogenetic analyses of chloroplast and mitochondrial genes”.

#86 L639: genomes not plants, mitochondrial not mitochondria

Response: Corrected.

#87 L651: mitochondrial not mitochondria

Response: Corrected.

#88 L651-652: “...trees were concatenated.” Do you mean alignments?

Response: Yes, genes were concatenated to reconstruct phylogenetic tree. Given most mitochondrial genes assigned *Acorus* in misplacements, we presented all single-gene trees and deleted the concatenation result in this revised version.

#89 L652-653. Superfluous? Isn't that described above?

Response: Revised as “All the protein-coding genes were aligned and trimmed following the same pipeline used for nuclear trees.”

#90 L651-662: unclear. How do you use the tree from the concatenated data set? Don't you use the individual gene trees to infer HGT?

Response: Thanks for pointing out this. Given most mitochondrial genes assigned *Acorus* in misplacements, we presented all single-gene trees and deleted the concatenation result in this revised version. Statements about HGT were deleted.

#91 L700: Arabidopsis

Response: Corrected.

#92 L701: no punctuation after Acorus

Response: Corrected.

#93 L705: sequence not sequencing

Response: Corrected.

#94 L707, 713, 714: hmmsearch or hmmersearch?

Response: Checked and corrected it as hmmsearch.

#95 L709: no punctuation after Acorus. Add italics

Response: Corrected.

#96 L713: genes not gene

Response: Corrected.

#97 L719: sequence not sequencing

Response: Corrected.

#98 L764: Explain the asterisk/star in 2b (or delete it from figure)

Response: Added "The red star highlights the position of *Acorus gramineus*."

#99L767: No arrowheads in figure!

Response: Revised the figure legend as "Summary of the results of mitochondrial gene trees, showing the misplacements of *Acorus* in individual gene analyses."

#100 L774: Explain the abbreviated taxon names used in 3b.

Response: Added in the corresponding Fig 4b.

#101 L776: basal!

Response: Deleted "the basal angiosperm".

I have not read all supplementary files carefully, but here are some corrections:

#102 Suppl. Fig 4. Explain the colours of quartet scores

Response: Added.

#103 Suppl. Fig. 9 text: correct "Mononcots" to "Monocots". Chloranthales mentioned twice. Change one to Ceratophyllum (or Ceratophyllales)

Response: Corrected.

#104 Suppl. Fig. 11. Write names in full (*Amborella trichopoda*, *Acorus gramineus*)
Response: Corrected.

#105 Suppl. Fig 12. The main legend just mentions *Acorus*, but the figure includes other taxa. Write names in full.
Response: Corrected.

#106 Suppl. Figs 15, 18, 20, 21, 22, 24. Write names in full
Response: Corrected.

Reviewer #2 (Remarks to the Author):

Guo et al. present a resubmission of the *Acorus gramineus* genome and accompanying analyses focused on identifying HGT and investigating genes associated with potential environmental adaptation. Though the genome of *Acorus* represents a valuable position in the angiosperm phylogeny to have a complete genome, the overall findings of the paper seem to be confirmatory to other papers. The claims of HGT in the mitochondrial genome were easier to assess given the trees and alignments, but there are issues in those analyses.

#107 The evidence used to suggest horizontal transfer of genes in the mitochondrial genomes leaves me with more questions than answers. The mitochondrial genome of land plants, especially angiosperms, is known to act as a DNA sponge and incorporate various other genome components into them. For example, the mitochondrial genome of *Amborella* has been demonstrated to have a sizeable portion of chloroplast genomes from other species in it. Looking at the alignments of the different mitochondrial genes presented in the supplemental figures makes me question the validity of the function of these genes. Each of the *Acorus* genes looks extremely different than the other taxa in their alignments, often being shorter than the other genes. This leads me to ask if these genes are translatable to a fully functioning protein. Are these genes indeed the result of HGT but there still exists a copy in the mitochondrial genome that is derived from vertical transmission?

Response: Thanks for pointing out this issue. We agree with the reviewer that misplacement of *Acorus* in mitochondrial gene trees does not necessarily reflect HGT. In this revised version, we generated ultra-long ONT sequences and achieved a complete mitochondrial genome (2.2M in size). As the reviewer mentioned that *Acorus* genes look extremely different than the other taxa in their alignments. Our results suggested that the high sequence divergence of *Acorus* mitochondrial protein-coding genes, with long branch lengths in d_N and d_S trees. The high sequence mutation rate might be the likely cause of the gene misplacements in mitochondrial genes. Therefore, we have removed the inference of HGT and rewritten the mitochondrial genome part.

#108 There is also the question of a few SNPs being used to suggest HGT when these could be the result of homoplasy.

Response: We have removed the descriptions and discussion about HGT in this revision. We added the discussion of a few SNPs of *Acorus* that shared with other lineages. "The

high sequence divergence specifically reflected in single-gene alignments. There are a large number of mutation sites identified in *Acorus* in contrast to little or no detectable sites in other angiosperms (Supplementary Fig. 10). Out of these mutation sites, a small proportion of indels coincidentally were shared among distantly related lineages and contributed to 'phylogenetic synapomorphies' that likely resulting the misplacements of *Acorus* in mitochondrial genes."

#109 The methods say that poorly aligned regions were trimmed using trimal, but the alignments presented in the supplemental figures show large regions that are not aligned well. There seems to be something going on with these alignments that is questionable for further phylogenetic analysis.

Response: Yes, we agree with the reviewer that large regions are not aligned well due to a high number of mutation sites in *Acorus*. Therefore, we noted in the discussion that mitochondrial genes of *Acorus* are not reliable for future phylogenetic analyses. Please refer to Line 348: "The rapid mutational rates also explained the misplacements of *Acorus* in single gene tree in previous studies using mitochondrial data, highlighting the need to exclude mitochondrial genes in future phylogenetic studies involving *Acorus*."

Other areas with questions/comments

Abstract

#110 Repetitive with "discovered". Again, this paper confirms and not "discovers" the absence of tau in acorus history.

Response: We changed the sentence as "Interestingly, the Acorales did not experience the tau (τ) whole-genome duplication like other monocot clades".

#111 Line 52: "Discrepancy" is the wrong word. I think you mean "diversity"

Response: Yes, changed as "diversity".

#112 Line 53: "Uphold" does not seem like the right word. "Provide" or a word like it is probably better.

Response: Corrected.

#113 Lines 66-67: Is there something missing here? Is this saying the uncertainty is in what clade is sister to the Poales? If so, something like "problems remain in the relative placement of Liliales to Asparagales and identification of the lineage sister to the Poales" would be a better way of saying this.

Response: Revised as "These investigations achieved well-supported monocot phylogenies, with the majority of clades completely resolved, however uncertainties remain in the relative placement of Liliales to Asparagales and identification of the lineage sister to the Poales".

#114 Line 69: The placement of *Acorus*/Acoraceae/Acorales—which is all equivalent—has been established for quite some time. It is true *Acorus* was once included in Araceae, but molecular phylogenetics has placed it sister to the rest of the monocots for many years. This was a critique I had in the previous version of this manuscript. This statement seems to be overstating the uncertainty of *Acorus* placement in the phylogeny.

Response: Thanks for pointing out this. We have removed the statement “placement remains ambiguous since its establishment”. We also rewrote this part in the introduction. First introduced nuclear and chloroplast phylogenetic trees in previous studies placed *Acorus* as the sister to the remaining monocots. Then mention mitochondrial genes assigned *Acorus* at a misplacement. Further came to one of the study objectives “to identify the likely cause of gene tree incongruence between nuclear and mitochondrial genomes”.

#115 Lines 90-112: This section is completely missing transcriptomic data that has been used to look at the phylogeny of this group. This is not a complete and accurate depiction of our understanding of the evolutionary placement of *Acorus*. The representation is such to make this seem more questionable in our understanding.

Response: We rewrote this paragraph as response to comment #114. Please refer to Line 91-106.

#116 Line 119: Missing “and” to link the description of the Nanopore and short read data. Currently the text just separates with a comma.

Response: Added “and”.

#117 Line 122: Hi-C data is more helpful if given in total read pairs instead of total base pairs.

Response: Added read pairs.

#118 Line 150: Monocots and the rest of the angiosperms diverged from each other at the same time, based on the shared common ancestor. It is better to say that monocots diverged prior to the diversification of eudicot, magnoliids, and *Ceratophyllum*. The term “basal angiosperms” is not a correct term since this is referring to the members of the ANA grade.

Response: Deleted the term “basal angiosperms”. We have revised the sentence as “The topology also showed monocots as the sister to the clade including magnoliids, eudicots and *Ceratophyllum*.”

#119 Lines 174-176: Language of this sentence needs to be fixed to improve the grammar.

Response: Revised. Now it reads as “In subsequent research, it will be necessary to collect samples from numerous individuals representing each species to elucidate the intrageneric relationship of *Acorus* and make necessary taxonomic revision”

#120 Lines 177-195: Language throughout this section needs to be check for clarity and grammar.

Response: We now deleted all descriptions and discussion about HGT. Rewrote the mitochondrial genome and phylogenetic analyses. Please refer to Line 161-210.

#121 Lines 220-222: I believe this sentence is trying to invoke the idea that there are several WGD events found around the timing of the K-Pg boundary suggesting these events are related to post K-Pg success. Given the 20 my between the event and the K-Pg, I would say this is too spurious to make a connection.

Response: We deleted this sentence.

#122 Lines 246-252: The language here is unclear. Are these meant to be gene families that are only found in these taxa? What does that mean exactly? These families are likely related to some other families—this seems like a lot of genes to be de novo in these taxa. Is this a case of gene family circumscription being too conservative?

Response: Our initial idea was that “Acorales and Alismatales that hold few total gene numbers than most other monocot species might have few specific genes”. This is the case when we compared *Acorus gramineus*, *Spirodela polyrhiza*, *Zostera marina* and *Oryza sativa*. However, this is not a rule when we make comparison with more representatives (Fig. 5b) and hence we deleted the descriptions about specific gene families.

Reviewer #3 (Remarks to the Author):

In the revised manuscript, the authors have corrected and addressed some of the raised concerns. I had two major issues with the manuscript: (i) The possible misassembly of the genome and (ii) weak arguments supporting the horizontal gene transfer (HTG) hypothesis. I am completely satisfied by the authors' response to the first concern but not to the second.

#123 The first concern was about the distribution of genes/TEs along the chromosomes which in the previous manuscript was atypical and did not follow what is expected in plant genomes. This first issue was due to genome misassembly which has now been corrected by the authors through a new Hi-C experiment and genome reassembly.

Response: Thanks for the comments. We have improved the assembly to a telomere-to-telomere quality genome by adding ultra-long Nanopore and HiFi data.

#124 2- The second issue had to do with horizontal transfers. I pointed out in the previous version of the manuscript that HGTs were not well supported. I am still not convinced that the observed phylogenetic incongruities of some mitochondrial genes are indeed due to HGTs.

Response: Thanks for pointing out this issue. We agree with the reviewer that misplacement of *Acorus* in mitochondrial gene trees does not necessarily reflect HGT. In this revised version, we sequenced ultra-long ONT sequence and achieved a complete mitochondrial genome (2.2M). We found that that *Acorus* genes look extremely different than the other taxa in their alignments. Our results suggested that high sequence divergence of *Acorus* mitochondrial protein-coding genes, with long d_N and d_S branch lengths. The high sequence mutation rate might be the likely cause of the gene misplacements in mitochondrial genes. Therefore, we have removed the inference of HGT and rewritten the mitochondrial genome part.

The number of conserved sites in the multiple alignments provided that lead to phylogenetic incongruities is very small. For example: *nad5* (2/1000 sites), *rps4* (2/1400 sites), *ccmFC* (3/1500 sites), *ccmFN* (4/2200)...etc. What are the probabilities that these mutations occur by chance between highly divergent lineages (tens of millions of evolution)? The answer to this question is essential to determine whether these genes were acquired by horizontal gene transfer or not.

Response: We agree that the small number of conserved sites likely occur by chance between *Acorus* and distantly related lineages. We added corresponding statements in the discussion in Line 341-346. "The high sequence divergence specifically reflected in single-gene alignments. There are a large number of mutation sites identified in *Acorus* in contrast to little or no detectable sites in other angiosperms (Supplementary Fig. 10). Out of these mutation sites, a small proportion of indels coincidentally were shared among distantly related lineages and contributed to 'phylogenetic synapomorphies' that likely resulting the misplacements of *Acorus* in mitochondrial genes."

For this reason, I asked the authors to compare the sequence similarity (Ks) of orthologous mitochondrial genes (for species with sequenced mitochondrial genomes) of the species involved in HGT with that of the transferred genes. If the rate of synonymous mutations is significantly lower for the transferred genes, this constitutes a strong argument for HGT together with the phylogenetic incongruence.

Response: Thanks for pointing out this issue. As the response to comment #124, we assessed the sequence divergence by estimating d_N and d_S values. The long branches suggested high sequence divergence. Therefore, we deleted all statements about HGT in the revision and inferred high mutation rate as the likely reason for phylogenetic incongruence.

Minor but important points

#125 Legend for Figure 1: Do you mean relative density or number? In any case, the authors should add a key for each track.

Response: We have added the key for each track in Fig1b and rewrote the Figure legends.

#126 Supplementary Table 7a:

- DNA vs. RNA TEs: Please correct Class I and Class II
- The word "Number" should be deleted because the authors do not provide numbers but repeat Size and percentage (Table 7a)

Response: Thank you for your advice, we have corrected to Class I and Class II and deleted the word "Number".

#127 Supplementary Table 7b:

The authors should simplify the classification system. For example, for LTRs-retrotransposons, you have Copia, Gypsy and LTRs. Copia and Gypsy are LTRs. There are many problems with the TE superfamilies provided. The other categories of LTRs are generally not found in plant genomes (e.g. Bel-Pao). In addition, there are many redundant superfamily classes in the table: Caulimovirus and Caulimoviru, Gypsy and Gypsy-Cigr. This is also the case for LINEs and Class II elements (TE DNA). Authors should refer to the

widely used Wicker TE classification system for eukaryotic genomes (<https://trep-db.uzh.ch/TEClassification.php>).

Response: Revised as suggested. We have downloaded the data from the database and re-identified it, and then categorized it with the classification of this database

#128 There are still many typos in the figure captions. I will not list them all, but for example, in the caption of Supplementary Figure 10.

Response: Thanks. We carefully checked and corrected the typos throughout the manuscript, figures and tables.

Reviewers' Comments:

Reviewer #3:

Remarks to the Author:

the authors addressed the issues I had raised regarding the previous manuscript